# ROBUST FEDERATED INFERENCE

**Akash Dhasade**[1]**, Sadegh Farhadkhani**[1]**, Rachid Guerraoui**[1]**, Nirupam Gupta**[2]**,**
**Maxime Jacovella**[1]**, Anne-Marie Kermarrec**[1]**, Rafael Pinot**[3]
[1]EPFL, Switzerland    [2]University of Copenhagen, Denmark
[3]Sorbonne Université and Université Paris Cité, CNRS, LPSM, France
`akash.dhasade@epfl.ch   nigu@di.ku.dk   pinot@lpsm.paris`

## ABSTRACT

Federated inference, in the form of one-shot federated learning, edge ensembles, or federated ensembles, has emerged as an attractive solution to combine predictions from multiple models. This paradigm enables each model to remain local and proprietary while a central server queries them and aggregates predictions. Yet, the robustness of federated inference has been largely neglected, leaving them vulnerable to even simple attacks. To address this critical gap, we formalize the problem of robust federated inference and provide the first robustness analysis of this class of methods. Our analysis of averaging-based aggregators shows that the error of the aggregator is small either when the dissimilarity between honest responses is small or the margin between the two most probable classes is large. Moving beyond linear averaging, we show that the problem of robust federated inference with non-linear aggregators can be cast as an adversarial machine learning problem. We then introduce an advanced technique using the DeepSet aggregation model, proposing a novel composition of adversarial training and test-time robust aggregation to robustify non-linear aggregators. Our composition yields significant improvements, surpassing existing robust aggregation methods by $4.7 - 22.2\%$ in accuracy points across diverse benchmarks.

## 1 INTRODUCTION

Over the past several years, concepts such as *one-shot federated learning* (OFL) (Dai et al., 2024; Diao et al., 2023; Zhang et al., 2022; Guha et al., 2019), *edge ensembles* (Malka et al., 2025; Shlezinger et al., 2021), and *federated ensembles* (Allouah et al., 2024a; Hamer et al., 2020) have gained traction in collaboratively performing inference from several client-local models. More recently, the availability of diverse open-source large language models (LLMs) has spurred interest in aggregating outputs from multiple models to answer a given query, giving rise to sophisticated *LLM ensembles* that leverage complementary strengths of individual models (Tekin et al., 2024; Wang et al., 2024; Jiang et al., 2023). Despite being introduced under different names, these techniques share a common principle: combining predictions from multiple client-held models to produce a single output. In this paper, we collectively refer to these approaches under the terminology of *federated inference*. In this setting, clients retain proprietary (locally trained) models, while a central server queries them for inference as illustrated in Figure 1. The individual predictions are then aggregated into a final prediction, either using averaging-based aggregations (Dai et al., 2024; Zhang et al., 2022; Gong et al., 2022) or server-side aggregator neural networks (Allouah et al., 2024a; Wang et al., 2024).

While federated inference is gaining traction, its robustness to model failures and poisoned outputs remains largely overlooked in the literature, despite some initial preliminary study (Liu et al., 2022). This gap is critical for two reasons: *(i)* failures and errors are practically unavoidable, and *(ii)* existing work in robust statistics and Byzantine-robust machine learning (ML) demonstrates that undefended models are inherently vulnerable, even to relatively simple attacks (Guerraoui et al., 2024; Diakonikolas & Kane, 2023). It is therefore of paramount importance to clearly define the potential threats that may arise in federated inference, so as to prevent a technological advantage from becoming a significant vulnerability. In this paper, we take a step toward closing this gap by defining potential failure modes of federated inference and presenting the first robustness analysis of this class of schemes.

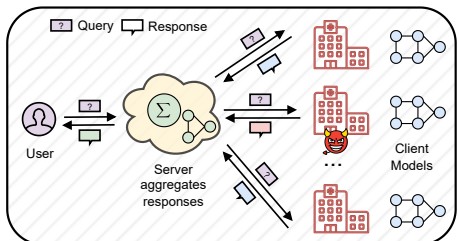

Figure 1: Federated Inference

Table 1: Evaluation of robust elements in a setup with $n = 17, f = 4$. We report the worst-case accuracy across 5 different attacks (see Appendix G.4 for details).

| DeepSet | CWTM | Adv. Tr. | CIFAR-10 | CIFAR-100 | AG-News |
|---------|------|----------|----------|-----------|---------|
| ✓ | ✗ | ✗ | $46.0 \pm 3.9$ | $47.4 \pm 3.1$ | $76.4 \pm 1.9$ |
| ✓ | ✓ | ✗ | $47.0 \pm 3.8$ | $67.0 \pm 0.7$ | $76.7 \pm 2.2$ |
| ✓ | ✗ | ✓ | $48.6 \pm 6.7$ | $65.1 \pm 0.8$ | $76.7 \pm 0.9$ |
| ✓ | ✓ | ✓ | $\mathbf{51.4 \pm 2.2}$ | $\mathbf{68.0 \pm 0.8}$ | $\mathbf{77.5 \pm 1.2}$ |

## 1.1 MAIN CONTRIBUTIONS

**Problem formulation and analysis of the averaging aggregator.** We formalize the problem of robust federated inference for the first time. We consider a system of $n$ clients, each with a local data distribution and probit-valued classifier, where the outputs of up to $f < \frac{n}{2}$ clients may be arbitrarily corrupted at inference time. Our goal is to design aggregation schemes that remain accurate with respect to the global data distribution despite the corruptions. When the server uses an averaging-based aggregation (Dai et al., 2024; Zhang et al., 2022; Gong et al., 2022), a natural way to robustify is to substitute it with a robust averaging scheme (Allouah et al., 2023) which ensures that the output of the aggregator is an estimate of the average of the honest probits. Prominent examples include coordinate-wise trimmed mean (CWTM), coordinate-wise median (CWMed), *etc*. (Guerraoui et al., 2024). However, we show that robust averaging may prove insufficient since the aggregator's output can be sufficiently close to the honest average, yet produce a misclassification. Nevertheless, in the cases where averaging suffices, we derive formal robustness certificates for federated inference. Our analysis shows that the error of the aggregator depends upon the fraction of corruptions $f/n$, the margin between the top two classes, and the dissimilarity between the outputs of different clients.

**Robust Inference as an adversarial ML problem.** Beyond averaging, recent work has shown that non-linear trained aggregators (*e.g.*, neural networks) often outperform averaging-based aggregators in the uncorrupted setting (Allouah et al., 2024a; Wang et al., 2024). When using such aggregators, we show that the problem of robust inference can be cast as an adversarial learning problem over the probit-vectors. Encouragingly, contrary to the difficulty of standard adversarial ML problems in the image space (Goodfellow et al., 2014), we show that the adversarial problem on probit-vectors can be more reliably addressed, thanks to the structure of the input space where each corrupted vector is confined to a probability simplex over the number of classes. Yet, naively leveraging adversarial training (Madry et al., 2018) to solve the problem remains computationally intractable due to the high cardinality of permuting through different choices of adversarial clients at training time, since *any* $f$ (unknown) clients out of $n$ can be malicious.

**Robust DeepSet Aggregator.** To alleviate this issue, we propose to use a neural network aggregator based on the DeepSet model (Zaheer et al., 2017), an architecture which is invariant to the order of inputs. Specifically, since the aggregator's output can be independent to the order of clients, leveraging DeepSets enables us to reduce the search of choosing $f$ adversaries to $\binom{n}{f}$ instead of requiring to permute through them *i.e.*, $\binom{n}{f} \times f!$. In practice, we show that adversarial training by sampling any $N$ choices of adversaries where $N \ll \binom{n}{f}$ suffices to achieve good performance with DeepSet. To further reduce the sensitivity of DeepSet to corrupted probits, we propose a composition of robust averaging with DeepSet at inference-time, which significantly boosts empirical performance. In particular, we show that this composition may only be applied at inference-time, preventing any escalation of training costs during adversarial training. By combining robust elements from both the adversarial ML and robust ML literature in our novel composition, we achieve the state-of-the-art (SOTA) performance for federated inference as illustrated in Table 1.

**Empirical validation.** To rigorously evaluate defenses, we design a new attack called the Strongest Inverted Attack (SIA) that challenges existing defenses. We conduct extensive experiments on three datasets (CIFAR-10, CIFAR-100, and AG-News) covering both vision and language modalities as well as diverse model families (ResNet-8, ViT-B/32, DistilBERT). Our approach yields a $4.7$–$22.2\%$ points improvement over existing methods across a suite of 6 different attacks, including SIA.

## 1.2 RELATED WORK

**Federated Inference.** Collaborative inference through client ensembles has been explored in one-shot Federated Learning (FL) (Guha et al., 2019; Gong et al., 2022; Dai et al., 2024) and decentralized edge networks (Shlezinger et al., 2021; Malka et al., 2025), motivated by communication efficiency, reduced training cost, or proprietary nature of client models accessible only via black-box inference. Another common scenario is when clients already possess trained models which are offered in model market scenarios (Li et al., 2021). Yet, the robustness of federated inference has received limited attention. An initial preliminary work was conducted by Liu et al. (2022) in the vertical federated learning setting, where they try to reconstruct the underlying uncorrupted responses using an autoencoder and a block-sparse optimization process. Their method, COPUR, purifies responses before aggregation on the server. However, purification alone is vulnerable to stronger attacks and highly sensitive to input magnitudes, since it operates in the logit space rather than probits.

**Byzantine Distributed Learning.** Byzantine robustness has become a central topic in distributed ML (Alistarh et al., 2018; Chen et al., 2017; Farhadkhani et al., 2022; Guerraoui et al., 2024), where malicious clients may send arbitrary updates during training. Our setting differs in focusing on inference rather than training, and in not assuming fixed Byzantine identities (*cf.*, (Dorfman et al., 2024)). While the objectives thus diverge, a key idea in Byzantine distributed learning, namely robust averaging, remains relevant to our analysis. Indeed, robust averaging techniques can be adapted to improve the resilience of federated inference in the presence of corrupted client outputs, at the cost of a some technicalities, as demonstrated in Section 3.

**Federated Distillation and Robust Voting.** In FL, many works propose to share logits on a public dataset instead of gradients to improve communication efficiency and privacy (Fan et al., 2023; Gong et al., 2022; Sattler et al., 2021). Recent work addresses adversarial logits by proposing robust aggregation methods (Roux et al., 2025; Li et al., 2024; Mi et al., 2021), but these approaches rely on assumptions inapplicable to our setting. For example, EXPGUARD (Roux et al., 2025) tracks client behavior across rounds, whereas in our case clients may act arbitrarily at each inference. Similarly, FEDMDR (Mi et al., 2021) requires clients to report accuracy on a public dataset to weigh contributions, information unavailable in our setup. A parallel line of work studies robust voting, where voters/clients cast scores that are aggregated to resist malicious participants, either in general settings (Allouah et al., 2024b) or in federated learning specifically (Chu & Laoutaris, 2024; Cao et al., 2022). These approaches, however, typically assume fixed client identities or some control over the training process, assumptions which do not hold in our setting.

## 2 PROBLEM OF ROBUST FEDERATED INFERENCE

We consider a classification task, mapping an input space $\mathcal{X}$ to an output space $\mathcal{Y} = [K] := \{1, \ldots, K\}$, and a system comprising $n$ clients, each with a local data generating distribution $\mathcal{D}_i$ over $\mathcal{X} \times \mathcal{Y}$. For each client $i \in [n]$, we are given a probit-valued regressor $h_i$ mapping $\mathcal{X}$ to the simplex $\Delta^K := \{z \in [0,1]^K \mid \sum_{k \in [K]} z_k = 1\}$, which plays the role of a local classifier. In this context, the goal of a federated inference scheme is to design a mapping $\psi : (\Delta^K)^n \to [K]$ that aggregates clients' local probits in order to minimize the expected prediction error on the mixture of distributions $\mathcal{D} := \frac{1}{n} \sum_{i=1}^n \mathcal{D}_i$. Specifically, denoting $\mathbf{h}(x) := (h_1(x), \ldots, h_n(x))$, we seek an aggregator $\psi_o$, from a set of candidate aggregators $\Psi$, that minimizes the *federated inference risk*

$$\mathcal{R}(\psi) := \mathbb{E}_{(x,y) \sim \mathcal{D}} [\ell_\psi(x,y)], \quad \text{where} \quad \ell_\psi(x,y) := \mathbb{1}\{\psi(\mathbf{h}(x)) \neq y\}. \qquad (1)$$

A typical example of aggregation is $\psi(\mathbf{h}(x)) := \arg\max_{k \in [K]} \left[\frac{1}{n} \sum_{i=1}^n h_i(x)\right]_k$, where $[\cdot]_k$ denotes the $k$-th coordinate of the vector and where the $\arg\max$ breaks ties arbitrarily. More generally, however, $\Psi$ may involve non-linear aggregation schemes. In fact, prior work (Allouah et al., 2024a; Wang et al., 2024) has shown that such aggregation strategies yield higher accuracy expectation, compared to any individual classifier $h_1, \ldots, h_n$.

**Robust federated inference.** We are interested in solving the problem of *robust federated inference*, wherein a fraction of the clients' is subject to corruption prior to aggregation. Specifically, we consider a scenario wherein for each query $x \in \mathcal{X}$, up to $f < n/2$ of the $n$ clients (of hidden identity) can return arbitrarily corrupted vectors in the probit space $\Delta^K$. Our goal is to design an aggregation

scheme that yields high accuracy, despite such corruptions. In what follows, for any input $x \in \mathcal{X}$, we denote by $\Gamma_f(x)$ the set of all possible probits after up to $f$ corruptions, i.e.,

$$\Gamma_f(x) := \left\{ \mathbf{z} = (z_1, \ldots, z_n) \in (\Delta^K)^n \mid \exists H \subseteq [n], |H| \geq n - f, \forall i \in H, z_i = h_i(x) \right\}. \quad (2)$$

Formally, we seek an aggregator $\psi_{\mathrm{rob}} \in \Psi$ minimizing the *robust federated inference risk*, given by

$$\mathcal{R}_{\mathrm{adv}}(\psi) := \mathbb{E}_{(x,y) \sim \mathcal{D}} \left[ \ell_\psi^{\mathrm{adv}}(x, y) \right], \quad \text{where} \quad \ell_\psi^{\mathrm{adv}}(x, y) := \max_{\mathbf{z} \in \Gamma_f(x)} \mathbb{1} \left\{ \psi(\mathbf{z}) \neq y \right\}. \quad (3)$$

In the following, we show that robust federated inference risk can be upper bounded by federated inference risk and an overhead resulting from probit corruptions. Thereby we show that if an optimal aggregator (with respect to the original federated inference risk (1)) is apriori known, then robust ensembling can be achieved by designing an aggregator that aims to minimize disagreement with the optimal aggregator, in the presence of corruptions. In doing so, consider an aggregation scheme $\psi_{\mathrm{o}}$ minimizing (1), i.e., the expected learning error without corruption. This aggregator $\psi_{\mathrm{o}}$ represents an *oracular aggregator*, i.e., optimal in the hypothetical scenario when we have access to the uncorrupted probits. Using $\psi_{\mathrm{o}}$ as reference for robustness, we can bound robust federated inference risk for an aggregator $\psi_{\mathrm{rob}}$ as per the following lemma (which we prove in Appendix A).

**Lemma 1.** *For any $x \in \mathcal{X}$, let $\hat{y}_o = \psi_o(\mathbf{h}(x))$. Then,*

$$\mathcal{R}_{adv}(\psi_{rob}) \leq \mathcal{R}(\psi_o) + \mathbb{E}_{(x,y) \sim \mathcal{D}} \left[ \ell_{\psi_{rob}}^{adv}(x, \hat{y}_o) \right]. \quad (4)$$

The overhead $\mathbb{E}_{(x,y) \sim \mathcal{D}} \left[ \ell_{\psi_{\mathrm{rob}}}^{\mathrm{adv}}(x, \hat{y}_o) \right]$, named the *robustness gap*, represents the excess error due to adversarial probit corruptions. This gap is the worst-case probability that $\psi_{\mathrm{rob}}$ disagrees with the oracular aggregator under up to $f$ probits being corrupted. In the following, we analyze the robustness gap in the case when the oracular aggregator is given by the averaging operation and the robust aggregator satisfies the property of $(f, \kappa)$-robust averaging (Allouah et al., 2023).[1]

## 3 AVERAGING AS ORACULAR AGGREGATOR

We first consider the case where the oracular aggregator $\psi_{\mathrm{o}}$ is based on computing the average of the uncorrupted probits, i.e.,

$$\psi_{\mathrm{o}}(\mathbf{h}(x)) := \operatorname*{argmax}_{k \in [K]} \left[ \overline{h}(x) \right]_k, \quad \text{where} \quad \overline{h}(x) := \frac{1}{n} \sum_{i=1}^n h_i(x).$$

In this particular case, we robustify the aggregation against probit corruptions by substituting the averaging by a robust averaging $\mathrm{ROBAVG} : (\Delta^K)^n \to \mathbb{R}^K$, taking inspiration from the literature of Byzantine-robust machine learning (Guerraoui et al., 2024) and robust mean estimation (Diakonikolas & Kane, 2023). Specifically, we set

$$\psi_{\mathrm{rob}}(\mathbf{z}) := \operatorname*{argmax}_{k \in [K]} \left[ \mathrm{ROBAVG}(\mathbf{z}) \right]_k. \quad (5)$$

Where ROBAVG is a robust averaging aggregation rule. The concept of robust averaging, initially introduced by Allouah et al. (2023) can be defined as follows.

**Definition 1** (Robust averaging). *Let $\kappa \geq 0$. An aggregation rule $\mathrm{ROBAVG}$ is $(f, \kappa)$-robust if, for any set of $n$ vectors $v_1, \ldots, v_n \in \mathbb{R}^d$ and any set $S \subseteq [n]$ of size $n - f$, the following holds true:*

$$\left\| \mathrm{ROBAVG}(v_1, \ldots, v_n) - \overline{v}_S \right\|_2^2 \leq \frac{\kappa}{|S|} \sum_{i \in S} \left\| v_i - \overline{v}_S \right\|_2^2,$$

*where $\overline{v}_S := \frac{1}{n-f} \sum_{i \in S} v_i$, and parameter $\kappa$ is called the* robustness coefficient.

Essentially, robust averaging ensures that despite up to $f$ inputs being arbitrarily corrupted the output of the aggregator is an estimate of the uncorrupted vectors' average. The estimation error is bounded by the "empirical variance" of the uncorrupted input vectors, times a constant value $\kappa$.

---

[1]The analysis can be easily extended to weighted averaging by simply re-scaling the inputs to the aggregator.

Examples of robust averaging include CWTM, CWMed and geometric median (GM) (Guerraoui et al., 2024, Chapter 4). Specifically, CWTM is $(f, \kappa)$-robust with $\kappa = \frac{6f}{n-2f}\left(1 + \frac{f}{n-2f}\right)$. While robust averaging ensures proximity to the average of uncorrupted input vectors in $\ell_2$-norm, even a small estimation error can result in large prediction error due to the non-continuity of the $\operatorname{argmax}_k$ operator. We illustrate this insufficiency of robust averaging through the following counter example.

**Counter example.** Let $\varepsilon \in (0, 1/6)$. Consider a point $x \in \mathcal{X}$ and three possible probit vectors for $x$, $h_1(x), h_2(x), h_3(x) \in \Delta^3$ defined as follows:

$$h_1(x) = (1, 0, 0), \quad h_2(x) = (0, 1, 0), \quad \text{and } h_3(x) = (1/2, 1/2 - 3\varepsilon, 3\varepsilon).$$

Note that $\overline{h}(x) = (1/2, 1/2 - \varepsilon, \varepsilon)$. Now, consider $\hat{v} = (1/2 - \varepsilon, 1/2, \varepsilon)$. We have $\left\|\overline{h}(x) - \hat{v}\right\|_2 = \left(2\varepsilon^2\right)^{1/2} = 2^{1/2}\varepsilon$. Here, by taking small enough $\varepsilon$, we have that $\hat{v}$ can be arbitrarily close to $\overline{h}(x)$. However we still have $\operatorname{argmax}_{k \in [K]} \left[\overline{h}(x)\right]_k \neq \operatorname{argmax}_{k \in [K]} \left[\hat{v}\right]_k$. Hence, demonstrating proximity under the euclidean norm does not guarantee preservation of the decision made by the $\operatorname{argmax}$.

From the above, we observe that in addition to proximity to the average, the robustness gap induced under robust averaging also depends on point-wise *model dissimilarity* at $x \in \mathcal{X}$, *i.e.*,

$$\sigma_x^2 = \max_{k \in [K]} \frac{1}{n} \sum_{i=1}^n \left([h_i(x)]_k - \left[\overline{h}(x)\right]_k\right)^2.$$

We are now ready to present the robustness gap for the case when $\text{ROBAVG} = \text{CWTM}$. Description of CWTM is deferred to Appendix B.2. The reason for using CWTM is its simplicity and its proven optimality in the formal sense of robust averaging (Allouah et al., 2023). To present the result, we introduce some additional notation. For a vector $z \in \Delta^K$, let $z^{(1)}$ and $z^{(2)}$ denote the largest and the 2nd-largest values in the set $\{[z]_k\}_{k=1}^K$. Then, we define $\text{MARGIN}(z) = z^{(1)} - z^{(2)}$. If $[z]_k = [z]_{k'}$ for all $k, k' \in [K]$ (i.e., all the coordinates of the vector are equal) then $\text{MARGIN}(z) = \infty$.

**Theorem 1.** *Consider $\psi_{rob}$ as defined in (5) with $\text{ROBAVG} = \text{CWTM}$. If the regressors $h_1, \ldots, h_n$ are such that $\overline{h}(x)$ has a unique maximum coordinate almost everywhere, then the following holds:*

$$\mathbb{E}_{(x,y)\sim\mathcal{D}} \left[\ell_{\psi_{rob}}^{adv}(x, \hat{y}_o)\right] \leq \mathbb{P}_{(x,y)\sim\mathcal{D}} \left[\text{MARGIN}\left(\overline{h}(x)\right) < 2\left(\sqrt{\frac{\kappa\,n}{n-f}} + \sqrt{\frac{f}{n-f}}\right)\sigma_x\right],$$

*where $\hat{y}_o = \psi_o(h_1(x), \ldots, h_n(x))$ and $\kappa = \frac{6f}{n-2f}\left(1 + \frac{f}{n-2f}\right)$.*

The proof for this theorem is deferred to Appendix B. This shows that the robustness gap for CWTM, when the oracular aggregator is given by the averaging operation, reduces with the fraction of corruptions $f/n$, the model dissimilarity and the inverse of the average probit's margin. We validate this theoretical finding through an empirical study summarized in Figure 3 (Appendix G.1).

## 4 DEEPSET AS ORACULAR AGGREGATOR

In this section, we consider the case of more general non-linear trainable (*i.e.*, data dependent) oracular aggregators, inspired by recent work demonstrating the efficacy of such aggregation in context of ensembling (Allouah et al., 2024a; Wang et al., 2024).

**Robust empirical federated inference risk minimization (RERM).** In this case, since the oracular aggregator $\psi_o$ need not be a pre-determined linear combination of input probits (like averaging), we propose to design a robust aggregator $\psi_{rob}$ by directly aiming to minimize the robust federated inference risk $\mathcal{R}_{adv}(\psi)$. In practice, we seek to minimize the following *robust empirical federated inference risk*:

$$\widehat{\mathcal{R}}_{adv}(\psi) := \frac{1}{|\mathcal{D}_{train}|} \sum_{(x,y)\in\mathcal{D}_{train}} \ell_\psi^{adv}(x, y), \quad \text{where} \quad \ell_\psi^{adv}(x, y) := \max_{\mathbf{z}\in\Gamma_f(x)} \mathbb{1}\left\{\psi(\mathbf{z}) \neq y\right\}, \quad (6)$$

where $\mathcal{D}_{train}$ comprises of a finite number of i.i.d. data points $(x, y)$ from the global distribution $\mathcal{D}$. In short, we refer to the optimization problem (6) as RERM.

**Connection to robustness to adversarial examples.** The problem of RERM reduces to the problem of *robustness against adversarial examples* (Madry et al., 2018; Goodfellow et al., 2014), with the input space for the classifier (*i.e.*, aggregator $\psi$) being a $K \times n$ real-valued matrix with each column in $\Delta^K$ and the input perturbation being restricted to corruption of up to $f$ columns of the input matrix. Let $(\Delta^K)^n$ denote the space of such matrices, and for a matrix $M \in (\Delta^K)^n$, let $\delta(M)$ denote the set of matrices from $\mathbb{R}^{K \times n}$ such that $M + V \in (\Delta^K)^n$ for all $V \in \delta(M)$. For any matrix $V \in \mathbb{R}^{K \times n}$, we denote by $\|V\|_0$ the number of non-zero columns (*i.e.*, columns with at least one non-zero entry). We can now formally express the RERM problem in (6) as robustness to adversarial examples as follows:

$$\psi_{\text{rob}} \in \underset{\psi \in \Psi}{\operatorname{argmin}} \frac{1}{|\mathcal{D}_{\text{train}}|} \sum_{(x,y) \in \mathcal{D}_{\text{train}}} \max_{\substack{V \in \delta(H(x)) \\ \|V\|_0 \leq f}} \mathbb{1}\left\{ \psi\left(H(x) + V\right) \neq y \right\}, \tag{7}$$

where $H(x) = [h_1(x), \ldots, h_n(x)] \in (\Delta^K)^n$. By solving the above adversarial learning problem, we obtain an aggregator with minimum sensitivity against arbitrary perturbation to at most $f$ input probits while ensuring high learning accuracy at the same time. We propose to solve the RERM problem (6) and the equivalent adversarial robustness problem (7) for the space of aggregators $\Psi$ defined by a parameterized deep neural network. However, the adversarial training still remains intractable due to high permutational cardinality of the input space of $H(x)$, totaling to the factor $\sum_{m=1}^f {}^nP_m$ since up to any $f$ columns can be perturbed (Liu et al., 2022) where ${}^nP_m = \frac{n!}{(n-m)!}$. To alleviate this issue, we leverage the property that the output of the aggregator must be invariant to the order of columns in $H(x)$. We thus exploit a specific neural network architecture which is permutation invariant to its inputs, as described below.

**Robust DeepSet aggregator.** Consider $\theta_1 \in \mathbb{R}^{d_1}$ and $\theta_2 \in \mathbb{R}^{d_2}$, and two mappings parameterized by these vectors: $\rho_{\theta_1} : \Delta^K \to \mathbb{R}^p$ and $\mu_{\theta_2} : \mathbb{R}^p \to \Delta^K$. Then, we define $\Psi$ by a set of parameterized mappings $\phi_\theta : (\Delta^K)^n \to \Delta^K$ composed with the $\operatorname{argmax}$ operation, where $\theta = (\theta_1, \theta_2)$ and

$$\phi_\theta(\mathbf{z}) \coloneqq \mu_{\theta_2}\left( \frac{1}{n} \sum_{i=1}^n \rho_{\theta_1}(z_i) \right). \tag{8}$$

This particular type of neural network is commonly known as DeepSet (Zaheer et al., 2017), in the case when there are no corruptions *i.e.*, $z_i = h_i(x), \forall i \in [n]$. Consequently, in this case, the RERM problem reduces to the following optimization problem:

$$\theta^* \in \underset{\theta \in \Theta}{\operatorname{argmin}} \frac{1}{|\mathcal{D}_{\text{train}}|} \sum_{(x,y) \in \mathcal{D}_{\text{train}}} \max_{\mathbf{z} \in \Gamma_f(x)} \mathbb{1}\left\{ \underset{k \in [K]}{\operatorname{argmax}} \left[ \phi_\theta(z_1, \ldots, z_n) \right]_k \neq y \right\}. \tag{9}$$

Solving (9) is still complex in practice due to the non-continuity of the indicator function. We address this by changing the point-wise loss function from the indicator function to the cross-entropy loss function, denoted as $\ell(\phi_\theta(\mathbf{z}), y)$. Finally, we still need to compute the maximum value of the point-wise loss at $(x, y)$ over all $\mathbf{z} \in \Gamma_f(x)$. We address this challenge by approximating the maximum value of $\ell(\phi_\theta(\mathbf{z}), y)$ by searching over only a subset of $\mathbf{z} \in \Gamma_f(x)$. Thanks to the permutation invariance property of DeepSet, the permutational cardinality of searching $f$ adversaries now reduces to $\sum_{m=1}^f \binom{n}{m}$ where $\binom{n}{m} = \frac{n!}{(n-m)!m!}$. Specifically, in each training iteration, we randomly select $m \leq f$ clients from the set $[n]$ to perturb their probits, repeated $N$ times. We then obtain the perturbed probits and approximate maximum loss by using a multi-step variant of the *fast gradient sign method* (FGSM) for *adversarial training* (Madry et al., 2018). The network parameters are then updated to correctly classify despite the perturbations. In practice, we show that $N \ll \binom{n}{f}$ suffices to obtain a good approximation. Our resulting algorithm is summarized in Algorithm 1.

Finally, we incorporate robust averaging to further reduce the overall sensitivity of $\phi_{\theta^*}$ to probit corruptions. Specifically, with $\theta^* = (\theta_1^*, \theta_2^*)$, the *robust DeepSet aggregator* is defined as follows:

$$\psi_{\text{rob}}(\mathbf{z}) \coloneqq \underset{k \in [K]}{\operatorname{argmax}} \left[ \mu_{\theta_2^*}\left( \textsc{RobAvg}(\rho_{\theta_1^*}(z_1), \ldots, \rho_{\theta_1^*}(z_n)) \right) \right]_k. \tag{10}$$

Note that we only incorporate robust averaging at the inference time since incorporating at training time renders adversarial training more expensive due to the additional cost of RobAvg.

---

**Algorithm 1:** Training of DeepSet Aggregator $\phi_\theta$

---

1    **Input:** Training Steps $E$, sampling count $N$, learning rates $\gamma$ and $\eta$, adversarial sample
         generation steps $S$, dataset $\mathcal{D}_{\text{train}}$, number of adversaries $f$ and training loss function $\ell$

2    **begin**

3      **for** 1 **to** $E$ **do**

4          Sample $x, y \sim \mathcal{D}_{\text{train}}$ [1]

5          **for** 1 **to** $N$ **do**

6             Choose $m \in \{1, \ldots, f\}$ with probability proportional to $\binom{n}{m}$

7             Randomly initialize $\boldsymbol{v} = (v_1, \ldots, v_m) \in (\mathbb{R}^K)^m$

8             Sample a random permutation $\pi$ of $\{1, \ldots, n\}$         ▷ *For selecting adversaries*

9             **for** 1 **to** $S$ **do**

10                $\hat{v}_i \leftarrow \text{softmax}(v_i), \forall i \in [m]$       ▷ *Enforce each $v_i$ to be in $\Delta^K$*

11                $\boldsymbol{z} = (h_{\pi(1)}(x), \ldots, h_{\pi(n-m)}(x), \hat{v}_1, \ldots, \hat{v}_m)$     ▷ *Last $m$ are adversarial*

12                $\boldsymbol{v} \leftarrow \boldsymbol{v} + \gamma \, \text{sgn}\left(\nabla_{\boldsymbol{v}} \ell\left(\phi_\theta(\boldsymbol{z}), y\right)\right)$     ▷ *Perturb probits using FGSM*

13             $\hat{v}_i \leftarrow \text{softmax}(v_i), \forall i \in [m]$        ▷ *Final projection of $v_i$ to $\Delta^K$*

14             $\boldsymbol{z} = (h_{\pi(1)}(x), \ldots, h_{\pi(n-m)}(x), \hat{v}_1, \ldots, \hat{v}_m)$

15             $\theta \leftarrow \theta - \eta \nabla_\theta \ell\left(\phi_\theta(\boldsymbol{z}), y\right)$        ▷ *Approximated maximum loss*

16      **return** Trained aggregator $\phi_\theta$

[1] In practice we sample mini-batches.

---

**Robustness gap of robust DeepSet aggregator.** In the theorem below, we present an upper bound on the robustness gap of the robust DeepSet aggregator (10), extending Theorem 1 to non-linear oracular aggregation. In what follows, we make some Lipschitz continuity assumptions. Specifically, there exists real values $L_\rho, L_\mu$ such that, for all $z, z' \in \Delta^K$, $v, v' \in \mathbb{R}^p$ and $k \in [K]$,

$$\left|[\rho_{\theta_1^*}(z)]_k - [\rho_{\theta_1^*}(z')]_k\right| \leq L_\rho \left\|z - z'\right\|_2 \ , \quad \text{and}$$

$$\left|[\mu_{\theta_2^*}(v)]_k - [\mu_{\theta_1^*}(v')]_k\right| \leq L_\mu \left\|v - v'\right\|_2 .$$

Let $\mathbf{h}(x) := (h_1(x), \ldots, h_n(x))$. Recall that $\sigma_x^2 := \max_{k \in [K]} \frac{1}{n} \sum_{i=1}^n \left([h_i(x)]_k - [\overline{h}(x)]_k\right)^2$.

**Theorem 2.** *Consider $\psi_{rob}$ as defined in (10) with* ROBAVG $=$ CWTM. *If the regressors $h_1, \ldots, h_n$ are such that $\phi_{\theta^*}(\mathbf{h}(x))$ has a unique maximum coordinate almost everywhere, then the following holds:*

$$\mathbb{E}_{(x,y) \sim \mathcal{D}}\left[\ell_{\psi_{rob}}^{adv}(x, \hat{y}_o)\right]$$

$$\leq \mathbb{P}_{(x,y) \sim \mathcal{D}}\left[\text{MARGIN}\left(\phi_{\theta^*}(\mathbf{h}(x))\right) < 2\sqrt{2} L_\mu L_\rho \left(\sqrt{\frac{\kappa n}{n-f}} + \sqrt{\frac{f}{n-f}}\right) \min\left\{1, \sqrt{K}\sigma_x\right\}\right],$$

*where $\hat{y}_o = \text{argmax}_{k \in [K]} [\phi_{\theta^*}(\mathbf{h}(x))]_k$ and $\kappa = \frac{6f}{n-2f}\left(1 + \frac{f}{n-2f}\right)$.*

The proof for this theorem is deferred to Appendix C, adapting Theorem 1 to the case of a non-linear aggregation. The theorem shows that the robustness gap of the robust DeepSet aggregator is controlled by the same three quantities as in the linear (averaging) case: the corruption fraction $f/n$, the model dissimilarity $\sigma_x$, and the inverse margin of the aggregated probit $\phi_{\theta^*}(\mathbf{h}(x))$. The only difference is a multiplicative sensitivity term $L_\mu L_\rho$, which quantifies how much the non-linear aggregator amplifies perturbations in the inputs. Appendix D additionally includes a formal reasoning on the benefits of using ROBAVG within DeepSet, proving that it removes the dependence on the degree of probits' corruption compared to using the original DeepSet (with simple averaging).

## 5   EXPERIMENTS

This section summarizes the key results of our experiments. In all that follows, we consider the ROBAVG in the robust DeepSet aggregator (10) to be CWTM, and we refer to the corresponding

model as DeepSet-TM. We evaluate DeepSet-TM against robust aggregators like CWMed, against SOTA baselines, and perform a scalability study in Section 5.2. We conduct an ablation study in Appendix G.4, and a comparison with more adversarial defenses in Appendix G.5.

## 5.1 EXPERIMENTAL SETUP

**Datasets.** We evaluate on CIFAR-10, CIFAR-100 (Krizhevsky, 2012), and AG-News (Zhang et al., 2015), covering vision and language tasks. We reserve $10\%$ of training data as server-side validation data for aggregator training, and partition the rest across clients using the Dirichlet distribution $\texttt{Dir}_n(\alpha)$, in line with previous works (Roux et al., 2025; Dai et al., 2024). Lower $\alpha$ indicates higher heterogeneity. We experiment with $\alpha = \{0.3, 0.5, 0.8\}$. In most of our evaluations, we consider $n = 17$ following prior work (Allouah et al., 2023), except for our scalability study where we vary $n = \{10, 17, 25\}$. Our choice of number of clients is in line with a typical cross-silo federated setting in real-world (Ogier du Terrail et al., 2022).

**Models.** For CIFAR-10, clients train a ResNet-8 from scratch (He et al., 2016), while for CIFAR-100 and AG-News they fine-tune a ViT-B/32 (Dosovitskiy et al., 2021; Radford et al., 2021) and a DistilBERT (Sanh et al., 2019), respectively. The DeepSet functions $\mu$ and $\rho$ are each two-layer MLPs with a ReLU non-linearity. The AutoEncoder in COPUR consists of two-layer MLP encoders and decoders with a leaky ReLU, while the server model is a three-layer MLP (Liu et al., 2022).

**Attacks.** We evaluate on a total of 6 attacks with varying difficulties depending upon the power of adversary. We consider 4 white-box and 2 black-box attacks where the former assumes access to the server's aggregation model. We propose a new attack called SIA, in the white-box as well as the black-box setting which tries to flip the aggregation decision by exploiting the second most probable class. The remaining attacks constitute the Logit Flipping Attack, Loss Maximization Attack (LMA), Class Prior Attack (CPA) and the Projected Gradient Descent (PGD) attack presented in prior work (Roux et al., 2025; Liu et al., 2022). All attacks and their characteristics are detailed in Appendix F.2, and summarized in Table 4 within. We vary $f \in \{3, 4, 5\}$ across different setups.

**Baselines.** We compare the performance of DeepSet against several robust aggregators including CWTM (Yin et al., 2018), GM (Pillutla et al., 2022; Small, 1990) and CWMed (Yin et al., 2018) alongside simple averaging. As the data-dependent baselines, we consider COPUR and manifold projection from Liu et al. (2022) where the latter simply projects the input probits onto a learned manifold using an AutoEncoder. Additionally, we also report the performance of the non-adversarially trained DeepSet model in our study of robust elements (Appendix G.4). We include considerations on adversarial training cost and inference latency for the static aggregations and the two DeepSet variants in Appendix G.8.

**Reproducibility and reusability.** We conduct each experiment with five random seeds, and report the mean and the standard deviation. Our code is also available for reproducibility and reusability[2]. We include additional details on the experimental setup and hyperparameters in Appendix F. All clean accuracy values (*i.e.*, for $f = 0$) are included in Tables 12 and 13 (Appendix G.7).

## 5.2 RESULTS

**Performance comparison to Robust Aggregators.** Across all datasets in Table 2, DeepSet-TM achieves substantially higher worst-case accuracy (minimum accuracy across attacks) than the robust aggregation baselines. Specifically, it improves over the strongest baseline by +4.7 to +22.2 percentage points depending on the dataset, demonstrating robustness even under the most challenging adversarial conditions. Beyond worst-case performance, DeepSet-TM also shows consistent gains across attacks: in 14 out of 18 dataset–attack combinations, it achieves the highest accuracy, with only small drops ($\leq 2.3\%$ points) in the remaining four cases. The advantage of DeepSet-TM is particularly pronounced against stronger attacks (specifically SIA whitebox and PGD-cw). On CIFAR-10 and CIFAR-100, baseline accuracy drops by 35-40 points, while our approach limits the drop to 20-30. On AG-News, DeepSet-TM retains 83.2% accuracy under PGD-cw compared to 53-55% for the baselines. Notably, as we are working in probit space, giving more power to the adversary (*i.e.*, conducting more PGD iterations during testing than for training) does not lead to performance degradation (see Table 11 in appendix).

---

[2]https://github.com/sacs-epfl/robust-federated-inference

Table 2: Accuracy (%) of DeepSet-TM against static aggregations on the CIFAR-10, CIFAR-100 and AG-News datasets, with heterogeneity $\alpha = 0.5$, $n = 17$ clients and $f = 4$ adversaries. Logit flipping uses an amplification factor of 2. Results with $\alpha = \{0.3, 0.8\}$ are included in Tables 6 and 7.

| | Aggregation | Logit flipping | SIA-bb | LMA | CPA | SIA | PGD-cw | Worst case |
|---|---|---|---|---|---|---|---|---|
| **CF-10** | Mean | $65.4 \pm 2.1$ | $59.8 \pm 1.1$ | $58.7 \pm 3.2$ | $55.6 \pm 4.0$ | $42.7 \pm 3.9$ | $24.6 \pm 4.9$ | $24.6 \pm 4.9$ |
| | CWMed | $56.7 \pm 4.3$ | $53.3 \pm 2.0$ | $53.8 \pm 2.5$ | $52.3 \pm 2.9$ | $49.3 \pm 3.2$ | $27.8 \pm 4.7$ | $27.8 \pm 4.7$ |
| | GM | $63.9 \pm 2.3$ | $59.1 \pm 1.4$ | $\mathbf{63.3 \pm 2.3}$ | $\mathbf{59.7 \pm 3.2}$ | $45.3 \pm 3.7$ | $25.4 \pm 4.8$ | $25.4 \pm 4.8$ |
| | CWTM | $63.3 \pm 2.7$ | $59.4 \pm 1.3$ | $62.5 \pm 2.7$ | $\mathbf{59.7 \pm 3.2}$ | $44.8 \pm 3.7$ | $27.2 \pm 5.1$ | $27.2 \pm 5.1$ |
| | DeepSet-TM | $\mathbf{67.6 \pm 0.8}$ | $\mathbf{62.6 \pm 1.6}$ | $61.0 \pm 4.4$ | $59.4 \pm 4.7$ | $\mathbf{51.4 \pm 2.2}$ | $\mathbf{48.2 \pm 4.2}$ | $\mathbf{48.2 \pm 4.2}$ |
| **CF-100** | Mean | $\mathbf{78.8 \pm 0.7}$ | $72.8 \pm 1.0$ | $66.6 \pm 0.3$ | $66.4 \pm 0.3$ | $56.0 \pm 1.0$ | $39.2 \pm 1.2$ | $39.3 \pm 1.3$ |
| | CWMed | $65.8 \pm 1.3$ | $62.3 \pm 1.1$ | $66.1 \pm 1.2$ | $66.1 \pm 1.2$ | $62.9 \pm 1.1$ | $41.7 \pm 1.3$ | $41.7 \pm 1.3$ |
| | GM | $75.4 \pm 0.9$ | $71.5 \pm 1.3$ | $75.4 \pm 0.8$ | $75.3 \pm 0.8$ | $59.6 \pm 1.0$ | $39.0 \pm 1.3$ | $39.0 \pm 1.3$ |
| | CWTM | $74.8 \pm 1.2$ | $71.5 \pm 1.3$ | $74.9 \pm 1.1$ | $74.8 \pm 1.0$ | $60.8 \pm 0.9$ | $44.9 \pm 1.3$ | $44.9 \pm 1.3$ |
| | DeepSet-TM | $78.0 \pm 0.3$ | $\mathbf{74.7 \pm 0.8}$ | $\mathbf{76.0 \pm 0.2}$ | $\mathbf{76.4 \pm 0.5}$ | $\mathbf{63.7 \pm 0.5}$ | $\mathbf{49.6 \pm 0.5}$ | $\mathbf{49.6 \pm 0.5}$ |
| **AG-News** | Mean | $84.5 \pm 0.9$ | $81.4 \pm 2.2$ | $\mathbf{81.2 \pm 2.2}$ | $76.4 \pm 4.0$ | $72.6 \pm 4.6$ | $54.9 \pm 6.7$ | $54.9 \pm 6.7$ |
| | CWMed | $78.9 \pm 3.0$ | $78.4 \pm 3.7$ | $75.9 \pm 5.0$ | $74.1 \pm 4.5$ | $74.4 \pm 4.4$ | $53.2 \pm 7.0$ | $53.2 \pm 7.0$ |
| | GM | $83.0 \pm 0.7$ | $80.7 \pm 2.8$ | $80.9 \pm 2.5$ | $76.6 \pm 3.6$ | $74.0 \pm 3.8$ | $52.6 \pm 7.3$ | $52.6 \pm 7.3$ |
| | CWTM | $84.3 \pm 1.0$ | $81.4 \pm 2.2$ | $80.2 \pm 2.7$ | $76.4 \pm 4.0$ | $72.6 \pm 4.6$ | $55.3 \pm 7.5$ | $55.3 \pm 7.5$ |
| | DeepSet-TM | $\mathbf{85.7 \pm 0.4}$ | $\mathbf{81.6 \pm 1.6}$ | $79.2 \pm 1.3$ | $\mathbf{77.5 \pm 1.2}$ | $\mathbf{80.1 \pm 1.6}$ | $\mathbf{83.2 \pm 1.4}$ | $\mathbf{77.5 \pm 1.2}$ |

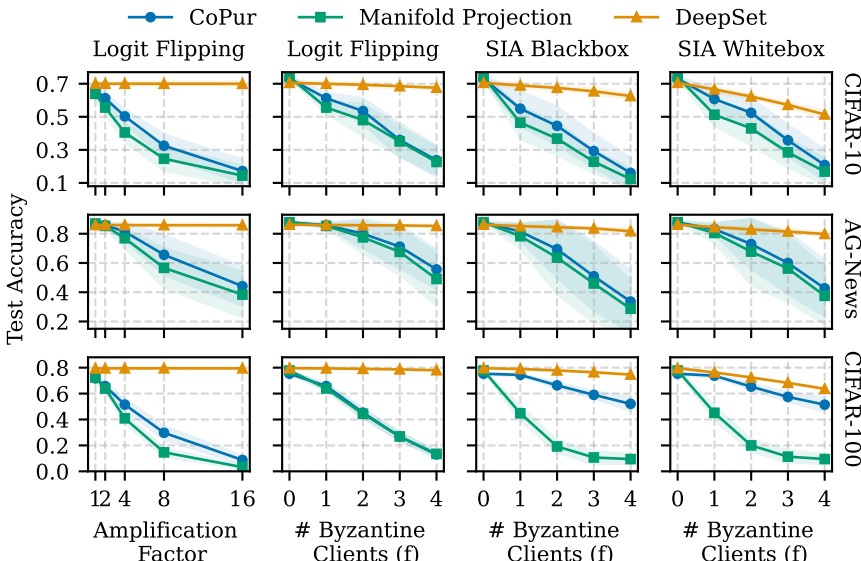

Figure 2: DeepSet-TM vs. baselines under $\alpha = 0.5$ and $n = 17$ clients. In column 1, we have $f = 1$ adversary; other columns use amplification factor 2. See Figure 4 (App. G.6) for results with $\alpha = 0.3$.

**Performance comparison with SOTA baselines.** We compare DeepSet-TM with COPUR and manifold projection in Figure 2. Since COPUR operates in logit rather than probit space, it is highly sensitive to input magnitudes, leading to sharp performance degradation under even mild attacks. For instance, in the logit-flipping attack, increasing amplification from 1 to 16 or raising the number of adversarial clients from 1 to 4 reduces COPUR's accuracy on CIFAR-10 nearly to random chance, while DeepSet-TM remains stable due to its probit-space design and adversarial training. A similar trend holds under the SIA attack, where COPUR suffers large drops on CIFAR-10 and AG-News, though it shows relative robustness on CIFAR-100. This is likely because COPUR leverages block-sparse structure of input probits where higher number of classes induce more sparsity, increasing its effectiveness. Manifold projection performs even worse, as its autoencoder struggles with adversarial patterns spread across many classes. In contrast, DeepSet-TM consistently achieves substantially higher accuracy across all attacks, with only a minor accuracy loss in the no-adversary case ($f = 0$), a known trade-off from adversarial training (Madry et al., 2018).

Table 3: Scalability study on the CIFAR-100 dataset under $\alpha = 0.3$.

| $n$ | $f$ | Aggregation | Logit flipping | SIA-bb | LMA | CPA | SIA-wb | PGD-cw | Worst case |
|---|---|---|---|---|---|---|---|---|---|
| 10 | 3 | Mean | **75.1 ± 1.3** | 63.2 ± 2.5 | 42.5 ± 2.9 | 42.2 ± 2.8 | 44.3 ± 1.7 | 19.7 ± 2.2 | 19.7 ± 2.2 |
| | | CWMed | 45.6 ± 2.7 | 44.5 ± 2.2 | 52.4 ± 1.6 | 50.9 ± 1.8 | 50.5 ± 2.0 | 16.5 ± 2.3 | 16.5 ± 2.3 |
| | | GM | 69.6 ± 1.3 | 61.3 ± 1.6 | **62.6 ± 2.2** | **62.2 ± 2.1** | 47.3 ± 1.5 | 18.3 ± 2.6 | 18.3 ± 2.6 |
| | | CWTM | 69.5 ± 1.3 | 62.4 ± 2.1 | 45.2 ± 2.9 | 44.9 ± 2.8 | 46.4 ± 1.6 | 24.0 ± 2.0 | 24.0 ± 2.0 |
| | | DeepSet-TM | 73.5 ± 1.3 | **66.8 ± 2.0** | 56.3 ± 2.0 | 60.0 ± 1.0 | **56.5 ± 1.8** | **33.2 ± 2.5** | 33.2 ± 2.5 |
| 17 | 4 | Mean | 75.6 ± 0.7 | 69.1 ± 0.8 | 57.0 ± 2.4 | 56.8 ± 2.4 | 44.5 ± 1.4 | 25.8 ± 2.8 | 25.8 ± 2.8 |
| | | CWMed | 55.1 ± 2.1 | 51.8 ± 2.6 | 55.9 ± 2.2 | 55.8 ± 2.2 | 50.7 ± 1.8 | 24.5 ± 1.9 | 24.5 ± 1.9 |
| | | GM | 71.2 ± 0.8 | 66.3 ± 1.1 | 70.2 ± 0.8 | 70.0 ± 0.9 | 48.3 ± 1.5 | 23.8 ± 2.6 | 23.8 ± 2.6 |
| | | CWTM | 69.1 ± 1.1 | 65.8 ± 1.2 | 69.1 ± 1.2 | 69.1 ± 1.2 | 49.6 ± 1.1 | 32.5 ± 2.4 | 32.5 ± 2.4 |
| | | DeepSet-TM | **75.7 ± 0.8** | **71.9 ± 1.1** | **72.2 ± 1.9** | **72.5 ± 0.6** | **56.7 ± 1.0** | **38.6 ± 2.5** | 38.6 ± 2.5 |
| 25 | 5 | Mean | **76.0 ± 1.1** | 71.1 ± 1.3 | 59.7 ± 2.3 | 59.4 ± 2.3 | 40.1 ± 2.3 | 24.1 ± 2.0 | 24.1 ± 2.0 |
| | | CWMed | 51.7 ± 2.9 | 49.2 ± 2.0 | 52.1 ± 2.3 | 52.0 ± 2.3 | 44.9 ± 1.8 | 20.6 ± 2.9 | 20.6 ± 2.9 |
| | | GM | 70.8 ± 1.4 | 67.3 ± 1.1 | **70.7 ± 1.5** | **70.5 ± 1.5** | 44.2 ± 1.9 | 21.3 ± 2.0 | 21.3 ± 2.0 |
| | | CWTM | 70.1 ± 1.7 | 67.7 ± 1.4 | 70.1 ± 1.7 | 70.1 ± 1.7 | 45.8 ± 2.0 | 30.4 ± 2.2 | 30.4 ± 2.2 |
| | | DeepSet-TM | 72.8 ± 1.1 | **72.8 ± 1.2** | 65.7 ± 2.3 | 69.6 ± 1.6 | **53.8 ± 1.5** | **50.7 ± 1.5** | 50.7 ± 1.5 |

**Scalability study.** To assess the generalizability of our approach, we conduct performance evaluations with varying the number of clients: $(n, f) \in \{(10, 3), (17, 4), (25, 5)\}$ on CIFAR-100 under high heterogeneity ($\alpha = 0.3$). In Table 3, we observe that DeepSet-TM consistently provides the highest robustness as the system scales. It improves worst-case accuracy over the best baseline by 9.2%, 6.1% and 20.3% points respectively when increasing $n$ and $f$. Beyond worst-case performance, DeepSet-TM notably achieves the highest performance in 12 out of 18 dataset–attack combinations. While static aggregations occasionally remain competitive on milder attacks (*e.g.*, LMA), DeepSet-TM is never substantially worse and typically remains close to the highest accuracy across most scenarios.

## 6 CONCLUSION

We formalized the problem of robust federated inference and derived a certification for robust averaging under adversarial corruptions. For non-linear aggregation, we proposed DeepSet-TM, a permutation-invariant neural network trained adversarially and combined at test-time with robust averaging. Our experiments demonstrate that DeepSet-TM consistently improves accuracy across datasets and attack types, substantially outperforming prior methods.

### ACKNOWLEDGMENTS

Rafael is partially supported by the French National Research Agency and the French Ministry of Research and Higher Education. The joint work of Nirupam and Rafael is also supported in part by the CNRS through the International Emerging Action (IEA) program and the French National Research Agency through the ANR TuLIP. This work has also been partly supported by the Swiss National Science Foundation, under the project "FRIDAY: Frugal, Privacy-Aware and Practical Decentralized Learning", SNSF proposal No. 10.001.796.

### REPRODUCIBILITY STATEMENT

We provide all the necessary details to reproduce our experiments in Section 5.1 and in Appendix F. Furthermore, we release our complete codebase at `https://github.com/sacs-epfl/robust-federated-inference`.

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

## A   PROOF OF LEMMA 1

We restate Lemma 1 for convenience and then prove it below.

**Lemma 1.** *For any $x \in \mathcal{X}$, let $\hat{y}_o = \psi_o(\mathbf{h}(x))$. Then,*

$$\mathcal{R}_{adv}(\psi_{rob}) \leq \mathcal{R}(\psi_o) + \mathbb{E}_{(x,y) \sim \mathcal{D}} \left[ \ell^{adv}_{\psi_{rob}}(x, \hat{y}_o) \right]. \tag{4}$$

*Proof.* We can decompose the robust ensemble risk (3) for an aggregator $\psi_{\text{rob}}$ as follows:

$$\mathcal{R}_{\text{adv}}(\psi_{\text{rob}}) = \mathcal{R}(\psi_o) + \mathbb{E}_{(x,y) \sim \mathcal{D}} \left[ \text{Gap}(x, y \mid \psi_o, \psi_{\text{rob}}) \right], \tag{11}$$

where $\text{Gap}(x, y \mid \psi_o, \psi_{\text{rob}}) := \ell^{\text{adv}}_{\psi_{\text{rob}}}(x, y) - \ell_{\psi_o}(x, y)$. Recall that we denote $\mathbf{h}(x) = (h_1(x), \ldots, h_n(x))$. Note that for any $x$ and any $\mathbf{z} = (z_1, \ldots, z_n) \in (\Delta^K)^n$, we have

$$\mathbb{1}\left\{\psi_{\text{rob}}(\mathbf{z}) \neq y\right\} - \mathbb{1}\left\{\psi_o(\mathbf{h}(x)) \neq y\right\} = \begin{cases} 1 & \text{if } \psi_o(\mathbf{h}(x)) = y \text{ and } \psi_{\text{rob}}(\mathbf{z}) \neq y, \\ -1 & \text{if } \psi_o(\mathbf{h}(x)) \neq y \text{ and } \psi_{\text{rob}}(\mathbf{z}) = y, \\ 0 & \text{otherwise.} \end{cases}$$

This implies that, for any $x$ and any $(z_1, \ldots, z_n) \in (\Delta^K)^n$,

$$\mathbb{1}\left\{\psi_{\text{rob}}(\mathbf{z}) \neq y\right\} - \mathbb{1}\left\{\psi_o(\mathbf{h}(x)) \neq y\right\} \leq \mathbb{1}\left\{\psi_{\text{rob}}(\mathbf{z}) \neq \psi_o(\mathbf{h}(x))\right\}.$$

Therefore, for any $x$,

$$\max_{\mathbf{z} \in \Gamma_f(x)} \mathbb{1}\left\{\psi_{\text{rob}}(\mathbf{z}) \neq y\right\} - \mathbb{1}\left\{\psi_o(\mathbf{h}(x)) \neq y\right\} \leq \max_{\mathbf{z} \in \Gamma_f(x)} \mathbb{1}\left\{\psi_{\text{rob}}(\mathbf{z}) \neq \psi_o(\mathbf{h}(x))\right\}.$$

Substituting from above in (11) concludes the proof. □

## B   PROOF OF THEOREM 1

We decompose the proof of Theorem 1 into three steps for clarity purposes.

### B.1   BOUNDING THE VARIANCE OF A SUBSET

We first state and prove a lemma that will guide us towards the proof of the main theorem.

**Lemma 2.** *Let $v_1, v_2, \ldots, v_n \in \mathbb{R}^d$ be a set of vectors. We denote by $\bar{v}_n = \frac{1}{n} \sum_{i=1}^{n} v_i$ the mean of these points. Let $S \subset [n]$ such that $|S| = n - f$, we denote by $\bar{v}_S = \frac{1}{n-f} \sum_{i \in S} v_i$ the average of the vectors in $S$. Then the following holds true:*

$$\frac{1}{n-f} \sum_{i \in S} \|v_i - \bar{v}_S\|^2 \leq \frac{n}{n-f} \left( \frac{1}{n} \sum_{i=1}^{n} \|v_i - \bar{v}_n\|^2 \right),$$

*Proof.* Let $S \subset [n]$ such that $|S| = n - f$, and $T = [n] \backslash S$. We have

$$\sum_{i \in S} \|v_i - \bar{v}_n\|^2 = \sum_{i \in S} \|v_i - \bar{v}_S\|^2 + \sum_{i \in S} \|\bar{v}_S - \bar{v}_n\|^2 + 2 \sum_{i \in S} \langle v_i - \bar{v}_S, \bar{v}_S - \bar{v}_n \rangle$$

As, $\sum_{i \in S} \langle v_i - \bar{v}_S, \bar{v}_S - \bar{v}_n \rangle = 0$, we get

$$\sum_{i \in S} \|v_i - \bar{v}_n\|^2 = (n - f)\zeta_S^2 + (n - f)\|\bar{v}_S - \bar{v}_n\|^2$$

With $\zeta_S^2 = \frac{1}{n-f} \sum_{i \in S} \|v_i - \bar{v}_S\|^2$. Similarly, we can show that

$$\sum_{i \in T} \|v_i - \bar{v}_n\|^2 = f\,\zeta_T^2 + f\|\bar{v}_T - \bar{v}_n\|^2.$$

From the above, we can simply rewrite

$$\sum_{i \in [n]} \|v_i - \bar{v}_n\|^2 = \sum_{i \in S} \|v_i - \bar{v}_n\|^2 + \sum_{i \in T} \|v_i - \bar{v}_n\|^2$$

$$= (n - f)\,\zeta_S^2 + (n - f)\|\bar{v}_S - \bar{v}_n\|^2 + f\,\zeta_T^2 + f\|\bar{v}_T - \bar{v}_n\|^2$$

Now we note that $\bar{v}_n = \frac{f\bar{v}_T + (n-f)\bar{v}_S}{n}$. Hence

$$\bar{v}_S - \bar{v}_n = \frac{f(\bar{v}_S - \bar{v}_T)}{n} \qquad \text{and} \qquad \bar{v}_T - \bar{v}_n = \frac{(n-f)(\bar{v}_T - \bar{v}_S)}{n.}$$

Substituting this in the above we get

$$\sum_{i \in [n]} \|v_i - \bar{v}_n\|^2 = (n - f)\,\zeta_S^2 + \frac{(n-f)f^2}{n^2}\|\bar{v}_S - \bar{v}_T\|^2 + f\,\zeta_T^2 + \frac{(n-f)^2 f}{n^2}\|\bar{v}_T - \bar{v}_S\|^2$$

$$= (n - f)\,\zeta_S^2 + f\,\zeta_T^2 + \frac{(n-f)f}{n}\|\bar{v}_T - \bar{v}_S\|^2$$

Simplifying the above and reorganizing the terms, we get

$$\frac{n}{n-f}\left(\frac{1}{n}\sum_{i \in [n]} \|v_i - \bar{v}_n\|^2\right) = \frac{1}{n-f}\sum_{i \in S} \|v_i - \bar{v}_S\|^2 + \frac{f}{n-f}\,\zeta_T^2 + \frac{f}{n}\|\bar{v}_T - \bar{v}_S\|^2.$$

As all the terms in the right-hand side are non-negative, the above concludes the proof. □

## B.2 DEFINITION AND PROPERTIES OF CWTM

As its name indicates, coordinate-wise trimmed mean, is based on a scalar protocol called trimmed mean. Given $n$ scalar values $v_1, \ldots, v_n \in \mathbb{R}$, trimmed mean denoted by TM, is defined to be

$$\text{TM}(v_1, \ldots, v_n) = \frac{1}{n-2f}\sum_{i=f+1}^{n-f} v_{(i)}, \tag{12}$$

where $v_{(i)}, \cdots, v_{(n)}$ denote the order statistics of $v_1, \cdots, v_n$, i.e., the sorting of the values in non-decreasing order (with ties broken arbitrarily)[3]. Using this primitive, we can define the coordinate-wise trimmed mean aggregation as follows. Given the input vectors $v_1, \ldots, v_n \in \mathbb{R}^d$, the coordinate-wise trimmed mean of $v_1, \ldots, v_n$, denoted by $\text{CWTM}(v_1, \ldots, v_n)$, is a vector in $\mathbb{R}^d$ whose $k$-th coordinate is defined as follows,

$$[\text{CWTM}(v_1, \ldots, v_n)]_k = \text{TM}([v_1]_k, \ldots, [v_n]_k).$$

**Definition 2** $((f, \kappa)$-**robustness**). *Let $f < \frac{n}{2}$ and $\kappa \geq 0$. An aggregation rule ROBAVG is said to be $(f, \kappa)$-robust if for any vectors $v_1, \ldots, v_n \in \mathbb{R}^d$, and any set $S \subseteq [n]$ of size $n - f$,*

$$\|\text{ROBAVG}(v_1, \ldots, v_n) - \overline{v}_S\|^2 \leq \frac{\kappa}{|S|}\sum_{i \in S} \|v_i - \overline{v}_S\|^2$$

*where $\overline{v}_S = \frac{1}{|S|}\sum_{i \in S} v_i$. We refer to $\kappa$ as the* robustness coefficient.

We will use the following lemma as part of the theorem proof. We refer to Allouah et al. (2023) for its proof.

**Lemma 3** (Allouah et al. (2023)). *Let $n \in \mathbb{N}^*$ and $f < \frac{n}{2}$. Then TM is $(f, \kappa)$-robust with $\kappa = \frac{6f}{n-2f}\left(1 + \frac{f}{n-2f}\right)$.*

---

[3] More formally, let $\tau$ be the permutation on $[n]$ such that $v_{\tau(1)} \leq \cdots \leq v_{\tau(n)}$. The $i$-th order statistic of $v_1, \cdots, v_n$ is simply $v_{(i)} = v_{\tau(i)}$ for all $i \in [n]$.

### B.3 PROOF OF THE THEOREM

Using the two lemmas presented above, we can now prove the main theorem, which we restate below for the reader's convenience.

**Theorem 1.** *Consider $\psi_{rob}$ as defined in (5) with* ROBAVG = CWTM. *If the regressors $h_1, \ldots, h_n$ are such that $\overline{h}(x)$ has a unique maximum coordinate almost everywhere, then the following holds:*

$$\mathbb{E}_{(x,y)\sim\mathcal{D}}\left[\ell_{\psi_{rob}}^{adv}(x,\hat{y}_o)\right] \leq \mathbb{P}_{(x,y)\sim\mathcal{D}}\left[\text{MARGIN}\left(\overline{h}(x)\right) < 2\left(\sqrt{\frac{\kappa\,n}{n-f}} + \sqrt{\frac{f}{n-f}}\right)\sigma_x\right],$$

*where $\hat{y}_o = \psi_o(h_1(x), \ldots, h_n(x))$ and $\kappa = \frac{6f}{n-2f}\left(1 + \frac{f}{n-2f}\right)$.*

*Proof.* Let us first recall that by definition

$$\ell_{\psi_{\text{rob}}}^{\text{adv}}(x,\hat{y}_o) = \begin{cases} 0 & \text{if } \psi_{\text{rob}}(\mathbf{z}) = \hat{y}_o, \forall \mathbf{z} \in \Gamma_f(x), \\ 1 & \text{otherwise.} \end{cases}$$

In particular, this means that

$$\mathbb{E}_{(x,y)\sim\mathcal{D}}\left[\ell_{\psi_{\text{rob}}}^{\text{adv}}(x,\hat{y}_o)\right] = 1 - \mathbb{P}_{(x,y)\sim\mathcal{D}}\left[\psi_{\text{rob}}(\mathbf{z}) = \hat{y}_o, \forall \mathbf{z} \in \Gamma_f(x)\right]. \tag{13}$$

Now recall that, for any $x \in \mathcal{X}$, by definition

$$\hat{y}_o = \psi_o(\mathbf{h}(x)) := \underset{k\in[K]}{\operatorname{argmax}}\left[\overline{h}(x)\right]_k,$$

where $\mathbf{h}(x) = (h_1(x), \ldots, h_n(x))$. Furthermore, still by definition, we have

$$\psi_{\text{rob}}(\mathbf{z}) = \underset{k\in[K]}{\operatorname{argmax}}\left[\text{CWTM}\left(\mathbf{z}\right)\right]_k \quad \forall \mathbf{z} \in \Gamma_f(x).$$

Hence, whenever $\overline{h}(x)$ has a unique maximum coordinate, to satisfy $\psi_{\text{rob}}(\mathbf{z}) = \hat{y}_o$ for all $\mathbf{z} \in \Gamma_f(x)$, it suffices (see Appendix B.4 for details) to have

$$\max_{k\in[K]}\left|\left[\text{CWTM}\left(\mathbf{z}\right)\right]_k - \left[\overline{h}(x)\right]_k\right| < \frac{\text{MARGIN}\left(\overline{h}(x)\right)}{2}, \ \forall \mathbf{z} \in \Gamma_f(x) \text{ almost surely.} \tag{14}$$

In the remaining of the proof, we show that (14) holds as soon as

$$\text{MARGIN}\left(\overline{h}(x)\right) > 2\left(\sqrt{\frac{\kappa n}{n-f}} + \sqrt{\frac{f}{n-f}}\right)\sigma_x.$$

To do so, we first observe that for any $x \in \mathcal{X}$, $k \in [K]$, and $\mathbf{z} \in \Gamma_f(x)$ we have

$$\left|\left[\text{CWTM}(\mathbf{z})\right]_k - \left[\overline{h}(x)\right]_k\right| \leq \left|\left[\text{CWTM}(\mathbf{z})\right]_k - \left[\overline{h}_H(x)\right]_k\right| + \left|\left[\overline{h}_H(x)\right]_k - \left[\overline{h}(x)\right]_k\right|.$$

We then study each term of the right-hand side independently in (i) and (ii) below.

**(i) First term.** By definition of CWTM and Lemma 3, we know that for any $\mathbf{z} \in \Gamma_f(x)$ and $k \in [K]$, the following holds

$$\left|\left[\text{CWTM}\left(\mathbf{z}\right)\right]_k - \left[\overline{h}_H(x)\right]_k\right|^2 \leq \frac{\kappa}{n-f}\sum_{i\in H}\left(\left[h_i(x)\right]_k - \left[\overline{h}_H(x)\right]_k\right)^2,$$

with $\kappa = \frac{6f}{n-2f}\left(1 + \frac{f}{n-2f}\right)$. Furthermore, using Lemma 2, in the above we get

$$\left|\left[\text{CWTM}\left(\mathbf{z}\right)\right]_k - \left[\overline{h}_H(x)\right]_k\right|^2 \leq \left(\frac{\kappa n}{n-f}\right)\frac{1}{n}\sum_{i=1}^{n}\left(\left[h_i(x)\right]_k - \left[\overline{h}(x)\right]_k\right)^2.$$

Finally, using the definition of $\sigma_x^2$, we obtain

$$\left|\left[\text{CWTM}\left(\mathbf{z}\right)\right]_k - \left[\overline{h}_H(x)\right]_k\right|^2 \leq \frac{\kappa n}{n-f}\sigma_x^2.$$

**(ii) Second term.** Similar to the proof of Lemma 2, we split $[n]$ in elements of $H$ and $T = [n]\backslash H$. Then we can rewrite for any $k \in [K]$

$$\frac{1}{n}\sum_{i=1}^{n}([h_i(x)]_k - [\bar{h}(x)]_k)^2 = \frac{1}{n}\left(\sum_{i\in H}([h_i(x)]_k - [\bar{h}(x)]_k)^2 + \sum_{i\in T}([h_i(x)]_k - [\bar{h}(x)]_k)^2\right).$$

Furthermore, using Jensen's inequality, we have

$$\sum_{i\in H}([h_i(x)]_k - [\bar{h}(x)]_k)^2 \geq (n-f)\left([\bar{h}_H(x)]_k - [\bar{h}(x)]_k\right)^2 \tag{15}$$

and

$$\sum_{i\in T}([h_i(x)]_k - [\bar{h}(x)]_k)^2 \geq f\left([\bar{h}_T(x)]_k - [\bar{h}(x)]_k\right)^2,$$

with $\bar{h}_T(x) = \frac{1}{f}\sum_{i\in T}h_i(x)$. As $\bar{h}_T(x) = \frac{n\bar{h}(x)-(n-f)\bar{h}_H(x)}{f}$, the above also gives us

$$\sum_{i\in T}([h_i(x)]_k - [\bar{h}(x)]_k)^2 \geq \frac{(n-f)^2\left([\bar{h}_H(x)]_k-[\bar{h}(x)]_k\right)^2}{f}. \tag{16}$$

Then, combining (15) and (16) and since $(n-f) + \frac{(n-f)^2}{f} = \frac{n(n-f)}{f}$ we get

$$\sum_{i=1}^{n}([h_i(x)]_k - [\bar{h}(x)]_k)^2 \geq \frac{n(n-f)}{f}\left([\bar{h}_H(x)]_k - [\bar{h}(x)]_k\right)^2.$$

Rearranging in the above we obtain

$$\left([\bar{h}_H(x)]_k - [\bar{h}(x)]_k\right)^2 \leq \frac{f}{n-f}\frac{1}{n}\sum_{i=1}^{n}([h_i(x)]_k - [\bar{h}(x)]_k)^2 \leq \frac{f}{n-f}\sigma_x^2.$$

**Combining.** By substituting (i) and (ii) in the initial decomposition (applying $\sqrt{\cdot}$ on each), we get

$$\left|[\text{CWTM}(\mathbf{z})]_k - [\bar{h}(x)]_k\right| \leq \left(\sqrt{\frac{\kappa n}{n-f}} + \sqrt{\frac{f}{n-f}}\right)\sigma_x.$$

Thus, condition (14) is satisfied whenever

$$\text{MARGIN}(\bar{h}(x)) > 2\left(\sqrt{\frac{\kappa n}{n-f}} + \sqrt{\frac{f}{n-f}}\right)\sigma_x.$$

In particular, because we assumed $\bar{h}(x)$ has a unique maximum coordinate almost everywhere, the above implies that

$$\mathbb{P}_{(x,y)\sim\mathcal{D}}\left[\psi_{\text{rob}}(\mathbf{z}) = \hat{y}_{\text{o}}, \forall\mathbf{z}\in\Gamma_f(x)\right] \geq \mathbb{P}_{(x,y)\sim\mathcal{D}}\left[\text{MARGIN}\left(\bar{h}(x)\right) \geq 2\left(\sqrt{\frac{\kappa n}{n-f}} + \sqrt{\frac{f}{n-f}}\right)\sigma_x\right].$$

**Conclusion.** Plugging this into (13) completes the proof, i.e.,

$$\mathbb{E}_{(x,y)\sim\mathcal{D}}\left[\ell_{\psi_{\text{rob}}}^{\text{adv}}(x,\hat{y}_{\text{o}})\right] \leq \mathbb{P}_{(x,y)\sim\mathcal{D}}\left[\text{MARGIN}\left(\bar{h}(x)\right) < 2\left(\sqrt{\frac{\kappa n}{n-f}} + \sqrt{\frac{f}{n-f}}\right)\sigma_x\right].$$

$\square$

### B.4 JUSTIFICATION FOR (14)

Here we prove that, whenever $\bar{h}(x)$ has a unique maximum coordinate, to satisfy $\psi_{\text{rob}}(\mathbf{z}) = \hat{y}_{\text{o}}$ for all $\mathbf{z}\in\Gamma_f(x)$, it suffices to have

$$\max_{k\in[K]}\left|[\text{CWTM}(\mathbf{z})]_k - [\bar{h}(x)]_k\right| < \frac{\text{MARGIN}\left(\bar{h}(x)\right)}{2}, \ \forall\mathbf{z}\in\Gamma_f(x).$$

*Proof.* Let $x \in \mathcal{X}$ and $\bar{h} : \mathcal{X} \to \Delta^K$ the average function. Let us assume that (14) holds true. Then for any $\mathbf{z} \in \Gamma_f(x)$ and $k \in [K]$ we have

$$-\frac{\text{MARGIN}(\bar{h}(x))}{2} < [\text{CWTM}(\mathbf{z})]_k - [\bar{h}(x)]_k < \frac{\text{MARGIN}(\bar{h}(x))}{2}. \tag{17}$$

Let us denote by $k^*$ the coordinate such that $[\bar{h}(x)]_{k^*} = \bar{h}(x)^{(1)}$, i.e., the unique maximum of $\bar{h}(x)$. Let us also consider $k \in [K] \backslash k^*$, and $\mathbf{z} \in \Gamma_f(x)$ we have

$$[\text{CWTM}(\mathbf{z})]_{k^*} - [\text{CWTM}(\mathbf{z})]_k$$
$$= [\text{CWTM}(\mathbf{z})]_{k^*} - [\bar{h}(x)]_{k^*} - \left([\text{CWTM}(\mathbf{z})]_k - [\bar{h}(x)]_k\right) + [\bar{h}(x)]_{k^*} - [\bar{h}(x)]_k.$$

Using (17) on the first two terms and observing that $[\bar{h}(x)]_{k^*} - [\bar{h}(x)]_k \geq \text{MARGIN}(\bar{h}(x))$ by definition, we get

$$[\text{CWTM}(\mathbf{z})]_{k^*} - [\text{CWTM}(\mathbf{z})]_k > -\frac{\text{MARGIN}(\bar{h}(x))}{2} - \frac{\text{MARGIN}(\bar{h}(x))}{2} + \text{MARGIN}(\bar{h}(x)) = 0.$$

Since the above holds for any $k \in [K] \setminus k^*$ and $\mathbf{z} \in \Gamma_f(x)$, we get that $k^*$ is the unique maximum coordinate of $\text{CWTM}(\mathbf{z})$ for any $\mathbf{z} \in \Gamma_f(x)$. Hence, we get

$$\psi_{\text{rob}}(\mathbf{z}) = \underset{k \in [K]}{\text{argmax}} \, [\text{CWTM}(\mathbf{z})]_k = \underset{k \in [K]}{\text{argmax}} \, [\bar{h}(x)]_k = \hat{y}_{\text{o}} \text{ for all } \mathbf{z} \in \Gamma_f(x).$$

$\square$

## C  PROOF OF THEOREM 2

In this appendix, we give a proof for Theorem 2, which we restate below for the reader's convenience.

**Theorem 2.** *Consider $\psi_{rob}$ as defined in (10) with $\text{ROBAVG} = \text{CWTM}$. If the regressors $h_1, \ldots, h_n$ are such that $\phi_{\theta^*}(\mathbf{h}(x))$ has a unique maximum coordinate almost everywhere, then the following holds:*

$$\mathbb{E}_{(x,y) \sim \mathcal{D}} \left[ \ell^{adv}_{\psi_{rob}}(x, \hat{y}_o) \right]$$

$$\leq \mathbb{P}_{(x,y) \sim \mathcal{D}} \left[ \text{MARGIN}(\phi_{\theta^*}(\mathbf{h}(x))) < 2\sqrt{2} L_\mu L_\rho \left( \sqrt{\frac{\kappa n}{n-f}} + \sqrt{\frac{f}{n-f}} \right) \min\left\{ 1, \ \sqrt{K} \sigma_x \right\} \right],$$

*where $\hat{y}_o = \text{argmax}_{k \in [K]} [\phi_{\theta^*}(\mathbf{h}(x))]_k$ and $\kappa = \frac{6f}{n-2f} \left(1 + \frac{f}{n-2f}\right)$.*

*Proof.* Recall that

$$\phi_{\theta^*}(\mathbf{h}(x)) := \mu_{\theta_2^*} \left( \frac{1}{n} \sum_{i=1}^n \rho_{\theta_1^*}(h_i(x)) \right).$$

In the following, we denote $\|\cdot\|_2$ (i.e., the Euclidean norm) simply by $\|\cdot\|$. Consider an arbitrary $k \in [K]$. Due to Lipschitz continuity of $\mu_{\theta_2^*}(\cdot)$ we obtain that

$$\left| \left[ \mu_{\theta_2^*} \left( \text{ROBAVG}(\rho_{\theta_1^*}(z_1), \ldots, \rho_{\theta_1^*}(z_n)) \right) \right]_k - \left[ \mu_{\theta_2^*} \left( \frac{1}{n} \sum_{i=1}^n \rho_{\theta_1^*}(h_i(x)) \right) \right]_k \right|$$

$$\leq L_\mu \left\| \text{ROBAVG}(\rho_{\theta_1^*}(z_1), \ldots, \rho_{\theta_1^*}(z_n)) - \frac{1}{n} \sum_{i=1}^n \rho_{\theta_1^*}(h_i(x)) \right\|. \tag{18}$$

Consider any $j \in [K]$. Since $\text{ROBAVG} = \text{CWTM}$ is a coordinate-wise aggregator, we have

$$\left[ \text{ROBAVG}(\rho_{\theta_1^*}(z_1), \ldots, \rho_{\theta_1^*}(z_n)) \right]_j = \text{ROBAVG}\left( [\rho_{\theta_1^*}(z_1)]_j, \ldots, [\rho_{\theta_1^*}(z_n)]_j \right).$$

Therefore,

$$\left[ \text{ROBAVG}(\rho_{\theta_1^*}(z_1), \ldots, \rho_{\theta_1^*}(z_n)) \right]_j - \left[ \frac{1}{n} \sum_{i=1}^n \rho_{\theta_1^*}(h_i(x)) \right]_j$$

$$= \text{ROBAVG}\left( [\rho_{\theta_1^*}(z_1)]_j, \ldots, [\rho_{\theta_1^*}(z_n)]_j \right) - \frac{1}{n} \sum_{i=1}^n [\rho_{\theta_1^*}(h_i(x))]_j.$$

Therefore, by the robustness property of CWTM, we obtain that for all $j \in [K]$,

$$
\left| \left[ \text{ROBAVG} \left( \rho_{\theta_1^*}(z_1), \dots, \rho_{\theta_1^*}(z_n) \right) \right]_j - \left[ \frac{1}{n} \sum_{i=1}^{n} \rho_{\theta_1^*}(h_i(x)) \right]_j \right|^2
$$
$$
\leq \left( \sqrt{\frac{\kappa n}{n-f}} + \sqrt{\frac{f}{n-f}} \right)^2 \frac{1}{n} \sum_{i=1}^{n} \left( \left[ \rho_{\theta_1^*}(h_i(x)) \right]_j - \left[ \overline{\rho_{\theta_1^*}}(x) \right]_j \right)^2, \tag{19}
$$

where $\overline{\rho_{\theta_1^*}}(x) = \frac{1}{n} \sum_{i=1}^{n} \rho_{\theta_1^*}(h_i(x))$. Applying Jensen's inequality, we obtain that

$$
\sum_{i=1}^{n} \left( \left[ \rho_{\theta_1^*}(h_i(x)) \right]_j - \left[ \overline{\rho_{\theta_1^*}}(x) \right]_j \right)^2 \leq \frac{1}{n} \sum_{i,l=1}^{n} \left( \left[ \rho_{\theta_1^*}(h_i(x)) \right]_j - \left[ \rho_{\theta_1^*}(h_l(x)) \right]_j \right)^2.
$$

Therefore, due to Lipschitz continuity of $\rho_{\theta_1^*}(\cdot)$, we have

$$
\sum_{i=1}^{n} \left( \left[ \rho_{\theta_1^*}(h_i(x)) \right]_j - \left[ \overline{\rho_{\theta_1^*}}(x) \right]_j \right)^2 \leq \frac{L_\rho^2}{n} \sum_{i,l=1}^{n} \left( \left[ h_i(x) \right]_j - \left[ h_l(x) \right]_j \right)^2
$$
$$
= 2 L_\rho^2 \sum_{i=1}^{n} \left( \left[ h_i(x) \right]_j - \left[ \overline{h}(x) \right]_j \right)^2.
$$

Therefore, for all $j \in [K]$,

$$
\frac{1}{n} \sum_{i=1}^{n} \left( \left[ \rho_{\theta_1^*}(h_i(x)) \right]_j - \left[ \overline{\rho_{\theta_1^*}}(x) \right]_j \right)^2 \leq \frac{2 L_\rho^2}{n} \sum_{i=1}^{n} \left( \left[ h_i(x) \right]_j - \left[ \overline{h}(x) \right]_j \right)^2.
$$

Using this in (19), we obtain that

$$
\left\| \text{ROBAVG} \left( \rho_{\theta_1^*}(z_1), \dots, \rho_{\theta_1^*}(z_n) \right) - \frac{1}{n} \sum_{i=1}^{n} \rho_{\theta_1^*}(h_i(x)) \right\|^2
$$
$$
\leq 2 L_\rho^2 \left( \sqrt{\frac{\kappa n}{n-f}} + \sqrt{\frac{f}{n-f}} \right)^2 \frac{1}{n} \sum_{i=1}^{n} \left\| h_i(x) - \overline{h}(x) \right\|^2.
$$

Substituting from above in (18), we obtain that

$$
\left| \left[ \mu_{\theta_2^*} \left( \text{ROBAVG} \left( \rho_{\theta_1^*}(z_1), \dots, \rho_{\theta_1^*}(z_n) \right) \right) \right]_k - \left[ \mu_{\theta_2^*} \left( \frac{1}{n} \sum_{i=1}^{n} \rho_{\theta_1^*}(h_i(x)) \right) \right]_k \right|
$$
$$
\leq \sqrt{2} L_\mu L_\rho \left( \sqrt{\frac{\kappa n}{n-f}} + \sqrt{\frac{f}{n-f}} \right) \sqrt{\frac{1}{n} \sum_{i=1}^{n} \left\| h_i(x) - \overline{h}(x) \right\|^2}. \tag{20}
$$

Note that for any $v \in R^d$, if $\left| [v]_j \right| \leq 1$ then $\|v\|^2 \leq \|v\|_1 := \sum_{j=1}^{n} \left| [v]_j \right|$. This, together with the fact that $h_i(x) \in \Delta^K$ for all $i$, implies that

$$
\frac{1}{n} \sum_{i=1}^{n} \left\| h_i(x) - \overline{h}(x) \right\|^2 = \frac{1}{n} \sum_{i=1}^{n} \left\| h_i(x) \right\|^2 - \left\| \overline{h}(x) \right\|^2 \leq \frac{1}{n} \sum_{i=1}^{n} \left\| h_i(x) \right\|^2 \leq \frac{1}{n} \sum_{i=1}^{n} \left\| h_i(x) \right\|_1 = 1.
$$

Moreover, recalling that $\sigma_x^2 := \max_{k \in [K]} \frac{1}{n} \sum_{i=1}^{n} \left( [h_i(x)]_k - \left[ \overline{h}(x) \right]_k \right)^2$, we also have

$$
\frac{1}{n} \sum_{i=1}^{n} \left\| h_i(x) - \overline{h}(x) \right\|^2 = \sum_{j=1}^{K} \frac{1}{n} \sum_{i=1}^{n} \left( [h_i(x)]_j - \left[ \overline{h}(x) \right]_j \right)^2 \leq K \sigma_x^2.
$$

Using the above in (20) yields,

$$
\left| \left[ \mu_{\theta_2^*} \left( \text{ROBAVG} \left( \rho_{\theta_1^*}(z_1), \ldots, \rho_{\theta_1^*}(z_n) \right) \right) \right]_k - \left[ \mu_{\theta_2^*} \left( \frac{1}{n} \sum_{i=1}^{n} \rho_{\theta_1^*}(h_i(x)) \right) \right]_k \right|
$$
$$
\leq \sqrt{2} L_\mu L_\rho \left( \sqrt{\frac{\kappa n}{n-f}} + \sqrt{\frac{f}{n-f}} \right) \min\{1, \sqrt{K}\sigma_x\}.
$$

Note that the above holds true for any $k \in [K]$. Hence, similar to (14), whenever $\mu_{\theta_2^*} \left( \frac{1}{n} \sum_{i=1}^{n} \rho_{\theta_1^*}(h_i(x)) \right)$ has a unique minimum coordinate almost everywhere, if

$$
\text{MARGIN} \left( \mu_{\theta_2^*} \left( \frac{1}{n} \sum_{i=1}^{n} \rho_{\theta_1^*}(h_i(x)) \right) \right) > 2\sqrt{2} L_\mu L_\rho \left( \sqrt{\frac{\kappa n}{n-f}} + \sqrt{\frac{f}{n-f}} \right) \min\{1, \sqrt{K}\sigma_x\},
$$

then

$$
\underset{k}{\text{argmax}} \left[ \mu_{\theta_2^*} \left( \text{ROBAVG} \left( \rho_{\theta_1^*}(z_1), \ldots, \rho_{\theta_1^*}(z_n) \right) \right) \right]_k = \underset{k}{\text{argmax}} \left[ \mu_{\theta_2^*} \left( \frac{1}{n} \sum_{i=1}^{n} \rho_{\theta_1^*}(h_i(x)) \right) \right]_k.
$$

The rest of the proof follows the same reasoning as in the final steps of the proof of Theorem 1. $\quad\square$

## D  SENSITIVITY OF DEEPSET AGGREGATOR WITHOUT ROBUST AVERAGING

In this appendix, we show how the use of robust averaging enhances the robustness of the DeepSet aggregator. Specifically, we analyze the robustness gap of the following aggregator and compare it with that of robust aggregator defined in (10). The robustness gap of the latter is characterized in Theorem 2.

$$
\psi_{\text{rob}}(\mathbf{z}) \coloneqq \underset{k \in [K]}{\text{argmax}} \left[ \mu_{\theta_2^*} \left( \frac{1}{n} \sum_{i=1}^{n} \rho_{\theta_1^*}(z_i) \right) \right]_k. \tag{21}
$$

Throughout this appendix, we work under the same assumptions and notations as those in Appendix C. Consider an arbitrary $k \in [K]$. Due to Lipschitz continuity of $\mu_{\theta_2^*}(\cdot)$ we obtain that

$$
\left| \left[ \mu_{\theta_2^*} \left( \frac{1}{n} \sum_{i=1}^{n} \rho_{\theta_1^*}(z_i) \right) \right]_k - \left[ \mu_{\theta_2^*} \left( \frac{1}{n} \sum_{i=1}^{n} \rho_{\theta_1^*}(h_i(x)) \right) \right]_k \right| \leq L_\mu \left\| \frac{1}{n} \sum_{i=1}^{n} \left( \rho_{\theta_1^*}(z_i) - \rho_{\theta_1^*}(h_i(x)) \right) \right\|.
$$

Due to Lipschitz continuity of $\rho_{\theta_1^*}(\cdot)$, for any $j \in [K]$, we have

$$
\left| \frac{1}{n} \sum_{i=1}^{n} \left[ \rho_{\theta_1^*}(z_i) - \rho_{\theta_1^*}(h_i(x)) \right]_j \right| \leq \frac{1}{n} \sum_{i=1}^{n} \left| \left[ \rho_{\theta_1^*}(z_i) - \rho_{\theta_1^*}(h_i(x)) \right]_j \right| \leq \frac{L_\rho}{n} \sum_{i=1}^{n} \| z_i - h_i(x) \|.
$$

Note that $z_i = h_i(x)$ for at least $n - f$ indices $i \in [n]$ (corresponding to honest uncorrupted probits). That is, there exists $B_x \subset [n]$ with $|B_x| = f$ such that $z_i = h_i(x)$ for all $i \in [n] \setminus B_x$. Therefore,

$$
\left| \frac{1}{n} \sum_{i=1}^{n} \left[ \rho_{\theta_1^*}(z_i) - \rho_{\theta_1^*}(h_i(x)) \right]_j \right| \leq \frac{L_\rho}{n} \sum_{i \in B_x} \| z_i - h_i(x) \|.
$$

Let $\delta \coloneqq \max_{i \in B_x} \| z_i - h_i(x) \|$, be the degree of probits' corruption. Then, for all $j \in [K]$,

$$
\left| \frac{1}{n} \sum_{i=1}^{n} \left[ \rho_{\theta_1^*}(z_i) - \rho_{\theta_1^*}(h_i(x)) \right]_j \right| \leq L_\rho \frac{f}{n} \delta.
$$

This implies that

$$
\left\| \frac{1}{n} \sum_{i=1}^{n} \left( \rho_{\theta_1^*}(z_i) - \rho_{\theta_1^*}(h_i(x)) \right) \right\| \leq \sqrt{K} L_\rho \frac{f}{n} \delta.
$$

Therefore, we have for all $k \in [K]$,

$$\left| \left[ \mu_{\theta_2^*} \left( \frac{1}{n} \sum_{i=1}^{n} \rho_{\theta_1^*}(z_i) \right) \right]_k - \left[ \mu_{\theta_2^*} \left( \frac{1}{n} \sum_{i=1}^{n} \rho_{\theta_1^*}(h_i(x)) \right) \right]_k \right| \leq \sqrt{K} L_\mu L_\rho \frac{f}{n} \delta.$$

Hence, $\mathrm{argmax}_{k \in [K]}[\phi_{\theta^*}(\mathbf{z})]_k = \mathrm{argmax}_{k \in [K]}[\phi_{\theta^*}(\mathbf{h}(x))]_k$ whenever

$$\mathrm{MARGIN}\left(\phi_{\theta^*}(\mathbf{h}(x))\right) > 2\sqrt{K} L_\mu L_\rho \frac{f}{n} \delta.$$

Under the assumption that $\phi_{\theta^*}(\mathbf{h}(x))$ has a unique maximum coordinate almost everywhere, the above yields the following bound on the robustness gap for the aggregator defined in (21).

$$\mathbb{E}_{(x,y)\sim\mathcal{D}} \left[ \ell_{\psi_{\mathrm{rob}}}^{\mathrm{adv}}(x, \hat{y}_{\mathrm{o}}) \right] \leq \mathbb{P}_{(x,y)\sim\mathcal{D}} \left[ \mathrm{MARGIN}\left(\phi_{\theta^*}(\mathbf{h}(x))\right) < 2\sqrt{K} L_\mu L_\rho \frac{f}{n} \delta \right]. \qquad (22)$$

**Comparison to Theorem 2.** Comparing (22) to the robustness gap shown in Theorem 2 for the robust aggregator defined in (10) we note a key difference:

> While the robustness gap without robust averaging depends upon the degree of corruption $\delta$, it is rendered independent of this degree of corruption when we use a robust averaging scheme like CWTM.

In fact, when the variance between honest probits is small, the robustness gap under robust averaging is much smaller compared to the case without robust averaging. For instance, consider the case when $h_i(x) = h_j(x)$ for all $i, j \in [n]$ and all $x$. While the robustness gap when using CWTM is $0$, as per Theorem 2, that need not be the case without robust averaging as is evident by (22).

## E  ADDITIONAL RELATED WORK

In the context of robust voting within federated learning, Chu & Laoutaris (2024) propose FedQV, a quadratic voting scheme that allocates voting budgets based on client reputation. This presumes stable client identities and therefore does not transfer to our setting where malicious participants may change per query. In contrast, the ensemble majority voting scheme introduced in Cao et al. (2022) does not apply because it requires central control over training to obtain several global models, each trained on a subset of clients. In our setting, however, training is fully local and independent.

Beyond federated learning, robust voting has also been studied in more general settings. Allouah et al. (2024b) propose Mehestan for robust sparse voting, but it requires normalization across voters, which assumes repeated score comparisons or stable voter behavior. This is not applicable to our per-query federated inference setting. Likewise, Melnyk et al. (2018) develop multi-round consensus for preference ranking and Datar et al. (2022) rely on repeated pairwise client comparisons. Both assume settings that are fundamentally different from our inference-time aggregation scenario.

## F  ADDITIONAL EXPERIMENTAL DETAILS

### F.1  HYPERPARAMETERS

In this section, we report the hyperparameters used for our experiments, and which can be used to reproduce the results. The local training/fine-tuning is conducted using SGD optimizer and a learning rate of $0.0025$ for CIFAR-10 and $0.01$ for both CIFAR-100 and AG-News. The training runs for $100$ local epochs for CIFAR-10 while the fine-tuning of CIFAR-100 and AG-News runs for $20$ epochs. The DeepSet model is trained using the Adam optimizer for $10$ epochs for each dataset using a learning rate of $5 \times 10^{-5}$. Our choices of parameters are based on prior work using similar setups (Allouah et al., 2024a; Gong et al., 2022). We train the autoencoder in COPUR using the Adam optimizer and a learning rate of $1 \times 10^{-3}$, and vary the epochs from $\{50, 100, 150\}$ depending on the dataset. The server's aggregator model in COPUR is also trained using Adam for $40$ epochs, and a learning rate of $1 \times 10^{-3}$. Finally, the number of optimization iterations for COPUR is varied from $10$ to $40$, while Liu et al. (2022) use a fixed value of $10$. During adversarial training, we sample different

choices of adversaries out of $[n]$ as their identify is unknown *i.e.*, parameter $N$ in Algorithm 1. While the number of choices total to $\sum_{i=m}^{f} \binom{n}{m}$, we only sample $N = 120, N = 300$ and $N = 5000$ for each of $n = 10, 17$ and $25$ respectively. Lastly, we set $S = 50$ and $E$ such that it corresponds to 5 epochs over the corresponding training dataset.

## F.2 DESCRIPTION OF ATTACKS

Table 4: Table listing all attacks for an input $x$ with its true label $y$. Here, $\psi$ is any aggregation in consideration which is employed at the server. We refer with $\Omega_{\text{benign}}$ the set of benign client indexes such that $\Omega_{\text{benign}} \subseteq [n]$ and $|\Omega_{\text{benign}}| = n - f$.

| Attack | Type | Definition |
|---|---|---|
| Logit Flipping Liu et al. (2022) | Black-box | $z_i = -\text{amplification} * h_i(x)$ |
| SIA-bb (ours) | Black-box | $z_i^j = \begin{cases} 1, & \text{if } j = \arg\max_{k \in [K] \setminus y} h_i(x) \\ 0, & \text{otherwise} \end{cases}$ |
| LMA Roux et al. (2025) | White-box | $z_i^j = \begin{cases} 1, & \text{if } j = \arg\min_{k \in [K]} \psi(h_1(x), \ldots, h_n(x)) \\ 0, & \text{otherwise} \end{cases}$ |
| CPA Roux et al. (2025) | White-box | $z_i^j = \begin{cases} 1, & \text{if } j \text{ is least similar to } \arg\max_{k \in [K]} \frac{1}{n} \sum_{i=1}^{n} h_i(x) \\ 0, & \text{otherwise} \end{cases}$ |
| SIA-wb (ours) | White-box | $z_i^j = \begin{cases} 1, & \text{if } j = \arg\max_{k \in [K] \setminus y} \psi(h_1(x), \ldots, h_n(x)) \\ 0, & \text{otherwise} \end{cases}$ |
| PGD Liu et al. (2022) | White-box | $z_1^*, \ldots, z_f^* = \arg\max_{(z_1, \ldots, z_f)} \ell(\psi(\{h_{\text{benign}}, h_{\text{adv}}\}), y)$ where $h_{\text{benign}} = \{h_i(x) \mid i \in \Omega_{\text{benign}}\}$ and $h_{\text{adv}} = \{z_1, \ldots, z_f\}$ |

We consider six adversarial attacks, designed to evaluate the robustness of the aggregator under varying levels of adversary knowledge and coordination. Beyond the mathematical definitions in Table 4, we provide a short intuitive description for each below:

- **Logit flipping (*Black-box*).**
  Each adversary inverts its own probits by negating and scaling them, effectively pushing its prediction away from the true class without knowledge of the server's aggregation or other clients' predictions.

- **Strongest Inverted Attack (SIA)** [Our Proposed Attack].

  - *Black-box:* Adversaries independently change their prediction to the second-most probable class (second-largest local probit) that is not the true class.

  - *White-box:* Same as black-box except that the adversaries use the global aggregation output (before perturbation) to identify the second-most probable class.

- **Loss Maximization attack (LMA, *White-box*).**
  Adversaries maximize the server's loss by targeting the least likely class among the global aggregation output.

- **Class Prior Attack (CPA, *White-box*).**
  Adversaries first identify the most likely class from the aggregation output. They then select the class least similar to it according to a pre-computed similarity matrix $S \in \mathbb{R}^{K \times K}$, derived from class embeddings of a pre-trained reference model.

- **PGD Attack (*White-box*).**
  Adversaries iteratively optimize their output via gradient ascent to maximize the aggregator's loss. This requires specifying a loss function for the aggregator. Two natural choices are the standard cross-entropy loss and the Carlini-Wagner (CW) loss (Carlini & Wagner, 2017). We adopt the latter (PGD-cw) in our experiments as it produces stronger adversarial attacks: instead of penalizing all classes equally, it focuses on the two most likely classes, thereby targeting the decision boundary more effectively.

### F.3 ADAPTING SIA ATTACK TO THE LOGIT SPACE

Since COPUR operates directly in logit space, we adapted the SIA definitions to ensure comparability with DeepSet-TM in Section 5.2.

**Proposed adaptation.** Each adversary identifies the largest logit $M$ either from its own output (black-box) or from the aggregation output (white-box). Given an amplification factor $\ell > 0$, the adversary then sets its own logits $(l'_i)_{i \in [K]}$ as such:

$$l'_i = \begin{cases} \ell \times M & \text{if } i \text{ is the index of the second-largest logit} \\ -\ell \times M & \text{o.w.} \end{cases}$$

using $-m$ instead of $M$ if $M \leq 0$ (where $m$ is the smallest logit).

In our experiments, we set $\ell = 2$ since we found the baseline COPUR to be highly negatively affected by larger values.

## G ADDITIONAL EXPERIMENTAL RESULTS

### G.1 EXPERIMENTAL VALIDATION OF THEOREM 1

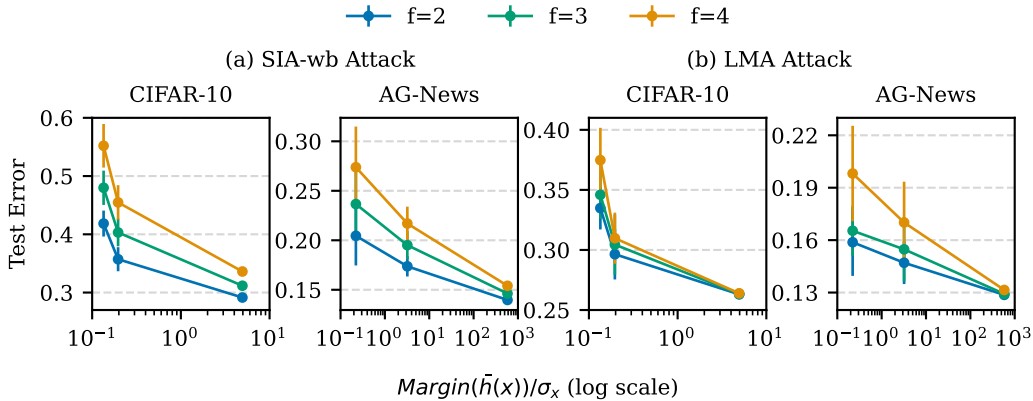

Figure 3: Validating Theorem 1 in practice. As the ratio of MARGIN$(\overline{h}(x))/\sigma_x$ increases, we observe a notable decrease in the test error, aligning with Theorem 1. Adversaries use SIA white-box and LMA (see Table 4).

We empirically validate the theoretical predictions of Theorem 1 by analyzing the evolution of test error as a function of MARGIN$(\overline{h}(x))/\sigma_x$. According to the theorem, larger values of this quantity translate into lower test error. Figure 3 is generated by varying the heterogeneity level $\alpha \in \{0.5, 1, 1000\}$ for each of the two datasets and computing the average MARGIN$(\overline{h}(x))/\sigma_x$ ratio on the testset of the respective dataset. Increasing $\alpha$ reduces the model dissimilarity $\sigma_x$, thereby increasing the ratio above. We then report test error for CWTM on each configuration to obtain the figure. As expected, a higher number of adversaries $f$ consistently leads to higher test error. The lower the heterogeneity, the less significant this effect is, since lower heterogeneity allows for more redundancy within client outputs.

### G.2 PERFORMANCE AGAINST ROBUST AGGREGATORS

#### G.2.1 DECREASING THE NUMBER OF ADVERSARIES

Table 5 shows the performance of DeepSet-TM and static aggregations as in Table 2, but decreasing the number of adversaries $f$ from 4 to 3.

Table 5: Accuracy (%) of DeepSet-TM against static aggregations on the CIFAR-10, CIFAR-100 and AG-News datasets, with heterogeneity $\alpha = 0.5$, $n = 17$ clients and $f = 3$ adversaries. Logit flipping uses an amplification factor of 2.

|  | Aggregation | Logit flipping | SIA-bb | LMA | CPA | SIA-wb | PGD-cw | Worst case |
|---|---|---|---|---|---|---|---|---|
| **CF-10** | Mean | $66.7 \pm 2.2$ | $62.5 \pm 1.8$ | $66.7 \pm 1.9$ | $64.7 \pm 2.1$ | $49.7 \pm 3.2$ | $35.8 \pm 4.4$ | $35.8 \pm 4.4$ |
|  | CWMed | $58.3 \pm 5.2$ | $55.8 \pm 3.0$ | $57.4 \pm 4.0$ | $56.7 \pm 4.2$ | $54.2 \pm 3.1$ | $36.5 \pm 4.1$ | $36.5 \pm 4.1$ |
|  | GM | $65.0 \pm 2.4$ | $61.5 \pm 1.8$ | $65.6 \pm 1.8$ | $64.2 \pm 2.0$ | $51.5 \pm 3.1$ | $36.0 \pm 4.8$ | $36.0 \pm 4.8$ |
|  | CWTM | $65.0 \pm 2.5$ | $61.9 \pm 1.7$ | $65.4 \pm 2.0$ | $64.5 \pm 2.0$ | $52.0 \pm 2.9$ | $38.3 \pm 4.2$ | $38.3 \pm 4.2$ |
|  | DeepSet-TM | $\mathbf{68.5 \pm 1.1}$ | $\mathbf{65.4 \pm 1.8}$ | $\mathbf{67.8 \pm 2.1}$ | $\mathbf{66.6 \pm 2.1}$ | $\mathbf{57.2 \pm 1.5}$ | $\mathbf{53.9 \pm 3.2}$ | $53.9 \pm 3.2$ |
| **CF-100** | Mean | $\mathbf{79.3 \pm 0.7}$ | $75.4 \pm 0.8$ | $76.1 \pm 0.7$ | $76.0 \pm 0.8$ | $62.9 \pm 0.9$ | $52.1 \pm 1.1$ | $52.1 \pm 1.1$ |
|  | CWMed | $68.9 \pm 1.6$ | $66.9 \pm 1.4$ | $68.9 \pm 1.5$ | $68.9 \pm 1.5$ | $66.2 \pm 1.2$ | $52.4 \pm 1.2$ | $52.4 \pm 1.2$ |
|  | GM | $75.9 \pm 1.0$ | $73.6 \pm 1.2$ | $76.1 \pm 1.0$ | $76.1 \pm 0.9$ | $64.9 \pm 1.1$ | $51.7 \pm 1.3$ | $51.7 \pm 1.3$ |
|  | CWTM | $75.9 \pm 1.0$ | $73.7 \pm 1.2$ | $75.9 \pm 1.0$ | $75.9 \pm 1.0$ | $66.5 \pm 0.9$ | $56.2 \pm 1.1$ | $56.2 \pm 1.1$ |
|  | DeepSet-TM | $78.6 \pm 0.3$ | $\mathbf{76.4 \pm 0.4}$ | $\mathbf{78.2 \pm 0.2}$ | $\mathbf{78.1 \pm 0.3}$ | $\mathbf{68.3 \pm 0.4}$ | $\mathbf{59.5 \pm 0.5}$ | $59.5 \pm 0.5$ |
| **AG-News** | Mean | $84.1 \pm 1.1$ | $82.7 \pm 2.4$ | $83.8 \pm 1.4$ | $81.2 \pm 2.3$ | $76.4 \pm 3.7$ | $65.3 \pm 6.2$ | $65.3 \pm 6.2$ |
|  | CWMed | $80.3 \pm 4.5$ | $81.3 \pm 3.5$ | $81.3 \pm 2.9$ | $80.1 \pm 3.0$ | $77.5 \pm 3.5$ | $63.5 \pm 6.8$ | $63.5 \pm 6.8$ |
|  | GM | $83.0 \pm 1.5$ | $82.1 \pm 2.7$ | $83.3 \pm 1.6$ | $80.7 \pm 2.4$ | $77.4 \pm 3.1$ | $63.1 \pm 6.9$ | $63.1 \pm 6.9$ |
|  | CWTM | $83.9 \pm 1.1$ | $82.6 \pm 2.4$ | $83.5 \pm 1.4$ | $81.2 \pm 2.3$ | $76.4 \pm 3.8$ | $65.4 \pm 6.2$ | $65.4 \pm 6.2$ |
|  | DeepSet-TM | $\mathbf{85.9 \pm 0.2}$ | $\mathbf{83.8 \pm 1.2}$ | $\mathbf{84.1 \pm 0.5}$ | $\mathbf{82.7 \pm 0.9}$ | $\mathbf{81.9 \pm 1.2}$ | $\mathbf{83.5 \pm 1.2}$ | $83.5 \pm 1.2$ |

Table 6: Accuracy (%) of DeepSet-TM against static aggregations on the CIFAR-10, CIFAR-100 and AG-News datasets, with heterogeneity $\alpha = 0.3$, $n = 17$ clients and $f = 4$ adversaries. Logit flipping uses an amplification factor of 2.

|  | Aggregation | Logit flipping | SIA-bb | LMA | CPA | SIA-wb | PGD-cw | Worst case |
|---|---|---|---|---|---|---|---|---|
| **CF-10** | Mean | $60.4 \pm 4.0$ | $55.8 \pm 4.3$ | $49.0 \pm 3.9$ | $43.8 \pm 4.6$ | $29.8 \pm 3.8$ | $13.6 \pm 2.8$ | $13.6 \pm 2.8$ |
|  | CWMed | $47.1 \pm 6.7$ | $45.8 \pm 3.7$ | $45.9 \pm 4.3$ | $42.4 \pm 4.2$ | $38.4 \pm 2.4$ | $14.1 \pm 3.4$ | $14.1 \pm 3.4$ |
|  | GM | $58.2 \pm 4.3$ | $55.0 \pm 4.3$ | $\mathbf{55.6 \pm 4.1}$ | $49.7 \pm 4.2$ | $34.8 \pm 2.8$ | $13.7 \pm 2.9$ | $13.7 \pm 2.9$ |
|  | CWTM | $57.6 \pm 4.2$ | $54.6 \pm 4.3$ | $54.2 \pm 4.2$ | $49.4 \pm 4.5$ | $32.1 \pm 3.6$ | $16.4 \pm 2.9$ | $16.4 \pm 2.9$ |
|  | DeepSet-TM | $\mathbf{61.9 \pm 3.1}$ | $\mathbf{57.9 \pm 3.2}$ | $46.6 \pm 3.6$ | $44.7 \pm 6.0$ | $\mathbf{43.0 \pm 2.2}$ | $\mathbf{44.9 \pm 2.7}$ | $43.0 \pm 2.2$ |
| **CF-100** | Mean | $75.6 \pm 0.7$ | $69.1 \pm 0.8$ | $57.0 \pm 2.4$ | $56.8 \pm 2.4$ | $44.5 \pm 1.4$ | $25.8 \pm 2.8$ | $25.8 \pm 2.8$ |
|  | CWMed | $55.1 \pm 2.1$ | $51.8 \pm 2.6$ | $55.9 \pm 2.2$ | $55.8 \pm 2.2$ | $50.7 \pm 1.8$ | $24.5 \pm 1.9$ | $24.5 \pm 1.9$ |
|  | GM | $71.2 \pm 0.8$ | $66.3 \pm 1.1$ | $70.2 \pm 0.8$ | $70.0 \pm 0.9$ | $48.3 \pm 1.5$ | $23.8 \pm 2.6$ | $23.8 \pm 2.6$ |
|  | CWTM | $69.1 \pm 1.1$ | $65.8 \pm 1.2$ | $69.1 \pm 1.2$ | $69.1 \pm 1.2$ | $49.6 \pm 1.1$ | $32.5 \pm 2.4$ | $32.5 \pm 2.4$ |
|  | DeepSet-TM | $\mathbf{75.7 \pm 0.8}$ | $\mathbf{71.9 \pm 1.1}$ | $\mathbf{72.2 \pm 1.9}$ | $\mathbf{72.5 \pm 0.6}$ | $\mathbf{56.7 \pm 1.0}$ | $\mathbf{38.6 \pm 2.5}$ | $38.6 \pm 2.5$ |
| **AG-News** | Mean | $77.0 \pm 4.6$ | $70.7 \pm 6.2$ | $67.1 \pm 8.1$ | $61.7 \pm 7.7$ | $62.3 \pm 7.7$ | $36.8 \pm 9.7$ | $36.8 \pm 9.7$ |
|  | CWMed | $60.9 \pm 9.4$ | $62.6 \pm 7.5$ | $54.4 \pm 8.7$ | $52.3 \pm 8.4$ | $66.0 \pm 7.4$ | $32.7 \pm 7.3$ | $32.7 \pm 7.3$ |
|  | GM | $74.7 \pm 4.9$ | $70.5 \pm 6.1$ | $\mathbf{67.8 \pm 7.8}$ | $62.6 \pm 7.0$ | $64.7 \pm 6.3$ | $33.2 \pm 8.7$ | $33.2 \pm 8.7$ |
|  | CWTM | $76.7 \pm 4.6$ | $70.6 \pm 6.2$ | $66.1 \pm 8.1$ | $\mathbf{61.8 \pm 7.7}$ | $62.6 \pm 7.6$ | $37.3 \pm 9.7$ | $37.3 \pm 9.7$ |
|  | DeepSet-TM | $\mathbf{80.4 \pm 3.5}$ | $\mathbf{73.6 \pm 4.5}$ | $61.1 \pm 8.9$ | $59.5 \pm 9.1$ | $\mathbf{73.6 \pm 3.8}$ | $\mathbf{72.0 \pm 10.3}$ | $59.5 \pm 9.1$ |

Table 7: Accuracy (%) of DeepSet-TM against static aggregations on the CIFAR-10, CIFAR-100 and AG-News datasets, with heterogeneity $\alpha = 0.8$, $n = 17$ clients and $f = 4$ adversaries. Logit flipping uses an amplification factor of 2.

|  | Aggregation | Logit flipping | SIA | LMA | CPA | SIA-wb | PGD-cw | Worst case |
|---|---|---|---|---|---|---|---|---|
| **CF-10** | Mean | $69.5 \pm 0.3$ | $64.6 \pm 1.5$ | $65.1 \pm 0.6$ | $63.1 \pm 0.9$ | $50.1 \pm 1.7$ | $33.5 \pm 2.6$ | $33.5 \pm 2.6$ |
|  | CWMed | $63.6 \pm 2.4$ | $59.4 \pm 2.0$ | $61.1 \pm 1.4$ | $60.1 \pm 1.4$ | $\mathbf{55.1 \pm 1.3}$ | $36.9 \pm 3.4$ | $36.9 \pm 3.4$ |
|  | GM | $68.4 \pm 0.4$ | $64.1 \pm 1.5$ | $\mathbf{67.9 \pm 0.6}$ | $\mathbf{66.1 \pm 0.9}$ | $52.8 \pm 1.2$ | $35.2 \pm 3.0$ | $35.2 \pm 3.0$ |
|  | CWTM | $68.3 \pm 0.3$ | $64.1 \pm 1.4$ | $67.8 \pm 0.7$ | $\mathbf{66.1 \pm 1.0}$ | $52.0 \pm 1.6$ | $36.2 \pm 2.6$ | $36.2 \pm 2.6$ |
|  | DeepSet-TM | $\mathbf{69.9 \pm 1.1}$ | $\mathbf{67.1 \pm 1.3}$ | $67.0 \pm 2.6$ | $\mathbf{66.1 \pm 2.1}$ | $55.0 \pm 2.2$ | $\mathbf{51.2 \pm 1.2}$ | $51.2 \pm 1.2$ |
| **CF-100** | Mean | $\mathbf{80.5 \pm 0.5}$ | $76.2 \pm 0.7$ | $71.5 \pm 1.2$ | $71.5 \pm 1.2$ | $62.5 \pm 1.1$ | $48.4 \pm 1.5$ | $48.4 \pm 1.5$ |
|  | CWMed | $72.9 \pm 1.5$ | $70.8 \pm 1.6$ | $73.0 \pm 1.4$ | $73.0 \pm 1.4$ | $\mathbf{69.0 \pm 1.3}$ | $54.8 \pm 1.7$ | $54.8 \pm 1.7$ |
|  | GM | $78.3 \pm 0.8$ | $75.5 \pm 0.9$ | $\mathbf{78.5 \pm 0.8}$ | $78.4 \pm 0.8$ | $66.0 \pm 1.1$ | $51.2 \pm 1.7$ | $51.2 \pm 1.7$ |
|  | CWTM | $78.1 \pm 1.0$ | $75.6 \pm 0.8$ | $78.1 \pm 1.0$ | $78.1 \pm 1.0$ | $67.1 \pm 1.0$ | $54.1 \pm 1.4$ | $54.1 \pm 1.4$ |
|  | DeepSet-TM | $79.5 \pm 0.4$ | $\mathbf{77.1 \pm 0.5}$ | $78.4 \pm 0.4$ | $\mathbf{78.6 \pm 0.6}$ | $68.0 \pm 0.8$ | $\mathbf{57.0 \pm 1.3}$ | $57.0 \pm 1.3$ |
| **AG-News** | Mean | $85.1 \pm 0.9$ | $81.1 \pm 2.8$ | $\mathbf{83.7 \pm 2.4}$ | $81.0 \pm 3.7$ | $77.8 \pm 3.2$ | $65.6 \pm 5.8$ | $65.6 \pm 5.8$ |
|  | CWMed | $80.6 \pm 3.2$ | $80.3 \pm 2.6$ | $81.8 \pm 2.4$ | $80.2 \pm 2.8$ | $78.1 \pm 2.6$ | $64.8 \pm 5.8$ | $64.8 \pm 5.8$ |
|  | GM | $84.3 \pm 1.1$ | $80.8 \pm 2.8$ | $83.6 \pm 2.4$ | $81.0 \pm 3.7$ | $78.2 \pm 3.1$ | $64.5 \pm 5.9$ | $64.5 \pm 5.9$ |
|  | CWTM | $84.9 \pm 0.9$ | $81.1 \pm 2.8$ | $83.5 \pm 2.6$ | $81.0 \pm 3.7$ | $77.9 \pm 3.2$ | $65.8 \pm 5.7$ | $65.8 \pm 5.7$ |
|  | DeepSet-TM | $\mathbf{85.9 \pm 0.4}$ | $\mathbf{82.9 \pm 1.0}$ | $83.5 \pm 0.9$ | $\mathbf{82.1 \pm 1.2}$ | $\mathbf{81.0 \pm 1.7}$ | $\mathbf{83.0 \pm 1.8}$ | $81.0 \pm 1.7$ |

### G.2.2 VARYING THE HETEROGENEITY $\alpha$

Tables 6 and 7 show the accuracy of DeepSet-TM against static aggregation schemes on the three datasets of interest for $\alpha = 0.3$ and $0.8$ respectively. This is similar to Table 2, which showed the same view for $\alpha = 0.5$.

### G.2.3 LETTING DIFFERENT ADVERSARIES PERFORM DIFFERENT ATTACKS

While the Byzantine machine learning literature generally considers adversaries that coordinate their attacks (Alistarh et al., 2018; Chen et al., 2017; Farhadkhani et al., 2022), it remains relevant to consider settings in which attacks are not the same.

Table 8 shows the accuracy of DeepSet-TM and other static aggregations on the three datasets under a "Mixed" attack, where each adversary performs a different attack (among Logit-flipping, LMA, CPA and SIA-wb). Across CIFAR-10, CIFAR-100 and AG-News, DeepSet-TM still outperforms the baselines under this new attack (+4,1, +1.8 and +2.8 points respectively, compared to the strongest baseline). These experiments show our method can generalize to heterogeneous adversaries and is not relying on a single shared attack pattern.

Table 8: Accuracy (%) of DeepSet-TM against static aggregations on the CIFAR-10, CIFAR-100 and AG-News datasets and under a "Mixed" attack (Logit-flipping with amplification 2, LMA, CPA and SIA-wb), with heterogeneity $\alpha = 0.5$, $n = 17$ clients and $f = 4$ adversaries.

|  | Aggregation | CF-10 | CF-100 | Ag-News |
|---|---|---|---|---|
|  | Mean | $59.4 \pm 3.0$ | $73.1 \pm 0.4$ | $81.0 \pm 2.6$ |
|  | CWMed | $54.6 \pm 3.7$ | $65.7 \pm 2.0$ | $78.0 \pm 5.1$ |
| Mixed attack | GM | $59.5 \pm 2.8$ | $71.7 \pm 0.8$ | $81.1 \pm 2.3$ |
|  | CWTM | $59.6 \pm 3.4$ | $72.3 \pm 0.8$ | $80.9 \pm 2.6$ |
|  | DeepSet-TM | $\mathbf{63.7 \pm 1.5}$ | $\mathbf{74.9 \pm 0.3}$ | $\mathbf{83.9 \pm 0.9}$ |

### G.3 GIVING MORE COMPUTE TO THE ADVERSARY

Table 9 shows the performance of DeepSet-TM and static aggregation schemes on AG-News and under PGD-cw attack, when the adversaries are given more compute than during training. Specifically, training of DeepSet-TM is conducted with 50 PGD iterations, and attackers are given up to 150 at inference time. Since we operate in probit space, we do not see a performance drop.

Table 9: Accuracy (%) of DeepSet-TM and static aggregations on AG-News ($\alpha = 0.5$, $n = 17$, $f = 4$) against PGD-cw attack with various number of iterations. Training was conducted with $S = 50$ iterations.

|  | Test time $S$ | 50 | 100 | 150 |
|---|---|---|---|---|
|  | Mean | $54.9 \pm 6.7$ | $54.9 \pm 6.7$ | $54.9 \pm 6.7$ |
|  | CWMed | $53.2 \pm 7.0$ | $53.1 \pm 6.4$ | $53.1 \pm 6.4$ |
| AG-News | GM | $52.6 \pm 7.3$ | $52.6 \pm 7.2$ | $52.6 \pm 7.2$ |
|  | CWTM | $55.3 \pm 7.5$ | $55.2 \pm 6.7$ | $55.2 \pm 6.7$ |
|  | DeepSet-TM | $83.2 \pm 1.4$ | $83.0 \pm 1.4$ | $83.0 \pm 1.4$ |

### G.4 EFFECTIVENESS OF ROBUSTNESS ELEMENTS

In this section, we assess the improvement brought about by the two robustness elements – CWTM and adversarial training. To evaluate this, we consider the worst case performance of the DeepSet aggregator across five attacks under different combinations of the robustness elements. The results are reported in Table 1 for $\alpha = 0.5$ and Table 10 for $\alpha = \{0.3, 0.8\}$. We specifically exclude PGD attack in computing the worst case as the adversarially trained model shows elevated performance by virtue of being trained on the same attack, rendering the comparison unfair to the non-adversarially trained cases. We recall that the CWTM operator is applied to the DeepSet model only at inference time and incurs no additional cost during training.

In Table 10, the lowest performance is achieved by the DeepSet model without any robust element, as expected. Interestingly, the improvements derived from enhancing DeepSet with either CWTM or adversarial training are comparable. For instance, on the CIFAR-10 dataset with $\alpha = 0.3$,

Table 10: Evaluation of robust elements in a setup with $n = 17$ clients and $f = 4$ adversaries. We report the worst-case test accuracy across 5 different attacks.

| DeepSet | CWTM | Adv. Tr. | $\alpha = 0.3$ CIFAR-10 | CIFAR-100 | $\alpha = 0.8$ CIFAR-10 | CIFAR-100 |
|---|---|---|---|---|---|---|
| ✓ | ✗ | ✗ | $32.1 \pm 2.6$ | $30.8 \pm 4.3$ | $51.9 \pm 1.3$ | $47.4 \pm 3.1$ |
| ✓ | ✓ | ✗ | $35.5 \pm 1.4$ | $55.0 \pm 0.8$ | $53.7 \pm 1.4$ | $67.0 \pm 0.7$ |
| ✓ | ✗ | ✓ | $35.7 \pm 2.8$ | $54.1 \pm 1.0$ | $\mathbf{55.2 \pm 2.8}$ | $65.1 \pm 0.8$ |
| ✓ | ✓ | ✓ | $\mathbf{43.0 \pm 2.2}$ | $\mathbf{56.7 \pm 1.0}$ | $55.0 \pm 2.2$ | $\mathbf{68.0 \pm 0.8}$ |

CWTM results in 35.5% test accuracy while adversarial training results in 35.7%. Similarly, on CIFAR-100 with $\alpha = 0.3$, they achieve 55.5% and 54.1% respectively, starting from 30.8% when neither is applied. However, the highest performance is achieved with both elements combined, resulting in 43.0% on the CIFAR-10 and 56.7% on CIFAR-100 in the above case. Our results for in Table 1 follow a similar trend. Thus robustifying DeepSet with both CWTM and adversarial training brings the best from both worlds – Byzantine ML and adversarial ML.

## G.5 PERFORMANCE OF OTHER ADVERSARIAL DEFENSES

Table 11: Accuracy (%) of DeepSet-TM against randomized ablation on the CIFAR-100 dataset, with heterogeneity $\alpha = 0.3$, $n = 17$ clients and $f = 4$ adversaries. Logit flipping uses an amplification factor of 2 and RA-CWTM trims 3 clients on each side. RA outputs are computed over 100 iterations.

| Aggregation | Logit flipping | SIA-bb | LMA | CPA | SIA-wb | Worst case |
|---|---|---|---|---|---|---|
| Mean | $75.6 \pm 0.7$ | $69.1 \pm 0.8$ | $57.0 \pm 2.4$ | $56.8 \pm 2.4$ | $44.5 \pm 1.4$ | $44.5 \pm 1.4$ |
| RA-Mean | $\mathbf{75.7 \pm 0.7}$ | $69.0 \pm 0.8$ | $55.8 \pm 2.4$ | $56.5 \pm 2.4$ | $49.1 \pm 1.2$ | $49.1 \pm 1.2$ |
| CWTM | $69.1 \pm 1.1$ | $65.8 \pm 1.2$ | $69.1 \pm 1.2$ | $69.1 \pm 1.2$ | $49.6 \pm 1.1$ | $49.6 \pm 1.1$ |
| RA-CWTM | $69.8 \pm 1.1$ | $68.2 \pm 0.9$ | $71.2 \pm 0.7$ | $71.5 \pm 0.7$ | $49.9 \pm 1.2$ | $49.9 \pm 1.2$ |
| CWMed | $55.1 \pm 2.1$ | $51.8 \pm 2.6$ | $55.9 \pm 2.2$ | $55.8 \pm 2.2$ | $50.7 \pm 1.8$ | $50.7 \pm 1.8$ |
| RA-CWMed | $54.7 \pm 1.9$ | $49.9 \pm 2.0$ | $53.4 \pm 2.0$ | $54.8 \pm 1.8$ | $53.9 \pm 1.7$ | $49.9 \pm 2.0$ |
| DS-RA | $69.2 \pm 1.8$ | $68.4 \pm 1.3$ | $59.5 \pm 3.9$ | $59.3 \pm 1.5$ | $54.2 \pm 1.8$ | $54.2 \pm 1.8$ |
| DeepSet-TM | $\mathbf{75.7 \pm 0.8}$ | $\mathbf{71.9 \pm 1.1}$ | $\mathbf{72.2 \pm 1.9}$ | $\mathbf{72.5 \pm 0.6}$ | $\mathbf{56.7 \pm 1.0}$ | $\mathbf{56.7 \pm 1.0}$ |

We also evaluate our approach against Randomized ablation (RA) (Levine & Feizi, 2020), a certified adversarial defense method designed to improve robustness through repeated random subsampling of clients. Specifically, at each round, RA discards $f$ randomly chosen clients and aggregates the remaining ones (either via averaging (RA-Mean), CWTM (Ra-CWTM) or CWMed (RA-CWMed)) to get a candidate classification. This procedure is repeated several times with different random subsets, and the final prediction is obtained by majority voting over the outcomes. Intuitively, robustness arises from averaging across multiple independent aggregations, which reduces the influence of any single malicious client.

Table 11 reports the accuracy of RA baselines and their non-RA counterparts against DeepSet-TM, when $\alpha = 0.3$ and on CIFAR-100. We also report performance of the DeepSet-TM model without adversarial training but using RA instead under the name DS-RA. Notably, DeepSet-TM outperforms all baselines under all 5 attack settings. In general, RA yields varying degrees of improvements and is only consistently better across all attacks in the case of CWTM. On average, it improves accuracy by 0.5 points compared to the non-RA approach. This is inferior to DeepSet-TM, which yields an average improvement of 1.5 points over the strongest baseline for each attack. We can also note that DeepSet-TM exhibits lower variance across runs, confirming greater stability. In general, most certified adversarial defenses are highly dependent on the input space metric (usually the $\ell_p$ norm), hence are not directly applicable to our setting.

## G.6 PERFORMANCE AGAINST SOTA BASELINES

Figure 4 shows a comparative view of DeepSet-TM, COPUR and Manifold projection with $\alpha = 0.3$. The same results for $\alpha = 0.5$ are available in Figure 2.

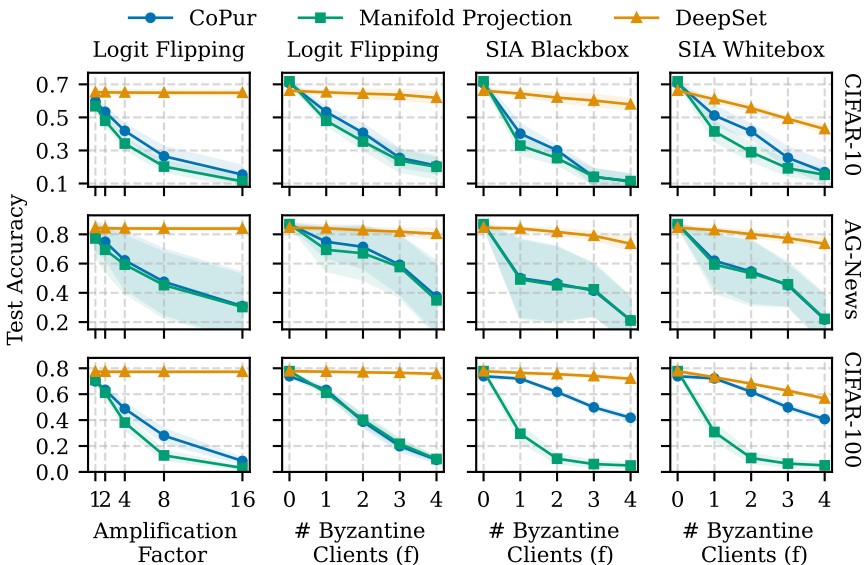

Figure 4: DeepSet-TM against baselines under heterogeneity $\alpha = 0.3$ and $n = 17$ clients. In the first column, we have $f = 1$ adversary. In all remaining columns, we use an amplification factor of 2 for the attacks.

## G.7 CLEAN ACCURACY OF DIFFERENT METHODS

Tables 12 and 13 report the clean accuracy for all aggregations *i.e.*, accuracy in the absence of adversaries.

Table 12: Accuracy (%) of robust aggregations with no adversaries ($f = 0$) and $n = 17$. DeepSet-TM achieves the highest accuracy in most scenarios.

| Heterogeneity | Dataset | Mean | CWTM | GM | CWMed | DeepSet-TM |
|---|---|---|---|---|---|---|
| $\alpha = 0.3$ | CIFAR-10 | $63.6 \pm 2.6$ | $61.4 \pm 3.0$ | $61.5 \pm 2.8$ | $54.9 \pm 2.6$ | $\mathbf{66.2 \pm 0.8}$ |
| | CIFAR-100 | $\mathbf{77.9 \pm 0.7}$ | $74.0 \pm 0.8$ | $73.7 \pm 0.4$ | $65.0 \pm 1.6$ | $77.7 \pm 0.7$ |
| | AG-News | $82.8 \pm 2.4$ | $82.7 \pm 2.4$ | $82.2 \pm 2.6$ | $82.1 \pm 2.5$ | $\mathbf{84.5 \pm 1.1}$ |
| $\alpha = 0.5$ | CIFAR-10 | $69.3 \pm 1.0$ | $68.4 \pm 1.1$ | $68.2 \pm 1.2$ | $64.3 \pm 2.5$ | $\mathbf{70.5 \pm 1.1}$ |
| | CIFAR-100 | $\mathbf{80.0 \pm 0.4}$ | $77.8 \pm 0.7$ | $77.0 \pm 0.7$ | $73.6 \pm 0.4$ | $79.6 \pm 0.4$ |
| | AG-News | $84.7 \pm 1.2$ | $84.7 \pm 1.2$ | $84.4 \pm 1.4$ | $84.5 \pm 1.4$ | $\mathbf{86.2 \pm 0.5}$ |
| $\alpha = 0.8$ | CIFAR-10 | $70.8 \pm 0.6$ | $70.2 \pm 0.8$ | $69.9 \pm 0.7$ | $67.8 \pm 1.7$ | $\mathbf{72.0 \pm 0.7}$ |
| | CIFAR-100 | $\mathbf{81.4 \pm 0.4}$ | $80.1 \pm 0.6$ | $79.7 \pm 0.5$ | $77.6 \pm 0.8$ | $80.8 \pm 0.4$ |
| | AG-News | $85.6 \pm 0.9$ | $85.6 \pm 1.0$ | $85.3 \pm 1.0$ | $85.3 \pm 0.9$ | $\mathbf{86.3 \pm 0.5}$ |

Table 13: Accuracy (%) of baselines with no adversaries ($f = 0$) and $n = 17$. Manifold Projection, the least robust baseline, has the highest accuracy in most scenarios. Nevertheless, the gap between the three algorithms is small in several cases.

| Heterogeneity | Dataset | CoPur | Manifold Projection | DeepSet-TM |
|---|---|---|---|---|
| $\alpha = 0.3$ | CIFAR-10 | $70.7 \pm 1.4$ | $\mathbf{71.8 \pm 0.8}$ | $66.2 \pm 0.8$ |
| | CIFAR-100 | $73.9 \pm 1.3$ | $\mathbf{77.8 \pm 0.4}$ | $77.7 \pm 0.7$ |
| | AG-News | $86.9 \pm 0.5$ | $\mathbf{87.0 \pm 0.5}$ | $84.5 \pm 1.1$ |
| $\alpha = 0.5$ | CIFAR-10 | $73.0 \pm 1.2$ | $\mathbf{74.1 \pm 0.8}$ | $70.5 \pm 1.1$ |
| | CIFAR-100 | $75.3 \pm 1.1$ | $77.8 \pm 0.3$ | $\mathbf{79.6 \pm 0.4}$ |
| | AG-News | $87.7 \pm 0.2$ | $\mathbf{87.9 \pm 0.3}$ | $86.2 \pm 0.5$ |

### G.8 COMPUTATIONAL EFFICIENCY OF DEEPSET-TM

#### G.8.1 INFERENCE LATENCY

Table 14 shows the inference latency for each static aggregation and for the two DeepSet variants (with and without CWTM). It is reported in milliseconds on a batch of 64 samples, and averaged across the whole test set. We believe the additional cost is very much acceptable in practice.

Table 14: Inference latency, in ms, for static aggregations, the standard DeepSet and DeepSet-TM on the CIFAR-10, CIFAR-100 and AG-News datasets. It is computed on a batch of 64 samples and averaged across the test set.

| Dataset | Mean | CWMed | CWTM | GM | DeepSet | DeepSet-TM |
|---|---|---|---|---|---|---|
| **CIFAR-10** | $0.06 \pm 0.01$ | $0.11 \pm 0.04$ | $0.14 \pm 0.03$ | $3.0 \pm 0.09$ | $0.26 \pm 0.05$ | $0.31 \pm 0.03$ |
| **CIFAR-100** | $0.05 \pm 0.01$ | $0.11 \pm 0.01$ | $0.13 \pm 0.03$ | $3.2 \pm 0.08$ | $0.25 \pm 0.03$ | $0.32 \pm 0.02$ |
| **AG-News** | $0.05 \pm 0.01$ | $0.10 \pm 0.01$ | $0.13 \pm 0.02$ | $3.0 \pm 0.07$ | $0.22 \pm 0.01$ | $0.30 \pm 0.02$ |

#### G.8.2 ADVERSARIAL TRAINING COST

We also compare the wall-clock time needed to train our DeepSet model with and without adversarial training in the configuration from Table 2, averaged over 5 runs. Without adversarial training, DeepSet takes about 1min, 1min40 and 3min to train on CIFAR-10, CIFAR-100 and AG-News respectively. When adding adversarial training ($S = 50$ inner steps and $N = 300$ adversary sets), this becomes 1h26m, 1h29m and 1h40m. Our work is specifically intended for cross-silo settings, and we thus believe that such a one-time training cost is acceptable in real applications.

## H LLM USAGE STATEMENT

We acknowledge the use of LLMs in this work, limited to coding assistance, identifying potentially relevant related work, and improving the clarity and grammar of the manuscript. All LLM-generated content was reviewed and verified by the authors.

