# OpenReview forum: "Robust Federated Inference"
_ICLR.cc/2026/Conference — ICLR 2026 Poster_

### Official Review · Reviewer_Dfp1 · 2025-10-26

**Soundness:** 2
**Presentation:** 3
**Contribution:** 3
**Rating:** 4
**Confidence:** 4

**Summary:**

This paper investigates the adversarial robustness of federated inference in the presence of corrupted clients. It demonstrates that robust averaging may be insufficient due to the non-continuity of the argmax operator. It also introduced a DeepSet-TM approach for robust non-linear aggregation by formulating the robust inference as an adversarial training problem. Experimental results demonstrate the superior robustness of the proposed DeepSet-TM framework over baselines in various attacking scenarios.

**Strengths:**

**Originality:** This work highlights the impact of the point-wise model dissimilarity and the probit’s margin on the robustness gap using linear averaging aggregation for federated inference. It also introduced a novel DeepSet-based framework to learn non-linear data-dependent robust aggregation function.

**Quality:** The provided counter example shows the prediction error induced by the non-continuity of the argmax operator. Theorem 1 theoretically analyzes the robustness gap of linear averaging aggregation for federated inference, followed by empirical validation in Figure 3. The robustness of the DeepSet-TM framework is also validated in the experiments using various attacks.

**Clarity:** The paper is well-written. The motivation behind robust federated inference is clearly discussed. The proofs of lemma and theorem are provided.

**Significance"** The problem of federated inference under potentially corrupted clients is highly significant in the context of real-world federated learning systems. A robust federated inference framework can largely improve the trustworthiness of the deployed systems.

**Weaknesses:**

(1) One major concern is the certification for DeepSet-TM under adversarial corruptions. Section 3 considers the linear averaging aggregation strategies and derives the theoretical result (Theorem 1) in shaping the robustness gap. Specifically, it highlights the impact of the corruption fraction, the model dissimilarity, and the probit’s margin. However, it is unclear how this robustness gap can be characterized in non-linear cases using DeepSet-TM. The highlighted potential factors, such as the point-wise model dissimilarity and the probit’s margin, are not explored in the non-linear aggregation of DeepSet-TM.

(2) Another concern is the computational efficiency of the proposed DeepSet-TM method in adversarial training (i.e., bi-level optimization problem). Though several simplification strategies are proposed, the running efficiency of DeepSet-TM is not well discussed compared to baselines.

(3) The impact of the simplification strategies for optimizing the robust DeepSet aggregator can be further discussed. With these simplification strategies (as well as ROBAVG operator in Eq. (10)), how will the derived solutions still approximate the optimal solution of the optimization problem in Eq. (9)?

**Questions:**

(1) Why is the assumption in line 241 required, i.e., if $[z]\_k = [z]\_{k'}$ for all $k,k'$, then $MARGIN(z) = \infty$?

(2) Lines 321-322 show that robust averaging is only incorporated at the inference time. Will the added ROBAVG operator change the optimality of the robust DeepSet aggregator defined in Eq. (8)? Why does it reduce the overall sensitivity of $\phi\_{\theta^*}$?

(3) Why does the proposed DeepSet-TM approach consistently achieve inferior robustness under Loss Maximization Attack (LMA) in Table 2?

(4) DeepSet-TM replaces the point-wise loss function from the indicator function with the cross-entropy loss. Will it better handle the counter examples in lines 222-225?

(5) How is the function $\bar{h}(x)$ calculated in the counter example?

---

> ### Author Response · Authors · 2025-11-20
> **Response to Reviewer Dfp1 (Part 1/3)**
>
> **We truly appreciate the reviewer’s constructive comments and would like to address the raised points below.**
>
> > W1] One major concern is the certification for DeepSet-TM under adversarial corruptions. Section 3 considers the linear averaging aggregation strategies and derives the theoretical result (Theorem 1) in shaping the robustness gap. Specifically, it highlights the impact of the corruption fraction, the model dissimilarity, and the probit’s margin. However, it is unclear how this robustness gap can be characterized in non-linear cases using DeepSet-TM. The highlighted potential factors, such as the point-wise model dissimilarity and the probit’s margin, are not explored in the non-linear aggregation of DeepSet-TM.
>
> **Response:** Thank you for giving us the opportunity to present the extension of Theorem 1 to Robust DeepSet-TM. We have added the following result in the revised paper, formally showing the impact of point-wise model dissimilarity and the probit’s margin on the robustness of the non-linear aggregation: DeepSet-TM.
>
> **Robustness gap of robust DeepSet aggregator.** In the theorem below, we present an upper bound on the robustness gap of robust DeepSet aggregator (10), extending Theorem 1 to nonlinear oracular aggregation. In what follows, we make some Lipschitz continuity assumptions.
>
> Specifically, there exists real values $L_{\rho}, L_{\mu}$ such that, for all $z, z' \in \Delta^K$, $v, v' \in \mathbb{R}^p$ and $k \in [K]$, $\\left\\lvert[\\rho_{\\theta^\ast_1}(z)]\_k - [\\rho_{\\theta^\ast_1}(z')]\_k \\right\\rvert  \\leq L_{\\rho} ||z - z'||\_2$ and $\\left\\lvert[\\mu_{\\theta^\ast_2}(v)]\_k - [\\mu_{\\theta^\ast_1}(v')]\_k\\right\\rvert  \\leq L_{\\mu} ||v - v'||\_2$.
>
> Let $\mathbf{h}(x) := \left(h_1(x), \ldots, h_n(x) \right)$. Recall that $\sigma_x^2 := \max_{k \in [K]} \frac{1}{n} \sum_{i = 1}^n \left( \left[ h_i(x) \right]_k - \left[ \overline{h}(x) \right]_k  \right)^2$.
>
> **Theorem 2.** Consider $\psi_{\text{rob}}$ as defined in (10) with $\text{ROBAVG} = \text{CWTM}$. If the regressors $h_1, \ldots, h_n$ are such that $\phi_{\theta^*}\left( \mathbf{h}(x) \right)$ has a unique maximum coordinate almost everywhere, then the following holds:
>
> $\\mathbb{E}\_{(x, y) \\sim \\mathcal{D}} \\left[\\ell^{\\text{adv}\_{\\psi_{\\text{rob}}}(x, \\hat{y}\_{\\text{o}} )} \\right] \leq \mathbb{P}\_{(x, y) \sim \mathcal{D}} \left[ \text{Margin}\left( \phi_{\theta^\ast}\left( \mathbf{h}(x) \right) \right) < 2\sqrt{2} L_\mu L_\rho  \left( \sqrt{\frac{\kappa n}{n - f}} + \sqrt{\frac{f}{n - f}}\right)\min\left\\{1, \sqrt{K} \sigma_x\right\\}  \right]$
>
> where $\hat{y}\_{\text{o}} = \arg\max_{k \in [K]} \left[ \phi_{\theta^\ast}\left( \mathbf{h}(x) \right) \right]_k$ and $\kappa = \frac{6 f}{n-2f}\, \left( 1 + \frac{f}{n-2f} \right)$.
>
> ----
>
> > W2] Another concern is the computational efficiency of the proposed DeepSet-TM method in adversarial training (i.e., bi-level optimization problem). Though several simplification strategies are proposed, the running efficiency of DeepSet-TM is not well discussed compared to baselines.
>
> **Response:** Thank you for raising this point. We agree that this analysis is crucial and will include it in a revised version of the paper.
>
> A. **Running efficiency.** We show in the table below the inference latency for each static aggregation and for the two DeepSet variants (with and without CWTM). It is reported in milliseconds on a batch of 64 samples, and averaged across the whole test set.
>
> | Dataset       | CWMed | Avg | CWTM | GM | DeepSet | DeepSet-TM |
> |--------------|------------|---------|-------|-----|-------|---------|
> | **CIFAR-10**  | 0.11±0.04 | 0.06±0.01 | 0.14±0.03| 3.0±0.09 | 0.26±0.05  | 0.31±0.03 |
> | **CIFAR-100** | 0.11±0.01 | 0.05±0.01 | 0.13±0.03 | 3.2±0.08 | 0.25±0.03 | 0.32±0.02 |
> | **AG-News**   | 0.10±0.01 | 0.05±0.01 | 0.13±0.02 | 3.0±0.07 | 0.22±0.01 | 0.30±0.02  |
>
> We believe the additional cost is very much acceptable in practice.
>
> B. **Adversarial training cost.** We also compare the wall-clock time needed to train our DeepSet model with and without adversarial training in the configuration from Table 2 ($n=17$, $f=4$), averaged across 5 runs.
> Without adversarial training, DeepSet takes about 1min, 1min40 and 3min to train on CIFAR-10, CIFAR-100 and AG-News respectively. When adding adversarial training ($S=50$ inner steps and $N=300$ adversary sets), this becomes 1h26m, 1h29m and 1h40m.
> Our work is specifically intented for cross-silo settings, and we thus believe such one-time training cost is acceptable in real applications.

---

> > ### Author Response · Authors · 2025-11-20
> > **Response to Reviewer Dfp1 (Part 2/3)**
> >
> > > W3] The impact of the simplification strategies for optimizing the robust DeepSet aggregator can be further discussed. With these simplification strategies (as well as ROBAVG operator in Eq. (10)), how will the derived solutions still approximate the optimal solution of the optimization problem in Eq. (9)?
> >
> > **Response:** Thank you for this question. The simplification strategy for solving for optimization problem (9) is inspired from the exisiting literature on adversarial training (Madry et al., 2018). First, we apply the standard approximation used in the training of classification models. Specifically, we replace the non-differentiable indicator loss with a surrogate differentiable cross-entropy loss $\ell(\phi_{\theta} \left( \mathbf{z} \right) , y)$. Then, we solve the resulting minimax optimization problem approximately by alternating between gradient ascent and gradient descent, a strategy that is commonly employed in adversarial training (Madry et al., 2018). The suboptimality analysis of this algorithm is highly non-trivial and remains an active area of research in the field of adversarial learning.
> >
> > Regarding the robustness certificate, we have now provided a rigorous analysis on the *robustness gap* (i.e., the bound on the additional suboptimality induced by adversarial probits) in Theorem 2. We have also discussed formally the benefits of using ROBAVG in Appendix G.
> >
> > ----
> >
> > > Q1] Why is the assumption in the line 241 required i.e., if $[z]\_k = [z]_{k'}$ for all $k, k'$, then $MARGIN(z) = \infty$?
> >
> > **Response:** We make this assumption to handle the degenerate case when each class $k$ has equal probability in the oracular aggregation, in which case any output is a valid output and the robustness gap is trivially zero. For the theoretical guarantee, shown in Theorem 1, we assume that the $\arg\max_k [\overline{h}(x)]_k$ is unique almost everywhere.
> >
> > > Q2] Lines 321-322 show that robust averaging is only incorporated at the inference time. Will the added ROBAVG operator change the optimality of the robust DeepSet aggregator defined in Eq. (8)? Why does it reduce the overall sensitivity of $\phi_{\theta^*}$?
> >
> > **Response:** That is very good point. Thank you for giving us the opportunity to explain this further. We have now included a formal reasoning on the benefits of using ROBAVG in Appendix G of the revised paper. We summarize the reasoning below.
> >
> > **Reduced Sensitivity by ROBAVG.** The robustness gap for the adversarially trained DeepSet aggregator without robust averaging depends upon the degree of corruption $\delta$ (shown in Appendix G). However, the robustness gap is rendered independent of this degree of corruption when we use a robust averaging scheme like CWTM, as in eq. (10). In fact, when the variance across honest probits is small, the robustness gap under robust averaging is much smaller compared to the case without robust averaging. For instance, consider the special case when $h_i(x) = h_j(x)$ for all $i, j \in [n]$ and all $x$. While the robustness gap when using CWTM is provably $0$, as per Theorem 2 (added to the revised paper), that need not be the case without robust averaging, see eq. (22) in Appendix G.
> >
> > > Q3] Why does the proposed DeepSet-TM approach consistently achieve inferior robustness under Loss Maximization Attack (LMA) in Table 2?
> >
> > **Response:** This is indeed an interesting point. While under LMA, DeepSet-TM does not seem to achieve the highest accuracy for all datasets, the overall accuracy remains within the uncertainty ranges of the best static aggregator. Moreover, it is not consistently worse that any given aggregator and performs best on CIFAR-100.
> >
> > In LMA, the adversary sets the probit for the least likely class to 1 and the rest to 0. This produces *extreme outliers* that static aggregators like GM are *specifically designed to filter*. By contrast, simply setting 1 to the second most likely class instead of the least likely (which is what both SIA variants do) significantly reduces their robustness, since adversaries become much closer to the honest outputs. And in this case, DeepSet-TM consistently outperforms static aggregations, across all our experiment settings.
> >
> > Our experiments show that DeepSet-TM outperforms all static aggregations in general, and specifically for attacks that mimic honest ones, while maintaining best or close to best accuracy on LMA (or the similar CPA).

---

> > > ### Author Response · Authors · 2025-11-20
> > > **Response to Reviewer Dfp1 (Part 3/3)**
> > >
> > > > Q4] DeepSet-TM replaces the point-wise loss function from the indicator function with the cross-entropy loss. Will it better handle the counter examples in lines 222-225?
> > >
> > > **Response:** We would like to stress that the counterexample illustrates an intrinsic issue: the argmax decision is discontinuous, so arbitrarily small $\ell_2$ perturbations can flip the class. Replacing the indicator with cross-entropy is a standard differentiable surrogate that enables training, but it does not remove the argmax discontinuity.
> > >
> > > Instead, we leveraged the counterexample to guide the design of our solution: our approach uses adversarial training to **increase the margin** and CWTM to reduce the sensitivity of aggregated probits to probit corruption, and thus reduce the probability of small perturbation flipping the argmax. This is now formally characterized in Theorem 2 and discussed in details in Appendix G.
> > >
> > > > Q5] How is the function $\bar{h}(x)$ calculated in the counter example?
> > >
> > > **Response:** Thank you for pointing that out!
> > > $\bar{h}(x)$ is supposed to be the average, but we indeed made a typo. In the counterexample, $h_3(x)$ should be $(1/2, 1/2 - 3\epsilon, 3\epsilon)$, for any $\epsilon \in (0, 1/6)$.

---

> > > > ### Comment · Reviewer_Dfp1 · 2025-11-25
> > > >
> > > > Thanks for the rebuttal. Here are some follow-up questions.
> > > >
> > > > (1) In the proof of Theorem 2 in Appendix F, (i) why does the result $\frac{L^2_{\rho}}{n} \sum_{i,l=1}^n ( [h_i(x)]_j - [h_l(x)]_j )^2 = 2 L^2\_{\rho} \sum\_{i=1}^n ( [h_i(x)]_j - [\bar{h}(x)]_j )^2$ hold (lines 1373-1377)?, and (ii) how can the robustness property of CWTM derive Eq. (19)?
> > > >
> > > > (2) Thanks for discussing the benefits of ROBAVG in reduced sensitivity. However, I still have concerns regarding how it will affect the optimality of Eq. (9). ROBAVG is **ONLY** used at inference time. It is unclear how this newly added strategy affects the model optimization. Generally, model training uses the standard average, while model inference uses ROBAVG. This might imply that training and inference stages have different learning functions.
> > > >
> > > > (3) Regarding the results in Table 2, does it mean that DeepSet-TM may be sensitive to adversarial attacks with extreme outliers like LMA, even when ROBAVG is used during inference?

---

> ### Author Response · Authors · 2025-11-26
> **Response to Reviewer Dfp1**
>
> **We thank the reviewer once again for their valuable comments and hope to answer their remaining questions below.**
>
> >(1) In the proof of Theorem 2 in Appendix F, (i) why does the result hold (lines 1373-1377)?, and (ii) how can the robustness property of CWTM derive Eq. (19)?
>
> (i) This equality comes from a standard arithmetic property on the definition of the empirical variance. We detail the steps below and will make this point clearer in the paper.
>
> Let $S_1:=\sum_{i,l=1}^n \Big([h_i(x)]\_j-[h_l(x)]\_j\Big)^2$ and $S_2:=\sum_{i=1}^n\Big([h_i(x)]\_j-[\overline{h}(x)]\_j\Big)^2$.
>
> On the one hand:
> $$S_1=n\sum_{i=1}^n [h_i(x)]\_j^2 + n\sum_{l=1}^n [h_l(x)]\_j^2-2\left(\sum_{i=1}^n[h_i(x)]\_j\right)\left(\sum_{l=1}^n[h_l(x)]\_j\right)= 2n\sum_{i=1}^n [h_i(x)]\_j^2 - 2 \left(\sum_{i=1}^n[h_i(x)]\_j\right)^2.
> $$
> On the other:
> $$S_2 = \sum_{i=1}^n [h_i(x)]\_j^2 + n[\overline{h}(x)]\_j^2 - 2[\overline{h}(x)]\_j\sum_{i=1}^n[h_i(x)]\_j = \sum_{i=1}^n [h_i(x)]\_j^2 + \frac{1}{n}\left(\sum_{i=1}^n[h_i(x)]\_j\right)^2 - \frac{2}{n}\left(\sum_{i=1}^n[h_i(x)]\_j\right)^2$$
> which indeed gives $2nS_2 = 2n\sum_{i=1}^n [h_i(x)]\_j^2 - 2\Big(\sum_{i=1}^n[h_i(x)]\_j\Big)^2 = S_1$.
>
>
> (ii) In addition to the robustness property of CWTM, we make use of Lemma 2 (just like in Appendix B.3). However, there is a small typo with a missing $\frac{1}{n}$ in (19) and in line 181 (which does not affect the correctness of the proof as it gets corrected afterwards, we updated the document). Below we detail the steps you need to apply starting from line 1360.
>
> Let $H$ be a subset of honest probits of cardinal $n-f$ (where $h_i(x) = z_i$).
> Fix a coordinate $j$ and define, for brevity, $[\rho_i]\_j:= \big[\rho_{\theta^\ast_1}(h_i(x))\big]\_j$ and $[\tilde\rho_i]\_j := \big[\rho_{\theta^\ast_1}(z_i)\big]\_j$, as well as $[\overline{\rho}\_H(x)]\_j := \frac{1}{n-f}\sum_{i\in H}[\rho_i]\_j = \frac{1}{n-f}\sum_{i\in H}[\tilde\rho_i]\_j$ and $[\overline{\rho}(x)]\_j := \frac{1}{n}\sum_{i=1}^n[\rho_i]\_j$.
> Let $\mathrm{Rob}_j := [\mathrm{ROBAVG}(\tilde\rho_1,\dots,\tilde\rho_n)]_j$.
>
> Then:
>
> $$
> |\mathrm{Rob}\_j - [\overline{\rho}(x)]\_j|
> \le |\mathrm{Rob}\_j - [\overline{\rho}\_H(x)]\_j| + |[\overline{\rho}\_H(x)]\_j - [\overline{\rho}(x)]\_j|
> \le \sqrt{\frac{\kappa}{n-f}\sum_{i\in H}\big([\rho_i]\_j - [\overline{\rho}\_H(x)]\_j\big)^2} + \sqrt{\frac{f}{n-f}\cdot\frac{1}{n}\sum_{i=1}^n\big([\rho_i]\_j - [\overline{\rho}(x)]\_j\big)^2}
> $$
>
> where the first term is bounded using the definition of robust averaging and the second term as in Appendix B.3.
>
> Applying Lemma 2 to the first term gives:
> $$\sqrt{\frac{\kappa}{n-f}\sum_{i\in H}\big([\rho_i]\_j - [\overline{\rho}\_H(x)]\_j\big)^2} \le \sqrt{\frac{\kappa n}{n-f}}\sqrt{\frac{1}{n}\sum_{i=1}^n\big([\rho_i]\_j - [\overline{\rho}(x)]\_j\big)^2}$$
> hence the final result.
>
> ---
>
> >(2) Thanks for discussing the benefits of ROBAVG in reduced sensitivity. I still have concerns regarding how it will affect the optimality of Eq. (9).
>
> Thank you for pointing this out. This is indeed a limitation of our method, but we chose to remove CWTM during training for computational efficiency. We recognise this might constitute a change in the learning objective, but we opted to report these result in order to present the strongest defense and establish the state-of-the-art for future work to improve upon. We will add a discussion on this in the final version of the paper.
>
> However, it is worth noting that the REERM optimization problem for training DeepSet is a *surrogate objective* designed to minimize sensitivity to adversarial probit corruptions. The ultimate objective is reducing sensitivity (i.e. robustness gap with respect to oracular aggregation with uncorrupted probits), and not solving the REERM problem. What we have shown in Appendix G is that the use of CWTM/ROBAVG *reduces the sensitivity compared to that of the solution of the REERM problem*. This proves that **using CWTM is strictly better** when variance amongst uncorrupted probits is small.
>
> >(3) Does it mean that DeepSet-TM may be sensitive to adversarial attacks with extreme outliers like LMA, even when ROBAVG is used during inference?
>
> Results in Table 2 do not indicate that DeepSet-TM could be particularly more sensitive to attacks with extreme outliers than other defenses. In fact, it is always either stronger or statistically tied with the strongest defense against LMA. Our previous response was not meant to suggest that DeepSet is particularly sensitive to LMA, but rather to explain why some static aggregations may *occasionally* achieve unusually strong performance on attacks that create extreme outliers (such as LMA).
>
> Note however that what ultimately matters is overall (worst-case) robustness. Being occasionally strong on a single attack does not constitute a proof of robustness. DeepSet-TM offers *uniformly strong performance across the full suite of attacks*, whereas other static aggregations fail in most of the settings we tested.

---

### Official Review · Reviewer_h5tr · 2025-10-30

**Soundness:** 4
**Presentation:** 4
**Contribution:** 4
**Rating:** 8
**Confidence:** 4

**Summary:**

This paper systematically investigates a critical security issue in federated learning and edge computing: the robustness of federated inference. The authors point out that existing model ensemble and aggregation paradigms are highly vulnerable to simple attacks from malicious clients. They formally define the problem of "robust federated inference" for the first time, filling a gap in the field. Through an initial theoretical analysis of averaging-based aggregators, the paper reveals that the error of these methods depends not only on the proportion of malicious clients (corruption fraction) but also on the dissimilarity between honest client outputs and the prediction margin of the models for the two most probable classes. This analysis lays a solid theoretical foundation for subsequent method design.
To address the robustness problem in non-linear scenarios, the paper proposes a novel DeepSet-TM aggregator. This method ingeniously reframes the robust inference problem as an adversarial machine learning problem in the probit vector space and leverages the permutation invariance of the DeepSet architecture to significantly reduce the computational complexity of adversarial training. By introducing adversarial samples during the training phase and combining them with robust averaging (CWTM) mechanisms during the testing phase, DeepSet-TM ensures the aggregator's resilience to malicious inputs. The paper conducts rigorous empirical validation on multiple benchmarks, including image and text classification. The results demonstrate that DeepSet-TM achieves a significant improvement in worst-case accuracy, ranging from 4.7% to 22.2%, compared to existing techniques across various attacks (including the newly proposed SIA attack), strongly proving the practical utility and robustness of the proposed scheme.

**Strengths:**

This paper makes several substantial contributions in the field of robust federated inference, demonstrating high originality and significance. Firstly, the paper formally defines the critical problem of "robust federated inference" for the first time, clearly delineating the challenge of ensuring aggregator accuracy in the presence of malicious clients. This issue has long been overlooked in existing literature and possesses significant theoretical and practical value. Secondly, the paper conducts an in-depth theoretical analysis of averaging-based aggregators, revealing the complex relationship between their robustness, client output disparities, and prediction margins. This provides a solid theoretical foundation for understanding the limitations of existing methods and points the way towards more refined future designs. Most importantly, the proposed DeepSet-TM aggregator is highly original; it creatively reframes the robust inference problem as an adversarial machine learning problem and cleverly leverages the permutation invariance of the DeepSet model to address the computational complexity of adversarial training. This method, by combining adversarial training and robust averaging during the testing phase, effectively enhances robustness. The experimental section also demonstrates DeepSet-TM's superior performance across various datasets and attack types, with accuracy improvements ranging from 4.7% to 22.2%, thereby proving its strong potential and importance in practical applications.

**Weaknesses:**

Despite the paper's excellent performance in both conceptual and methodological aspects, there remain areas for improvement. Firstly, while the paper defines robust federated inference in the "Problem Formulation" section, some critical assumptions, such as those regarding the security of the client models themselves, could be elaborated upon in more detail. Currently, the paper primarily focuses on the robustness of the aggregation phase. However, if client models are susceptible to poisoning or backdoor attacks, this could impact the overall system robustness, a topic that receives relatively less discussion in the paper. Secondly, the computational cost and model complexity of the DeepSet-TM aggregator are potential weaknesses. Although the paper mentions leveraging permutation invariance to reduce the search space for adversaries, its actual overhead during training and inference (especially in large-scale federated deployments) still requires more detailed analysis and quantification compared to simpler averaging or median aggregators, such as providing end-to-end training time or inference latency comparisons for different aggregators. Furthermore, while the experimental results are impressive, it would be beneficial to further investigate DeepSet-TM's performance under varying degrees of federated heterogeneity (e.g., larger data distribution shifts among different clients) to comprehensively evaluate its generalization capabilities.

**Questions:**

1.The paper states that "the DeepSet-TM aggregator incorporates robust averaging (CWTM) at the inference time." Could you please elaborate on the specific mechanism of this "incorporation"? How are the outputs of DeepSet-TM combined with or selected alongside the results of CWTM during the testing phase, and how does this combination synergistically enhance robustness?
2.Regarding the computational efficiency of the DeepSet-TM aggregator, could you provide a more detailed analysis? Specifically, how much do the training time and inference latency of DeepSet-TM increase after introducing adversarial training and robust averaging at the testing phase, compared to a pure DeepSet or other non-linear aggregators? In large-scale federated deployments, is this increase acceptable in practice?
3.The paper highlights experiments on federated heterogeneity (controlled by the Dirichlet distribution parameter α), using α={0.3, 0.5, 0.8}. Could you further investigate DeepSet-TM's performance under higher heterogeneity (e.g., smaller α values, or other means of introducing larger data distribution shifts)? Does the design of DeepSet-TM have inherent advantages or limitations when dealing with extreme data heterogeneity?

---

> ### Author Response · Authors · 2025-11-20
> **Response to Reviewer h5tr (Part 1/2)**
>
> **We appreciate the reviewer’s encouraging feedback and clarify the remaining questions below.**
>
> > W1] ... some critical assumptions, such as those regarding the security of the client models themselves, could be elaborated upon in more detail. Currently, the paper primarily focuses on the robustness of the aggregation phase. However, if client models are susceptible to poisoning or backdoor attacks, this could impact the overall system robustness, a topic that receives relatively less discussion in the paper.
>
> **Response:** Our threat model allows *for arbitrary outputs* of up to $f$ of the $n$ clients. This encompasses outputs resulting from poisoning or backdoor attacks. Thus, the proposed aggregator remains robust as long as such compromised clients fall within the tolerated $f$. However, indeed, if any of the *remaining honest* clients themselves are also compromised, additional mechanisms would be required to detect or mitigate these cases specifically. This scenario is currently outside the scope of the present study, however, but we will explicitly clarify this point in the final version.
>
>
> ----
>
> > Q1] The paper states that "the DeepSet-TM aggregator incorporates robust averaging (CWTM) at the inference time." Could you please elaborate on the specific mechanism of this "incorporation"? How are the outputs of DeepSet-TM combined with or selected alongside the results of CWTM during the testing phase, and how does this combination synergistically enhance robustness?
>
> **Response:** Yes, the DeepSet architecture has the following functional form (eq. 8):
>
> $$\phi_{\theta}(\mathbf{z}) = \mu_{\theta_2} \left(\frac{1}{n}\sum_{i=1}^n \rho_{\theta_1}(z_i)\right) $$
>
> where $\mu$ and $\rho$ are two neural networks parameterized by $\theta_2$ and $\theta_1$ respectively.
>
> We incorporate robust averaging by replacing the summation operation in the above with CWTM as follows:
>
> $$\phi_{\theta}(\mathbf{z}) = \mu_{\theta_2} \Big( \text{CWTM}(\rho_{\theta_1}(z_1), \rho_{\theta_1}(z_2), \ldots, \rho_{\theta_1}(z_n))\Big).$$
>
> This is an inference-only replacement (no extra training cost), and enhances robustness by reducing the sensitivity of this intermediate aggregation operation to outliers (see Appendix E.4 where we ablate the contribution of CWTM). Note also that we now provide an additional theoretical analysis on the usefulness of CWTM added to DeepSet in Appendix G of the revised version of the paper.
>
> ----
>
> > W2 and Q2] Regarding the computational efficiency of the DeepSet-TM aggregator, could you provide a more detailed analysis? Specifically, how much do the training time and inference latency of DeepSet-TM increase after introducing adversarial training and robust averaging at the testing phase, compared to a pure DeepSet or other non-linear aggregators? In large-scale federated deployments, is this increase acceptable in practice?
>
> **Response:** Thank you for mentioning this. We agree that this analysis is important and will include it in the final version of the paper.
>
> A. **Inference latency.** We show in the table below the inference latency for each static aggregation and for the two DeepSet variants (with and without CWTM). It is reported in milliseconds on a batch of 64 samples, and averaged across the whole test set.
>
> | Dataset       | CWMed | Avg | CWTM | GM | DeepSet | DeepSet-TM |
> |--------------|------------|---------|-------|-----|-------|---------|
> | **CIFAR-10**  | 0.11±0.04 | 0.06±0.01 | 0.14±0.03| 3.0±0.09 | 0.26±0.05  | 0.31±0.03 |
> | **CIFAR-100** | 0.11±0.01 | 0.05±0.01 | 0.13±0.03 | 3.2±0.08 | 0.25±0.03 | 0.32±0.02 |
> | **AG-News**   | 0.10±0.01 | 0.05±0.01 | 0.13±0.02 | 3.0±0.07 | 0.22±0.01 | 0.30±0.02  |
>
> We believe the additional cost is very much acceptable in practice.
>
> B. **Adversarial training cost.** We also compare the wall-clock time needed to train our DeepSet model with and without adversarial training in the configuration from Table 2 ($n=17$, $f=4$), averaged across 5 runs.
> Without adversarial training, DeepSet takes about 1min, 1min40 and 3min to train on CIFAR-10, CIFAR-100 and AG-News respectively. When adding adversarial training ($S=50$ inner steps and $N=300$ adversary sets), this becomes 1h26m, 1h29m and 1h40m.
> Our work is specifically intented for cross-silo settings, and we thus believe such one-time training cost is acceptable in real applications.

---

> > ### Author Response · Authors · 2025-11-20
> > **Response to Reviewer h5tr (Part 2/2)**
> >
> > > W3 and Q3] Could you further investigate DeepSet-TM's performance under higher heterogeneity (e.g., smaller α values, or other means of introducing larger data distribution shifts)? Does the design of DeepSet-TM have inherent advantages or limitations when dealing with extreme data heterogeneity?
> >
> > **Response:** Thank you for suggesting these experiments. We report below results in the configuration from Table 2 (i.e. $n=17$, $f=4$), but lowering $\alpha$ to $0.1$.
> > It is worth noting that this setting represents extreme heterogeneity. As an example, on CIFAR-10, several clients are trained on only 1 or 2 classes out of the 10 that exist. This makes the overall accuracy on this dataset go below 50%, even when averaging client responses without any attacker.
> >
> > Under these conditions, DeepSet-TM still outperforms all baselines (average absolute improvement of **+9.2 percentage points** against the strongest baseline for each attack). However, due to the extreme heterogeneity, many attacks make the system no better than random guessing.
> >
> > - CIFAR-10:
> >
> > | Aggregation  | Logit flipping | SIA-bb       | LMA          | CPA          | SIA          | PGD-cw       | Worst case           |
> > | :-----       | :-----------   | :----------- | :----------- | :----------- | :----------- | :----------- | :------------------- |
> > | Mean         | 39.8±3.7       | 38.8±3.3     | 18.6±4.8     | 13.6±3.6     | 6.4±2.7      | 1.6±1.8      | 1.6±1.8              |
> > | CWMed        | 22.3±2.8       | 23.9±3.1     | 19.0±3.4     | 13.6±3.0     | 13.5±2.4     | 1.5±2.1      | 1.5±2.1              |
> > | GM           | 35.8±2.5       | 37.8±2.8     | **24.9±5.9** | 15.6±4.7     | 11.6±3.6     | 1.2±1.5      | **1.2±1.5**          |
> > | CWTM         | 31.9±2.5       | 35.5±2.6     | 22.3±6.0     | 15.3±4.6     | 9.7±3.6      | 2.5±1.9      | 2.5±1.9              |
> > | DeepSet-TM   | **48.0±4.6**   | **42.2±4.5** | 22.4±8.1     | **21.0±8.8** | **30.0±4.0** | **37.1±5.6** | **21.0±8.8**         |
> >
> >
> > - CIFAR-100:
> >
> > | Aggregation  | Logit flipping | SIA-bb       | LMA          | CPA          | SIA          | PGD-cw       | Worst case           |
> > | :------------ | :------------- | :----------- | :----------- | :----------- | :----------- | :----------- | :------------------- |
> > | Mean         | 61.0±1.7       | 52.5±1.6     | 17.9±2.3     | 17.7±2.4     | 10.5±1.5     | 2.9±0.7      | 2.9±0.7              |
> > | CWMed        | 23.4±2.2       | 26.0±1.6     | 27.3±1.4     | 24.6±1.5     | 15.3±0.9     | 1.3±0.7      | **1.3±0.7**          |
> > | GM           | 52.1±1.4       | 46.2±1.5     | 36.2±1.7     | 35.3±1.8     | 14.3±1.8     | 2.0±0.6      | 2.0±0.6              |
> > | CWTM         | 40.5±1.7       | 38.4±1.6     | 40.8±1.9     | 39.7±1.9     | 18.4±1.8     | 9.1±1.3      | 9.1±1.3              |
> > | DeepSet-TM   | **62.2±1.9**   | **59.2±0.6** | **50.0±3.2** | **50.3±2.6** | **36.2±0.9** | **12.8±0.9** | **12.8±0.9**         |
> >
> > - AG-News:
> >
> > | Aggregation  | Logit flipping | SIA-bb       | LMA          | CPA          | SIA          | PGD-cw       | Worst case           |
> > | :------------ | :------------- | :----------- | :----------- | :----------- | :----------- | :----------- | :------------------- |
> > | Mean  | 56.1±12.3      | 42.3±16.0      | 25.6±10.6      | 27.4±13.6      | 16.7±9.2       | 4.4±5.6        | 4.4±5.6|
> > | CWMed  | 43.2±12.7      | 38.4±11.5      | 18.0±7.4       | 25.2±7.9       | 24.0±7.2       | 3.7±4.0        | **3.7±4.0**|
> > | GM  | 52.9±14.7      | 40.8±14.8      | **30.5±12.9**      | **30.9±13.7**      | 21.7±10.9      | 4.9±5.6        | 4.9±5.6|
> > | CWTM  | 54.4±11.8      | 42.3±15.9      | 25.6±11.5      | 27.6±13.9      | 16.1±8.4       | 4.6±5.6        | 4.6±5.6 |
> > | DeepSet-TM  | **61.5±14.0**      | **43.7±11.7**      | 14.6±6.0       | 22.2±9.9       | **47.5±6.6**       | **49.6±17.3**      | **14.6±6.0** |

---

> > > ### Comment · Reviewer_h5tr · 2025-11-27
> > >
> > > Thank you for the rebuttal. Your clarifications adequately address my previous concerns. I appreciate the improvements made and encourage the continued refinement of the manuscript.

---

### Official Review · Reviewer_6LL3 · 2025-11-01

**Soundness:** 3
**Presentation:** 3
**Contribution:** 3
**Rating:** 8
**Confidence:** 4

**Summary:**

This paper provides the first formal robustness analysis of federated inference, where a central server aggregates model predictions (probits) from multiple local clients to make a global decision. It shows that widely used averaging-based and trimmed-mean aggregators, though intuitively robust, can still fail even small deviations in logit (prob) space can flip the final argmax decision. The authors formally characterise the robustness gap and prove upper bounds showing that robustness depends on three key factors: the corruption fraction the inter-client dissimilarity of honest outputs, and the classification margin of the average prob vector. To go beyond linear aggregation, they cast robust federated inference as an adversarial learning problem over probability vectors, and propose a DeepSet-based neural aggregator that combines adversarial training with test-time robust averaging.

**Strengths:**

* The paper makes aa original theoretical contribution by providing the first formal robustness framework for federated inference, a setting previously treated mainly empirically.
* the theoretical strength is the counterexample construction that demonstrates the argmax discontinuity problem, showing that even infinitesimal perturbations in averaged probits can cause misclassification
* non-linear aggregation as an adversarial learning problem over probit vectors is conceptually novel. By recognizing that the adversarial space is structured (the probability simplex), the authors identify a tractable route to robustness.
* The proposed DeepSet-based robust aggregator then builds on this insight to combine permutation invariance, adversarial training, and inference-time robust averaging, achieving both theoretical justification and strong empirical performance.

**Weaknesses:**

* In REERM, the inner maximization is combinatorial and non-differentiable, requiring heuristic relaxations that undermine its minimax semantics.
* The adversarial model is overpowered and underspecified, leading to unrealistic robustness assumptions and potentially excessive conservatism.
* Moreover, the learned DeepSet aggregator, though effective empirically, lacks analytical interpretability or formal robustness guarantees. Thus, REERM advances the conceptual framing of robust federated inference but leaves open key challenges in tractable optimization, principled approximation, and theoretical certification.
* What about KRUM aggregation as the robust aggregator?

**Questions:**

Kindly refer to weaknesses

---

> ### Author Response · Authors · 2025-11-20
> **Response to Reviewer 6LL3 (Part 1/2)**
>
> **We thank the reviewer for the positive remarks and respond to their remaining concerns below.**
>
>
> > W1] In REERM, the inner maximization is combinatorial and non-differentiable, requiring heuristic relaxations that undermine its minimax semantics.
>
> **Response:** Thank you for this question. The simplification strategy for solving for optimization problem (9) is inspired from the exisiting literature on adversarial training (Madry et al., 2018). First, we apply the standard approximation used in the training of classification models. Specifically, we replace the non-differentiable indicator loss with a surrogate differentiable cross-entropy loss $\ell(\phi_{\theta} \left( \mathbf{z} \right) , y)$. Then, we solve the resulting minimax optimization problem approximately by alternating between gradient ascent and gradient descent, a strategy that is commonly employed in adversarial training (Madry et al., 2018). The suboptimality analysis of this algorithm is highly non-trivial and remains an active area of research in the field of adversarial learning.
>
> ----
>
> > W2] The adversarial model is overpowered and underspecified, leading to unrealistic robustness assumptions and potentially excessive conservatism.
>
> **Response:** Thank you for your remark, this is a valid concern. We have intentionally considered a strong adversarial model (formally specified in Section 2), wherein $f$ out of $n$ probits can be arbitrarily altered prior to aggregation, in order to develop a strong theoretical foundation for Robust Federated Inference. The results, both theoretical and empirical, presented in the current paper serve as a foundation for future work wherein one could consider weaker adversarial models.
>
> > W3] Moreover, the learned DeepSet aggregator, though effective empirically, lacks analytical interpretability or formal robustness guarantees. Thus, REERM advances the conceptual framing of robust federated inference but leaves open key challenges in tractable optimization, principled approximation, and theoretical certification.
>
> **Response:** Thank you for your remark, this was indeed a sensible concern. We have added the following result in the revised paper (Theorem 2), formally showing the impact of point-wise model dissimilarity and the probit’s margin on the robustness of the non-linear aggregation: DeepSet-TM. We have also included a formal reasoning on the benefits of using ROBAVG within DeepSet in Appendix G of the revised paper.
>
> **Robustness gap of robust DeepSet aggregator.** In the theorem below, we present an upper bound on the robustness gap of robust DeepSet aggregator (10), extending Theorem 1 to nonlinear oracular aggregation. In what follows, we make some Lipschitz continuity assumptions.
>
> Specifically, there exists real values $L_{\rho}, L_{\mu}$ such that, for all $z, z' \in \Delta^K$, $v, v' \in \mathbb{R}^p$ and $k \in [K]$, $\\left\\lvert[\\rho_{\\theta^\ast_1}(z)]\_k - [\\rho_{\\theta^\ast_1}(z')]\_k \\right\\rvert  \\leq L_{\\rho} ||z - z'||\_2$ and $\\left\\lvert[\\mu_{\\theta^\ast_2}(v)]\_k - [\\mu_{\\theta^\ast_1}(v')]\_k\\right\\rvert  \\leq L_{\\mu} ||v - v'||\_2$.
>
> Let $\mathbf{h}(x) := \left(h_1(x), \ldots, h_n(x) \right)$. Recall that $\sigma_x^2 := \max_{k \in [K]} \frac{1}{n} \sum_{i = 1}^n \left( \left[ h_i(x) \right]_k - \left[ \overline{h}(x) \right]_k  \right)^2$.
>
> **Theorem 2.** Consider $\psi_{\text{rob}}$ as defined in (10) with $\text{ROBAVG} = \text{CWTM}$. If the regressors $h_1, \ldots, h_n$ are such that $\phi_{\theta^*}\left( \mathbf{h}(x) \right)$ has a unique maximum coordinate almost everywhere, then the following holds:
>
> $\\mathbb{E}\_{(x, y) \\sim \\mathcal{D}} \\left[\\ell^{\\text{adv}\_{\\psi_{\\text{rob}}}(x, \\hat{y}\_{\\text{o}} )} \\right] \leq \mathbb{P}\_{(x, y) \sim \mathcal{D}} \left[ \text{Margin}\left( \phi_{\theta^\ast}\left( \mathbf{h}(x) \right) \right) < 2\sqrt{2} L_\mu L_\rho  \left( \sqrt{\frac{\kappa n}{n - f}} + \sqrt{\frac{f}{n - f}}\right)\min\left\\{1, \sqrt{K} \sigma_x\right\\}  \right]$
>
> where $\hat{y}\_{\text{o}} = \arg\max_{k \in [K]} \left[ \phi_{\theta^\ast}\left( \mathbf{h}(x) \right) \right]_k$ and $\kappa = \frac{6 f}{n-2f}\, \left( 1 + \frac{f}{n-2f} \right)$.

---

> > ### Author Response · Authors · 2025-11-20
> > **Response to Reviewer 6LL3 (Part 2/2)**
> >
> > > W4] What about KRUM aggregation as the robust aggregator?
> >
> > **Response:** Thank you for the suggestion. We include the results for Krum below, along with the previous results from Table 2 for comparison ($n=17$, $f=4$ and $\alpha=0.5$).
> > We observe that Krum generally underperforms other aggregators on most attacks and is on average significantly worse than DeepSet-TM.
> >
> > - AG-News:
> >
> > | Aggregation | Logit flipping | SIA-bb       | LMA          | CPA          | SIA          | PGD-cw       | Worst case           |
> > | :----------- | :------------- | :----------- | :----------- | :----------- | :----------- | :----------- | :------------------- |
> > | Mean         | 84.5±0.9       | 81.4±2.2     | **81.2±2.2** | 76.4±4.0     | 72.6±4.6     | 54.9±6.7     | 54.9±6.7             |
> > | CWMed        | 78.9±3.0       | 78.4±3.7     | 75.9±5.0     | 74.1±4.5     | 74.4±4.4     | 53.2±7.0     | 53.2±7.0             |
> > | GM           | 83.0±0.7       | 80.7±2.8     | 80.9±2.5     | 76.6±3.6     | 74.0±3.8     | 52.6±7.3     | **52.6±7.3**         |
> > | Krum         | 74.2±6.3       | 76.7±3.5     | 73.6±3.4     | 73.1±3.6     | 74.1±4.4     | 52.6±6.7     | **52.6±6.7**         |
> > | CWTM         | 84.3±1.0       | 81.4±2.2     | 80.2±2.7     | 76.4±4.0     | 72.6±4.6     | 55.3±7.5     | 55.3±7.5             |
> > | DeepSet-TM   | **85.7±0.4**   | **81.6±1.6** | 79.2±1.3     | **77.5±1.2** | **80.1±1.6** | **83.2±1.4** | **77.5±1.2**         |
> >
> > - CIFAR-100:
> >
> > | Aggregation | Logit flipping | SIA-bb       | LMA          | CPA          | SIA          | PGD-cw       | Worst case           |
> > | :----------- | :------------- | :----------- | :----------- | :----------- | :----------- | :----------- | :------------------- |
> > | Mean         | **78.8±0.7**   | 72.8±1.0     | 66.6±0.3     | 66.4±0.3     | 56.0±1.0     | 39.2±1.2     | 39.3±1.3             |
> > | CWMed        | 65.8±1.3       | 62.3±1.1     | 66.1±1.2     | 66.1±1.2     | 62.9±1.1     | 41.7±1.3     | 41.7±1.3             |
> > | GM           | 75.4±0.9       | 71.5±1.3     | 75.4±0.8     | 75.3±0.8     | 59.6±1.0     | 39.0±1.3     | 39.0±1.3             |
> > | Krum         | 54.7±1.2       | 69.3±1.1     | 71.5±0.7     | 71.5±0.7     | **66.0±0.9** | 25.9±1.5     | **25.9±1.5**         |
> > | CWTM         | 74.8±1.2       | 71.5±1.3     | 74.9±1.1     | 74.8±1.0     | 60.8±0.9     | 44.9±1.3     | 44.9±1.3             |
> > | DeepSet-TM   | 78.0±0.3       | **74.7±0.8** | **76.0±0.2** | **76.4±0.5** | 63.7±0.5     | **49.6±0.5** | **49.6±0.5**         |
> >
> > - CIFAR-10:
> >
> > | Aggregation  | Logit flipping | SIA-bb       | LMA          | CPA          | SIA          | PGD-cw       | Worst case
> > | :-----       | :-----------   | :----------- | :----------- | :----------- | :----------- | :----------- | :------------------- |
> > | Mean         | 65.4±2.1       | 59.8±1.1     | 58.7±3.2     | 55.6±4.0     | 42.7±3.9     | 24.6±4.9     | 24.6±4.9             |
> > | CWMed        | 56.7±4.3       | 53.3±2.0     | 53.8±2.5     | 52.3±2.9     | 49.3±3.2     | 27.8±4.7     | 27.8±4.7             |
> > | GM           | 63.9±2.3       | 59.1±1.4     | **63.3±2.3** | **59.7±3.2** | 45.3±3.7     | 25.4±4.8     | 25.4±4.8             |
> > | Krum         | 57.3±3.4       | 55.7±2.3     | 56.8±3.1     | 56.4±3.1     | **52.1±2.8** | 23.1±4.6     | **23.1±4.6**         |
> > | CWTM         | 63.3±2.7       | 59.4±1.3     | 62.5±2.7     | **59.7±3.2** | 44.8±3.7     | 27.2±5.1     | 27.2±5.1             |
> > | DeepSet-TM   | **67.6±0.8**   | **62.6±1.6** | 61.0±4.4     | 59.4±4.7     | 51.4±2.2     | **48.2±4.2** | **48.2±4.2**         |

---

### Official Review · Reviewer_5CEW · 2025-11-01

**Soundness:** 3
**Presentation:** 3
**Contribution:** 3
**Rating:** 4
**Confidence:** 5

**Summary:**

The paper formally define and analyze the problem of robust federated inference, where a central server aggregates predictions from multiple local models in the presence of adversarial clients. They introduce a method combining adversarial training with test-time robust aggregation (e.g., trimmed mean), which improves robustness without increasing training cost. The proposed DeepSet-TM method outperforms existing robust aggregation techniques.

**Strengths:**

1. Systematically address robustness in federated inference (as opposed to training), with both theoretical and empirical contributions.
2. The DeepSet-based model with adversarial training and test-time robust aggregation is effective.
3. The consistent performance gains as the number of clients and adversaries increases

**Weaknesses:**

1. Adversarial training with DeepSet, while more efficient than full combinatorial search, is still computationally intensive, especially as n grows.
2. The analysis assumes f < n/2, which is common but may not hold in highly adversarial real-world settings.
3. Some privacy concerns. The work focuses on robustness but does not discuss potential privacy implications of sharing probits or using server-side models.

**Questions:**

See weakness.

---

> ### Author Response · Authors · 2025-11-20
> **Response to Reviewer 5CEW**
>
> **We appreciate the reviewer’s comments and provide responses below.**
>
> > W1] Adversarial training with DeepSet, while more efficient than full combinatorial search, is still computationally intensive, especially as n grows.
>
> **Response:** Thank you for raising this point. We agree that adversarial training remains computationally expensive. In our main setting ($n=17$, $f=4$), adversarial training induces a wall-clock training time for DeepSet of 1h26m to 1h40m depending on the dataset. This overhead stems from two components: (i) sampling adversarial client subsets and (ii) running $S=50$ PGD iterations. Importantly, this is purely a one-time server-side cost, and inference remains extremely fast (about 0.3ms per batch of 64 samples on each dataset when including CWTM in DeepSet).
>
> Our work targets cross-silo settings, where the robust aggregator is trained infrequently. In such contexts, we believe a 1-2 hour one-time training cost to be generally acceptable, especially given the improved robustness.
>
> However, we agree that exploring techniques to reduce the training cost is an interesting direction for future work. In this paper, we introduce a strong foundation that demonstrates adversarial training is *effective* for robust federated inference, and provide a setting for subsequent research to build upon. We will add a discussion on the computational cost of adversarial training in the paper.
>
> ----
>
> > W2] The analysis assumes f < n/2, which is common but may not hold in highly adversarial real-world settings.
>
> **Response:** This is a valid criticism that applies to robust machine learning and robust statistics at large, and is not a specific weakness of our paper. It is indeed theoretically impossible to guarantee robustness when the number of adversarial clients are in the majority, see the impossibility results in prior work (Allouah et al., 2023; Chen et al., 2017). In the considered problem of Robust Federated Inference, we can try to relax this condition by making further assumptions on the degree of corruption (i.e., disallowing arbitary alteration of probits) and availability of clean labelled data at the server. This constitutes an interesting future work and calls for a new line of research in robust machine learning. The results we present in this paper represent a foundation for future work relaxing these assumptions.
>
> ----
>
> > W3] Some privacy concerns. The work focuses on robustness but does not discuss potential privacy implications of sharing probits or using server-side models.
>
> **Response:** This is a valid concern, but orthogonal to the contributions made in the current paper. We would like to reiterate that the current paper provides the first rigorous formalism for the problem of robust federated inference, laying the foundations for both theoretical and pragmatic aspects of this problem. A systematic study that jointly combines privacy and robustness, while we agree would be highly relevant, is outside the scope of this paper and left to future work.

---

### Official Review · Reviewer_exau · 2025-11-05

**Soundness:** 2
**Presentation:** 3
**Contribution:** 3
**Rating:** 4
**Confidence:** 4

**Summary:**

This paper addresses the problem of robust federated inference, where the goal is to design an aggregator capable of handling mixtures of clean and corrupted predictions from multiple clients in a federated setting. The authors propose a training paradigm that estimates the residual test error through an adversarial loss, enabling robustness against prediction corruption. The method is theoretically grounded and demonstrates strong empirical performance compared to existing baselines.

**Strengths:**

* Good writing, clear motivations
* Theoretically guaranteed proposal with outstanding performance.

**Weaknesses:**

* Theorem 1: How can the bound be controlled or estimated in practice? Since the left-hand side represents a loss function with an arbitrary value, how can we ensure it remains smaller than 1, given that the bound is a probability? The authors should provide a discussion after the theorem to clarify its practical implications.
* Applicability to Regression: How can the proposed framework (loss function, training algorithm, and theoretical results) be extended to regression tasks?
* Ablation Study: The authors should include ablation studies on the DeepSet aggregator to justify its necessity. Why are recently published aggregator architectures [1,2] not suitable? Would the findings remain consistent if a stronger or alternative aggregator were used?
* Experimental Settings: The current experimental setup appears somewhat simplified. In realistic scenarios, different clients may face distinct types of attacks. Have the authors evaluated the proposed approach under heterogeneous corruption settings to demonstrate its robustness and practical applicability?

[1] IIse et al., “Attention-based Deep Multiple Instance Learning”, arxiv 2018

[2] Xiang et al., “Exploring low-rank property in multiple instance learning for whole slide image classification”, ICLR’23

**Questions:**

See Weaknesses.

The authors should address the practical implications of theorem 1 and extending experiments / ablation studies.

---

> ### Author Response · Authors · 2025-11-20
> **Response to Reviewer exau (Part 1/2)**
>
> **We thank the reviewer for their valuable comments, and would like to adress their concerns below.**
>
> ----
>
> > W1] Theorem 1: How can the bound be controlled or estimated in practice? Since the left-hand side represents a loss function with an arbitrary value, how can we ensure it remains smaller than 1, given that the bound is a probability? The authors should provide a discussion after the theorem to clarify its practical implications.
>
> **Response:** There seems to be a slight misunderstanding. In the left-hand side, the loss function is not arbitrarily valued but is indeed bounded by 1. This is because the definition of the loss is taking the supremum of an indicator function, as defined in (3). In fact the left-hand side of Theorem 1 measures the probability of the robust estimate mismatching the oracle aggregate, as explained below Lemma 1. The result in Theorem 1 shows that when the margin of the average probit is high, or the point-wise variance of honest probits is low, the robustness gap (measured by this probability) is low. This is also verified in practice in Figure 3 (Appendix E.1).
>
> ----
>
> > W2] Applicability to Regression: How can the proposed framework (loss function, training algorithm, and theoretical results) be extended to regression tasks?
>
> **Response:** The current theoretical results do not directly apply to the robust regression  problem. If one considers the regression problem as estimating the average of the real valued outputs, then it can be shown to have some clear connections  with the problem of robust averaging (which is well studied (Diakonikolas & Kane, 2023)). In contrast, our problem cannot be directly related to robust averaging, as illustrated in the counter example.
>
> The regression problem could also go beyond estimating the average of the real valued outputs by trying to define a better aggregation function for regression (e.g. through using neural network-based aggregations). We believe that this problem remains to be studied and constitutes an interesting future direction for our work. We definitely think that the results we provide for classification constitute a good inspiration, but providing a definitive answer goes beyond the scope of our current paper.
>
> ----
>
> > W3] Ablation Study: The authors should include ablation studies on the DeepSet aggregator to justify its necessity. Why are recently published aggregator architectures [1,2] not suitable? Would the findings remain consistent if a stronger or alternative aggregator were used?
>
> **Response:** Thank you for pointing out these papers. We however believe that these recently published aggregators would not be suitable for our task. Indeed, the two architectures you mention come from multiple-instance learning (MIL) and are meant to aggregate many *high-dimensional* instances (e.g. patches of a WSI [2]), relying on attention to focus on a few key instances.
>
> In contrast, federated inference operates over low-dimensional inputs (K-dimensional probits) and the core challenge is adversarial robustness, where adversarial training is the dominant computational cost. In this context, we chose to use DeepSet because it is permutation-invariant, lightweight and easy to combine with inference-time robust averaging (CWTM). It enables adversarial training at a practical cost, which is central to our contribution. We believe that trying to apply attention-based [1] or low-rank attention MIL [2], even though they are also permutation-invariant, would increase computational cost dramatically and make the approach less tractable.
>
> As pointed out by reviewers h5tr and 6LL3, we provide a good *conceptual* framework for demonstrating that adversarial training plus test-time robust averaging is an effective approach to robust federated inference. Finding other suitable permutation-invariant architectures is an interesting direction for future work. We will add a short discussion in the related work to summarize the points made above.

---

> > ### Author Response · Authors · 2025-11-20
> > **Response to Reviewer exau (Part 2/2)**
> >
> > > W4] Experimental Settings: The current experimental setup appears somewhat simplified. In realistic scenarios, different clients may face distinct types of attacks. Have the authors evaluated the proposed approach under heterogeneous corruption settings to demonstrate its robustness and practical applicability?
> >
> > **Response:** Thank you for suggesting these experiments. In fact, the main reason why we chose to make every adversaries run the same attack is by consistency with the Byzantine machine learning literature, where adversaries always coordinate their attacks (Alistarh et al., 2018; Chen et al., 2017; Farhadkhani et al., 2022).
> >
> > But indeed, it is an interesting point to study settings in which attacks are not the same. Hence we ran a "Mixed" attack in the configuration from Table 2 ($n=17$, $f=4$, $\alpha=0.5$), where each of the 4 adversaries performs a different attack (among Logit-flipping, LMA, CPA and SIA-whitebox). We report the results on each dataset below, as well as the full table for comparison.
> > Across CIFAR-10, CIFAR-100 and AG-News, DeepSet-TM still outperforms the baselines under the Mixed attack (+4,1, +1.8 and +2.8 points respectively, compared to the strongest baseline). These experiments show our method can generalize to heterogeneous adversaries and is not relying on a single shared attack pattern.
> >
> > - CIFAR-10:
> >
> > | Aggregation  | Mixed | Logit flipping | SIA-bb       | LMA          | CPA          | SIA          | PGD-cw        | Worst case
> > | :-----       | :-----------   | :----------- | :----------- | :----------- | :----------- | :----------- | :----------- | :------------------- |
> > | Mean         |  59.4±3.0 | 65.4±2.1        | 59.8±1.1     | 58.7±3.2     | 55.6±4.0     | 42.7±3.9     | 24.6±4.9     |    24.6±4.9             |
> > | CWMed        | 54.6±3.7 | 56.7±4.3          | 53.3±2.0     | 53.8±2.5     | 52.3±2.9     | 49.3±3.2     | 27.8±4.7     |    27.8±4.7             |
> > | GM           | 59.5±2.8  | 63.9±2.3        | 59.1±1.4     | **63.3±2.3** | **59.7±3.2** | 45.3±3.7     | 25.4±4.8     |    25.4±4.8             |
> > | CWTM         | 59.6±3.4 | 63.3±2.7          | 59.4±1.3     | 62.5±2.7     | **59.7±3.2** | 44.8±3.7     | 27.2±5.1     |    27.2±5.1             |
> > | DeepSet-TM   | **63.7±1.5**   | **67.6±0.8**   | **62.6±1.6** | 61.0±4.4     | 59.4±4.7     | **51.4±2.2** | **48.2±4.2** |    **48.2±4.2**         |
> >
> >
> >
> > - CIFAR-100:
> >
> > | Aggregation | Mixed | Logit flipping | SIA-bb       | LMA          | CPA          | SIA          | PGD-cw        | Worst case           |
> > | :----------- | :------------- | :----------- | :----------- | :----------- | :----------- | :----------- | :----------- | :------------------- |
> > | Mean         |   73.1±0.4 | **78.8±0.7**   | 72.8±1.0     | 66.6±0.3     | 66.4±0.3     | 56.0±1.0     | 39.2±1.2        | 39.3±1.3             |
> > | CWMed        |   65.7±2.0 | 65.8±1.3       | 62.3±1.1     | 66.1±1.2     | 66.1±1.2     | 62.9±1.1     | 41.7±1.3       | 41.7±1.3             |
> > | GM           |   71.7±0.8 | 75.4±0.9       | 71.5±1.3     | 75.4±0.8     | 75.3±0.8     | 59.6±1.0     | 39.0±1.3       | 39.0±1.3             |
> > | CWTM         |   72.3±0.8 | 74.8±1.2       | 71.5±1.3     | 74.9±1.1     | 74.8±1.0     | 60.8±0.9     | 44.9±1.3       | 44.9±1.3             |
> > | DeepSet-TM   | **74.9±0.3** | 78.0±0.3       | **74.7±0.8** | **76.0±0.2** | **76.4±0.5** | **63.7±0.5** | **49.6±0.5**  | **49.6±0.5**         |
> >
> > - AG-News:
> >
> > | Aggregation | Mixed | Logit flipping | SIA-bb       | LMA          | CPA          | SIA          | PGD-cw        | Worst case           |
> > | :----------- | :------------- | :----------- | :----------- | :----------- | :----------- | :----------- | :----------- | :------------------- |
> > | Mean         |   81.0±2.6  | 84.5±0.9       | 81.4±2.2     | **81.2±2.2** | 76.4±4.0     | 72.6±4.6     | 54.9±6.7       | 54.9±6.7             |
> > | CWMed        |   78.0±5.1  | 78.9±3.0       | 78.4±3.7     | 75.9±5.0     | 74.1±4.5     | 74.4±4.4     | 53.2±7.0     |   78.0±5.1   | 53.2±7.0             |
> > | GM           |   81.1±2.3  | 83.0±0.7       | 80.7±2.8     | 80.9±2.5     | 76.6±3.6     | 74.0±3.8     | 52.6±7.3       | **52.6±7.3**         |
> > | CWTM         |   80.9±2.6  | 84.3±1.0       | 81.4±2.2     | 80.2±2.7     | 76.4±4.0     | 72.6±4.6     | 55.3±7.5       | 55.3±7.5             |
> > | DeepSet-TM   | **83.9±0.9** | **85.7±0.4**   | **81.6±1.6** | 79.2±1.3     | **77.5±1.2** | **80.1±1.6** | **83.2±1.4**  | **77.5±1.2**         |

---

> > > ### Comment · Reviewer_exau · 2025-11-26
> > >
> > > Thanks for the rebuttal.  Hera are a few follow-up questions:
> > >
> > > 1. Based on the authors’ response, the DeepSet aggregator appears to be one of possible choices within the proposed framework. Could the authors provide an additional table demonstrating that the framework can generalize to different aggregators? Otherwise, the proposed solution seems to be designed for DeepSet.
> > >
> > > 2. The authors note that DeepSet was selected for its efficiency and low computational cost. However, in practice - especially in domains such as healthcare - users may be willing to incur higher computational cost for improved performance. In such scenarios, would replacing DeepSet lead to performance gains? More broadly, can the proposed framework maintain a favorable efficiency - performance tradeoff and generalization when using different aggregators?
> > >
> > > 3. In the provided table, the proposed method appears to underperform on the “LMA” attack, even falling below simple aggregation methods like “Mean” on AG-News. Is this a limitation of DeepSet-TM? Which components might be responsible for this behavior?

---

> > > > ### Author Response · Authors · 2025-11-30
> > > > **Response to Reviewer exau**
> > > >
> > > > > 1. and 2. Based on the authors’ response, the DeepSet aggregator appears to be one of possible choices within the proposed framework. Could the authors provide an additional table demonstrating that the framework can generalize to different aggregators? Otherwise, the proposed solution seems to be designed for DeepSet. / The authors note that DeepSet was selected for its efficiency and low computational cost. However, in practice - especially in domains such as healthcare - users may be willing to incur higher computational cost for improved performance. In such scenarios, would replacing DeepSet lead to performance gains? More broadly, can the proposed framework maintain a favorable efficiency - performance tradeoff and generalization when using different aggregators?
> > > >
> > > > **Response:** Thank you for raising the question of using alternative aggregators apart from DeepSet. We would like to re-emphasize that our framework itself is fully general: it only requires the chosen aggregator to be permutation-invariant. In practice, however, identifying permutation-invariant architectures that remain computationally feasible and perform well is non-trivial.
> > > >
> > > > We based our analysis on DeepSet because it combines: (1) permutation invariance, (2) a lightweight encoder/decoder structure that keeps adversarial training tractable and inference cheap, and (3) a functional form that lets us apply CWTM to encoder outputs at inference, which we show yields robustness guarantees independent of the degree of corruption $\delta$ (see Theorem 2 and Appendix G). In fact, identifying DeepSet as a candidate aggregation for our framework is in itself one of our contributions (in light of the difficulty of finding such aggregators).
> > > >
> > > > It is maybe possible that some slightly heavier architectures could offer higher performance for an acceptable increase in computational cost in particular application domains. However, it is not obvious which alternatives would simultaneously preserve all the properties highlighted above. Additionally, prior work in the non-robust setting (Allouah 2024a, Section 3.6) has investigated this idea and shown that more complex architectures do not necessarily lead to accuracy improvements. A more systematic study of alternate aggregators is valuable but non-trivial, and beyond the scope of this submission. We will include a Future Work section that explicitly frames this as a possible future research direction and clearly states that DeepSet can in principle be replaced within our general framework.
> > > >
> > > >
> > > > ---
> > > >
> > > > > 3. In the provided table, the proposed method appears to underperform on the “LMA” attack, even falling below simple aggregation methods like “Mean” on AG-News. Is this a limitation of DeepSet-TM? Which components might be responsible for this behavior?
> > > >
> > > > **Response:** Results in Table 2 and Tables 5-7 (Appendix E.2.2) do not indicate that DeepSet-TM underperforms static aggregations on LMA. In fact, it is always either stronger or within the uncertainty ranges (standard deviation over seeds) of the strongest defense against this attack, including on AG-News.
> > > >
> > > > In LMA, the adversary sets the probit for the least likely class to 1 and the rest to 0. This produces extreme outliers that static aggregators *are specifically designed to filter*. By contrast, simply setting 1 to the second most likely class instead of the least likely (as in SIA) significantly reduces their robustness since adversarial outputs become much closer to the honest outputs, *without requiring any additional adversarial knowledge*. And in this scenario, DeepSet-TM consistently outperforms static aggregations, across all our experiment settings.
> > > >
> > > > Note, however, that what we focus on in this paper is overall (worst-case) robustness. Being occasionally strong on a single attack does not constitute a proof of robustness. DeepSet-TM offers uniformly strong performance across the full suite of attacks, whereas other static aggregations fail in comparison for most of the settings we tested. Overall, LMA is a comparatively weak attack and should not be considered in isolation when comparing aggregators.

---

### Author Response · Authors · 2025-12-03
**Authors' Official Summary**

We thank all reviewers for their thoughtful assessments and constructive feedback. Several explicitly emphasised the originality and significance of our work. Reviewer **h5tr** described the contribution as *“high[ly] original and significant”*, stressing the value of formally defining robust federated inference and exposing the argmax discontinuity counterexample. Reviewer **6LL3** noted that the paper *"makes an original theoretical contribution by providing the first formal robustness framework for federated inference, a setting previously treated mainly empirically"*, while Reviewers **exau**, **5CEW** and **Dfp1** highlighted DeepSet-TM's empirical strength and the soundness of our theoretical foundation.

We summarise below how we addressed key concerns.
## 1. Theoretical guarantees for DeepSet-TM
**Concern:** Lack of an analogue of Theorem 1 (robustness gap) for non-linear aggregators like DeepSet.

**Response:** We added **Theorem 2**, extending the robustness-gap analysis to DeepSet-TM. The bound explicitly incorporates point-wise dissimilarity, probit margin and Lipschitz constants. The proof is in Appendix F.
## 2. Role and incorporation of CWTM (ROBAVG)
**Concerns:** Why ROBAVG reduces sensitivity; why it is used only at inference; effect on optimality.

**Response:** We added **Appendix G**, formally showing that ROBAVG strictly reduces sensitivity to adversarial probits when honest-probit variance is small, and that it makes the robustness gap **independent of the degree of corruption**. We acknowledge the discrepancy between the training objective and the inference operator, but made this choice for computational efficiency. We state this as a limitation and a direction for future work.
## 3. Practicality of the approach
**Concern:** Practicality of DeepSet-TM in real deployments.

**Response:** We reported **inference costs**: static aggregators take 0.05–0.14 ms per batch; DeepSet 0.22–0.26 ms; DeepSet-TM 0.30–0.32 ms. **Training times** with adversarial training are 86–100 min. As this is a one-time cost in cross-silo settings, we believe the overhead is acceptable given the added robustness.
## 4. Generality of the framework and choice of aggregator
**Concern:** Whether our framework is tied to DeepSet;
 suitability of heavier permutation-invariant architectures (suggested by Reviewer exau).

**Response:** The framework is **fully general**: any permutation-invariant aggregator can replace DeepSet. We chose DeepSet for its (i) permutation invariance; (ii) lightweight structure that enables practical adversarial training; (iii) compatibility with inference-time CWTM. MIL-style attention networks are far heavier and would make adversarial training substantially less tractable. We stated that exploring heavier yet tractable architectures is promising future work.
## 5. Additional experiments
**Concerns:** Adversaries performing different attacks; higher data heterogeneity; comparison with Krum; behaviour under LMA.

**Response:**
- We ran **heterogeneous “Mixed” attacks** where each adversary runs a different attack. DeepSet-TM consistently outperforms all baselines across datasets (**+4.1**, **+1.8** and **+2.8 points** over the strongest).
- We evaluated **extreme heterogeneity** with $\alpha=0.1$. Despite severe degradation of clean accuracy (some clients see only 1–2 classes), DeepSet-TM remains superior (average **+9.2 points** against the strongest baseline for each attack).
- We added **Krum** results, showing its comparatively poor performance.
- We clarified that **DeepSet-TM remains either best or statistically tied with the best method under LMA** in all experiments (Table 2 and 5-7). Static aggregations sometimes appear strong on LMA because it creates extreme outliers they are designed to filter, but they fail on attacks that mimic honest clients without additional adversarial knowledge (e.g., SIA). We underlined that **overall worst-case robustness** is the key metric, and DeepSet-TM leads by a large margin.
## 6. Scope, threat model, privacy
**Concerns:** Behaviour under data poisoned/backdoored clients; $f < n/2$; privacy concerns.

**Response:**
- Our threat model already allows arbitrary outputs from up to $f$ clients, covering backdoored or poisoned models.
- We clarified that $f<n/2$ matches impossibility results in Byzantine ML and could only be avoided by relaxing the threat model, which is left to future work.
- We explained that privacy is orthogonal but important, and will require future work.

### Closing remark
We believe the reviewers viewed the paper as original, theoretically grounded and empirically strong. We added new theorems, proofs, experiments and clarifications to address all conceptual, theoretical, and empirical concerns. We also documented limitations and future directions.

We hope this clearly demonstrates both the novelty and the rigour of the work, and we thank the reviewers again for the constructive dialogue that helped improve the paper.

---

### Meta-Review · Area_Chair_5ZrJ · 2026-01-05

**Summary:**

This paper addresses the problem of robust federated inference, where a server aggregates predictions from different local models, some of which may be adversarially corrupted. They formalize the robust federated inference problem and analyze averaging-based aggregators. They recast robust federated inference as an adversarial machine learning problem over probability vectors. Hence, they propose DeepSet-TM, a permutation-invariant neural network (nonlinear) aggregator that combines adversarial training with robust averaging at test time.

**Positive Points:**
- Novel formalization of robust federated inference with solid theoretical foundations, framing aggregation as adversarial learning over probit vectors.
- Effective combination of adversarial training and test-time robust aggregation
- Strong empirical performance across multiple datasets (CIFAR-10, CIFAR-100, AG-News) and attack types
- Clear writing and well-motivated problem formulation

The paper makes a significant contribution by providing the first treatment of robust federated inference.
The reviewers raised very good points acknowledging the shortcomings of the first version of the paper, which have been addressed by the authors in their responses (see my summary below). My assessment of the final version is positive, and I recommend the paper for acceptance.

**Reviewer Concerns:**

**Negative Points & Rebuttal Responses:**

[Addressed] **Theoretical results are only valid for linear aggregation** (6LL3, Dfp1): added **Theorem 2** which extends the robustness gap analysis specifically to the non-linear DeepSet-TM aggregator.

[Addressed] **Computational efficiency concerns** (5CEW, exau, h5tr): provided detailed inference latency (0.3ms/batch) and training time analysis (one-time cost of 86-100 min).

[Addressed] **Simplified attack patterns and data distribution:** (exau, h5tr): new experiments with **"Mixed" attacks** and **extreme data heterogeneity** ($\alpha=0.1$). DeepSet-TM maintained its superiority.

[Addressed] **Krum comparison** (6LL3): Added results showing Krum underperforms.

[Partially addressed] **Alternative aggregator architectures** (exau): explained DeepSet's advantages but acknowledged exploring heavier architectures as future work.

[Partially addressed] **Training/Inference mismatch: why ROBAVG is used only at inference time** (Dfp1): added Appendix G with formal analysis, but acknowledged the training/inference discrepancy as a limitation.

[Acknowledged] **Privacy concerns** not discussed, acknowledged as orthogonal future work.

[Acknowledged] **f < n/2 assumption may not hold in extreme settings** (5CEW): authors noted this is a fundamental limitation in robust statistics, with impossibility results in prior work.

**Reviewer Scores:**

- exau 4-> 6

- Dfp1 4-> 6

- 5CEW 4

- 6LL3 8

- h5tr 8

---

### Decision · Program_Chairs · 2026-01-26

Accept (Poster)